# Role of Vitamin D in Head and Neck Cancer—Immune Function, Anti-Tumour Effect, and Its Impact on Patient Prognosis

**DOI:** 10.3390/nu15112592

**Published:** 2023-05-31

**Authors:** Katarzyna Starska-Kowarska

**Affiliations:** 1Department of Physiology, Pathophysiology and Clinical Immunology, Medical University of Lodz, Żeligowskiego 7/9, 90-752 Lodz, Poland; katarzyna.starska@umed.lodz.pl; Tel.: +48-604-541-412; 2Department of Clinical Physiology, Medical University of Lodz, Żeligowskiego 7/9, 90-752 Lodz, Poland; 3Department of Otorhinolaryngology, EnelMed Center Expert, Lodz, Drewnowska 58, 91-001 Lodz, Poland

**Keywords:** anti-cancer therapy, apoptosis, cancer pathogenesis, cancer prevention, cellular signalling, cell-cycle progression, head and neck cancer (HNC), head and neck squamous cell carcinoma (HNSCC), vitamin D

## Abstract

Head and neck squamous cell carcinoma (HNSCC) describes a heterogeneous group of human neoplasms of the head and neck with high rates of morbidity and mortality, constituting about 3% of all cancers and ~1.5% of all cancer deaths. HNSCC constituted the seventh most prevalent human malignancy and the most common human cancer in the world in 2020, according to multi-population observations conducted by the GLOBOCAN group. Since approximately 60–70% of patients present with stage III/IV neoplastic disease, HNSCC is still one of the leading causes of death in cancer patients worldwide, with an overall survival rate that is too low, not exceeding 40–60% of these patients. Despite the application of newer surgical techniques and the implementation of modern combined oncological treatment, the disease often follows a fatal course due to frequent nodal metastases and local neoplastic recurrences. The role of micronutrients in the initiation, development, and progression of HNSCC has been the subject of considerable research. Of particular interest has been vitamin D, the pleiotropic biologically active fat-soluble family of secosteroids (vitamin-D-like steroids), which constitutes a key regulator of bone, calcium, and phosphate homeostasis, as well as carcinogenesis and the further development of various neoplasms. Considerable evidence suggests that vitamin D plays a key role in cellular proliferation, angiogenesis, immunity, and cellular metabolism. A number of basic science, clinical, and epidemiological studies indicate that vitamin D has multidirectional biological effects and influences anti-cancer intracellular mechanisms and cancer risk, and that vitamin D dietary supplements have various prophylactic benefits. In the 20th century, it was reported that vitamin D may play various roles in the protection and regulation of normal cellular phenotypes and in cancer prevention and adjunctive therapy in various human neoplasms, including HNSCC, by regulating a number of intracellular mechanisms, including control of tumour cell expansion and differentiation, apoptosis, intercellular interactions, angio- and lymphogenesis, immune function, and tumour invasion. These regulatory properties mainly occur indirectly via epigenetic and transcriptional changes regulating the function of transcription factors, chromatin modifiers, non-coding RNA (ncRNAs), and microRNAs (miRs) through protein-protein interactions and signalling pathways. In this way, calcitriol enhances intercellular communication in cancer biology, restores the connection with the extracellular matrix, and promotes the epithelial phenotype; it thus counteracts the tumour-associated detachment from the extracellular matrix and inhibits the formation of metastases. Furthermore, the confirmation that the vitamin D receptor (VDR) is present in many human tissues confirmed the physiopathological significance of vitamin D in various human tumours. Recent studies indicate quantitative associations between exposure to vitamin D and the incidence of HNC, i.e., cancer risk assessment included circulating calcidiol plasma/serum concentrations, vitamin D intake, the presence of the VDR gene polymorphism, and genes involved in the vitamin D metabolism pathway. Moreover, the chemopreventive efficacy of vitamin D in precancerous lesions of the head and neck and their role as predictors of mortality, survival, and recurrence of head and neck cancer are also widely discussed. As such, it may be considered a promising potential anti-cancer agent for developing innovative methods of targeted therapy. The proposed review discusses in detail the mechanisms regulating the relationship between vitamin D and HNSCC. It also provides an overview of the current literature, including key opinion-forming systematic reviews as well as epidemiological, prospective, longitudinal, cross-sectional, and interventional studies based on in vitro and animal models of HNSCC, all of which are accessible via the PubMed/Medline/EMBASE/Cochrane Library databases. This article presents the data in line with increasing clinical credibility.

## 1. Introduction

Head and neck squamous cell carcinoma (HNSCC) is the most common histological type of a heterogeneous group of malignant neoplasms originating from the mucosa of the upper respiratory system and the gastrointestinal tract [1,2]. A typical feature of all HNSCC tumours of various origins is their large diversity in terms of morphological and molecular changes, and thus also the clinical phenotype of their carcinomas. In general, men are at a 2- to 4-fold higher risk than women of developing HNSCC. The diverse biological features of HNSCC largely determine the clinical course of the oncological disease and the analysed parameters, i.e., factors inducing the development of neoplastic lesions, their progression, and further advancement of the cancer, as well as mortality rates, treatment results, and long-term prognosis, as well as local and nodal recurrences and patient survival [3]. According to the latest data from Global Cancer Statistics (GLOBCAN 2020), HNSCC constitutes the seventh most common cancer in the world, developing in over 660,000 patients and causing nearly 450,000 deaths. The number of HNSCC diagnoses has increased to almost 900,000 cases per year in 2020, which, according to population data, accounts for ~4.5% of human malignancies. Researchers predict that by 2030, the incidence rate of HNSCC will increase by 30%, resulting in 1.08 million new cancer cases per year [GLOBOCAN 2020; gco.iarc.fr/today (accessed on 31 March 2023)] [4,5]. Unfortunately, the five-year survival of patients with head and neck squamous cell carcinoma (HNSCC) has not changed dramatically over the last decades despite the use of various therapeutic modalities, including surgery and/or chemoradiation.

Well-characterised risk factors for the development of HNSCC include smoking and alcohol consumption, as well as exposure to environmental pollutants and other specific carcinogen-containing products, which traditionally account for 90% of cases of this type of cancer, according to the National Comprehensive Cancer Network (NCCN) [6,7]. Among these, the most widely distributed high-risk factors are tobacco and alcohol consumption; although, in some populations, these can include areca nut products, including ‘betel quid’ and betel leaf, which are chewed, as well as slaked lime, poor oral hygiene, and diets lacking in vegetables [7]. Alcohol causes epithelial atrophy and promotes the failure of cell membrane lipids, which facilitates the action of tobacco-derived carcinogens. Strong spirits are believed to act as co-carcinogens that increase the incidence of HNSCC in a dose-dependent way in concomitant smokers. Additionally, the main metabolite of ethanol, acetaldehyde, is also a strong mutagen. The interaction of the above-mentioned common risk factors increases the risk of HNSCC diagnosis up to 35 times [8,9,10]. Studies clearly show that tar and tobacco smoke contain ~7000 toxic compounds, of which 60 are active carcinogens. Tobacco components, such as benzo[a]pyrene, a crucial polycyclic aromatic hydrocarbon (PAH), as well as tobacco-specific nitrosamines (TNAs) and N′-nitrozonornicotine (NNN), increase the risk of cancer initiation and further development, as well as the formation of metastases to lymph nodes and distant metastases, by inducing a phenotype similar to the epithelial-mesenchymal transition (EMT) [8,9,10]. Unfortunately, a significant population of patients is diagnosed with HNSCC at a very advanced stage of cancer, i.e., in the III and IV WHO classifications; as such, despite rapid progress in recent years in the field of diagnostic and surgical techniques and the use of combination therapies, the parameters of global overall survival or 5-year tumour-free survival do not exceed 40–60% in the HNSCC population [11,12]. According to the Cancer Genome Atlas Network (TCGA), tobacco-dependent cancers demonstrate abnormalities in four groups of genes, viz., those regulating the cell cycle (*CDKN2A* and *CCND1*), those determining cell proliferation and survival (*TP53*, *HRAS*, *PIK3CA,* and *EGFR*), those controlling cell differentiation (*NOTCH1*), and a gene regulating the Wnt signalling pathway [3]. Moreover, the deprivation of chromosome 9p, responsible for the reduction of p16 (CDKN2A) expression, and the replication of chromosome 7p, causing overexpression of the epidermal growth factor receptor (EGFR), are also common [13,14]. Proto-oncogene mutations are also well described, i.e., *c-myc*, *RAS* family genes (*KRAS*, *HRAS*, *NRAS*, *ERB-B*, *BRAF*, *HER-2*, *c-KIT*, *BCL-2*, *STAT3),* tumour suppressor genes called anti-oncogenes (*RB1*, *P53*, *PTEN*, *CDKN2A*, *INK4*), and genes controlling pro-inflammatory tumour microenvironment [3,15].

In recent years, it has been shown that squamous cell carcinoma of the oral cavity and oropharynx associated with infection with oncogenic strains of human papillomavirus (HPV), primarily HPV-16 and, to a lesser extent, HPV-18, is a biologically and clinically unique neoplastic process that accounts for as much as 38–80% of new diagnoses of HNSCC in an oropharyngeal location (OPSCC) [16,17]. OPSCC contains the HPV genome, which is believed to cause cancer through its oncoproteins E6 and E7. Epidemiological studies from recent years clearly indicate that the incidence of HPV-related HNSCC, especially oropharyngeal squamous cell carcinoma (OPSCC), has increased both in Europe and globally [18,19]. The molecular changes occurring in OPSCC have been attributed to the incorporation of HPV genomic DNA into the genetic material of the host epithelial cells. The resulting heightened level of HPV16/18 viral antigens E5, E6, and E7, which are oncoproteins/oncogenic factors, promotes initiation and further malignant development. HPV16/18 E6 and pRB1 by HPV16/18 E7 degrade cell cycle regulatory proteins such as the p53 tumour suppressor protein, thus disrupting important intracellular signal pathways that regulate the cell cycle [9,20,21,22]. In addition, HPV16/18 E6 protein affects the deregulation of the *c-myc* oncogene, thus activating the transcription of the human telomerase catalytic subunit (hTERT), regulating the phenomenon of tumour cell immortalization, and disrupting the function of CDK, cyclins, and E2F transcription factors. It has also been found that myc reverses the inhibitory activity of CDK p27^KIP1^ and p21^CIP1/WAF1^ [15,21].

According to TCGA, typical gene mutations for HPV-associated cancers include *PIK3CA*, *FGFR*, *DDX3X*, and *CYLD* [23]. HPV-induced tumours are also characterised by the occurrence of amplification on chromosome 3q and the loss of chromosomes 11q, 13q, 14q, 16p, and 16q [13,14]. Importantly, survival rates significantly differ depending on tumour HPV status. Significantly, patients with high-risk HPV OPSCC have more satisfactory survival rates than patients with unrelated HPV OPSCC, regardless of the mode of treatment [16]. Typically, HPV^+ve^ OPSCC is significantly more often diagnosed in the early stages. In the HPV-positive OPSCC population, the proportion of relatively younger patients is high. Furthermore, HPV-associated OPSCCs are more susceptible to chemoradiation and immune checkpoint inhibitors (ICI) than non-HPV-associated cancers. This justifies the use of the current de-escalation and de-intensified cure protocols used in clinical trials and highlights the need to improve the stratification strategy for patients with HPV-associated HNSCC, in particular for OPSCC [24,25]. With the growth in understanding of the biological nature of HPV^+ve^ tumours and their relationship to prognostic indicators of HPV-related OPSCC, the traditional staging of HNSCC using the pTNM system has been replaced by the 2017 AJCC/UICC new staging system in the 8th edition of the American Joint Committee on Cancer (AJCC), which proposes a de-intensified treatment protocol for individuals with HPV^+ve^ OPSCC to reduce long-term associated morbidity [26]. It should be noted, however, that recent clinical trials (RTOG 1016 and De-ESCALaTE) indicate a more complex therapeutic problem in patients with HPV-related OPSCC, as a large cohort of patients who received de-intensified treatment had significantly worse results compared with those who received standard care [27,28]. Moreover, patients with HPV-positive multi-regional primary OPSCC and a second primary cancer at any head and neck location showed, inter alia, a lower pT/pN grade compared with those with a single primary tumour [29].

A number of recent basic science, clinical, and epidemiological studies indicate that vitamin D3 has multidirectional biological effects and influences anti-cancer intracellular mechanisms and cancer risk, and that vitamin D dietary supplements have various prophylactic benefits. Data indicates that vitamin D3 activity is mediated by the repression of key intracellular signalling pathways, i.e., NF-κB, AKT, ERK, MAPKp38, JNK kinases, and MAPK subfamilies, and by blocking the PI3K/AKT pathway and cell cycle progression at crucial transition points. These pathways also appear to inhibit proliferation and angiogenesis, thus reducing the efficacy of chemotherapy against HNC.

Moreover, vitamin D3 and its derivatives may also act as antioxidants, anti-inflammatories, immunoprotectors, immune regulators, cellular oncogenic signalling and apoptosis regulators, as well as cell-cycle and angiogenesis controllers. Proteomic profiling studies have also confirmed that vitamin D3 can negatively regulate oncogenic transcription and stimulate a number of key genes encoding antioxidant enzymes, and that it can promote significant changes in other intracellular proteins via micro-RNA activity, a phenomenon essential for HNC cancerogenesis. Recent studies indicate quantitative associations between exposure to vitamin D and the incidence of HNC. Cancer risk assessment included circulating calcidiol plasma/serum concentrations, vitamin D intake, the presence of the VDR gene polymorphism, and genes involved in the vitamin D metabolism pathway. Moreover, the chemopreventive efficacy of vitamin D in precancerous lesions of the head and neck and their role as predictors of mortality, survival, and recurrence of head and neck cancer are also widely discussed. As such, it may be considered a promising potential anti-cancer agent for developing innovative methods of targeted therapy.

The present review extensively discusses the links between vitamin D and HNSCC. It also provides an overview of the current literature, including key opinion-forming systematic reviews as well as epidemiological, cross-sectional, longitudinal, prospective, and interventional studies based on in vitro and animal models of HNSCC.

### 1.1. The Biochemistry and Physiology of Vitamin D

#### 1.1.1. Vitamin D Active Metabolite Synthesis and Metabolism

Naturally occurring vitamin D3 is found in a limited number of dietary sources, including cod liver oil and oily fish. A principal function of vitamin D, a member of the family of fat-soluble secosteroids, is to increase intestinal phosphate, magnesium, and calcium absorption in the intestine. The most important compounds in humans in this family include vitamin D3 (known as cholecalciferol) and vitamin D2 (known as ergocalciferol) [30,31,32]. The first, cholecalciferol, an inactive form, constitutes the main source of vitamin D in the body, and it is synthesised from the skin by exposure to the sun. The second precursor, ergocalciferol, can be used as a source of vitamin D via oral medication or through enriched foods. Although both vitamin D2 and D3 are used as drugs, studies have shown that vitamin D3 yields a higher serum 25OHD2 vitamin level than vitamin D2. In addition, it has been shown that active vitamin D obtained from vitamin D3 has a higher affinity for the vitamin D receptor (VDR) [30]. Calcitriol [1α,25(OH)_2_D3], i.e., the biological active form of vitamin D, is obtained from dietary sources (animal foods, i.e., fish, egg yolks) or medicines (vitamin supplements) and non-dietary sources, namely from skin sunlight exposure to UV-B within just fifteen minutes [33]. In the latter case, optimal vitamin D synthesis is obtained at a wavelength of 297 nm (290–315 nm) [34,35]. The creation of the active form of vitamin D begins with the creation of pro-vitamin D3 in the intestinal epithelium following the oxidation of cholesterol to 7-dehydrocholesterol. The pro-vitamin D3 is then transported to the skin, where, under the influence of ultraviolet radiation (270–300 nm), it is transformed into pre-vitamin D3 in the basal layer of the skin epidermis. Pre-vitamin D3 is then isomerised to vitamin D3 and cholecalciferol in a thermo-sensitive reaction.

The activation of vitamin D3 by transformation to calcitriol happens through two hydroxylation reactions. The first occurs in the liver, where active microsomal and mitochondrial vitamin D 25-hydroxylases (CYP27A1 enzymes) lead to the formation of 25-hydroxyvitamin D3 [25(OH)D3]. In practise, the level of vitamin D in the body is estimated based on the total serum 25(OH)D3 (known as calcidiol or calcifidiol) level because it has the physiologically highest concentration of all forms of vitamin D3. The second hydroxylation is found in the mitochondria of the kidney proximal tubule cells; it is carried out by 1-hydroxylase (CYP27B1), leading to the synthesis of 1α,25(OH)_2_D3: 1α,25-dihydroxyvitamin D3 (known as calcitriol), which also spreads in the circulation bound to DBP. Importantly, the synthesis of calcitriol can also occur in various tissues and cells, such as the skin, immunocompetent cells, parathyroid gland, intestine, breast, keratinocytes, prostate, and neoplastic cells. Vitamin D3 is metabolised for excretion by the CYP24A1 enzyme, which produces the inactive metabolite 24,25(OH)_2_D3: 24,25-dihydroxyvitamin D3. It should be emphasised that 24,25(OH)_2_D3 has the ability to inactivate all other circulating forms of vitamin D [30,34,36,37,38].

#### 1.1.2. VDR-Related Mechanism of Vitamin D Action

Importantly, only ~0.04% of calcitriol is present in peripheral blood as a free hormone. The main metabolite of vitamin D in peripheral blood, calcidiol, is transported in the greatest amount by vitamin D binding protein (DBP; ~85%) and albumin (~15%). Hence, only ~0.03% of calcidiol is in free form and can be taken up by target cells in this chemical form. Moreover, calcitriol demonstrates 10–100 times lower affinity for DBP compared with calcidiol but is sufficient to be its main transporter [39]. Both 25(OH)D3 (calcidiol) and 1α,25(OH)_2_D3 (calcitriol) are circulating forms that bind to a specific transport protein known as vitamin D binding protein (DBP). Calcitriol pairs with the vitamin D receptor (VDR) in target cells and mediates intracellular effects by an additional ligand (non-genomic) binding pocket or genomic effects by the genomic pocket. All secosteroids, including vitamin D3, calcidiol, and calcitriol, as well as various other vitamin D metabolites, are absorbed by cells by passing directly across the plasma membrane. In the first step, after entering the target cell membrane, active vitamin D binds to the ligand-binding domain of its own receptor in the cytoplasm of the target cell. Calcitriol is bound to the ligand-binding domain, followed by the H12 alpha-helix (AF-2) region of H12, which is the terminal part of the ligand-binding structure [40]. VDRs are found in over 400 different tissues (www.proteinatlas.org/ENSG00000111424-VDR/tissue; accessed on 31 March 2023) [41]. After binding, the VDR then heterodimerises with the retinoid X receptor (RXR). All three forms of RXR (α, β, γ) can bind to VDR without significant differences in metabolic activity. In the next step, the vitamin D3-VDR complex joins the retinoid X receptor α (RXRα), inducing its dimerization. The vitamin D3-VDR-RXRα-cofactor complex then interacts with DNA regulatory elements termed vitamin D response elements (VDREs) in the cell’s DNA, which join to specific regulatory regions that are found in promoter and distal regions of target genes with a high affinity, acting as co-modulators and exerting the genomic effects of VDR, i.e., initiating gene activation or inhibition [42,43,44,45]. The fact that a wide range of specific VDRE genes exist, such as those associated with bone metabolism, detoxification of xenobiotics, drug resistance, cell growth and differentiation, angiogenesis, apoptosis, and immune function, suggests that vitamin D has numerous regulatory and modulatory roles in various organs or tissues [46]. Additionally noteworthy is that endoplasmic reticulum stress protein 57 (ERP57; also known as 1,25D3-MARRS and GRP58) is engaged in protecting calcitriol against DNA damage caused by sunlight [47].

The VDR molecule is a member of the nuclear receptor superfamily, which includes steroids, thyroid hormones, and retinoic acid receptors. It weighs approximately 50–60 kDa, depending on species, and is present in numerous tissues, where it functions as a transcription factor and determines the physiological significance of vitamin D [48]. The VDR gene is located on chromosome 12. The structure of the VDR comprises various functional domains, including a short *N*-terminal domain, two zinc fingers, which determine the specificity of the VDRE, i.e., the DNA binding site, and a highly variable *C*-terminal region, together with a hinge region connecting these domains [37]. The ligand-binding part of the receptor, also called AF-2, is composed of 12 α-helix structures (H1-12; H12 part) and three β-sheet structures (S1-3) and acts as a binding site for co-activators and co-repressors. Co-activator binding initiates transcription [49]. The formation of the vitamin D-VDR-RXR-VDRE complex is followed by the transcription process, which is under the control of coactivators and corepressors. The best-known coactivators are the p160 family (e.g., CBP/p300 and p/CAF), SRC 1-3 (steroid receptor coactivator), and the DRIP complex (vitamin D receptor interacting protein, Mediator); while SRCs bind to AF-2 and exhibit histone acetyltransferase (HAT) activity, the DRIP complex does not. This property suggests that both proteins play complementary roles in the transcription process. The interaction between the two coactivators and the AF-2 part enables the opening of the histone structure and thus facilitates gene expression [43].

The DNA-binding region of the VDR also contains NLS (nuclear localization signal) regions, which are also necessary for proper transcriptional activity. In addition, there is a hinge region between the ligand-binding domains and the DNA-binding domains of the VDR that is responsible for stabilising the molecule [37]. The formation of the DRIP205/MED1 mediator multi-protein complex, also known as MED1, increases the RNA polymerase II activity of the initiation complex. This complex then interacts with the TATA region in the promoter region of the target genes and leads to transcription initiation [50,51]. Corepressors (e.g., SMRT and NCoR) have histone deacetylase activity and inhibit transcription by preventing the histone core from unfolding. It has been shown that VDR not only plays a key role in the well-known functions of vitamin D, i.e., intestinal phosphate absorption, magnesium and calcium homeostasis, skeletal health, mediating inflammation and immune function, and regulating cell proliferation and differentiation; it also acts as an intermediary in oestrogen-related pathways, the renin-angiotensin system, and insulin-like growth factor signalling [34,42,52,53]. Moreover, numerous in vitro and in vivo studies clearly confirm that vitamin D plays a complex role in carcinogenesis and the progression of tumours of different origins [54,55,56].

Calcitriol may also exert non-genomic effects through interaction with a distinct VDR (referred to as a membrane VDR, mVDR). mVDR binds to caveolin-1 on the cell surface and in the perinuclear region [57,58]. Recent studies have shown that membrane-mediated mechanisms are involved in the activation of intracellular signalling molecules such as PI3K, PKC, and MAP kinases that, in turn, trigger additional transcriptional factors like RXR, p63, C-EBPα, C-EBPβ, SP1, SP3, Runx2, and PU.1; these can cooperate with VDR and VDR coregulators to influence 1α,25(OH)_2_D3 responses in target cells [38,53]. Calcitriol is known to suppress NF-κB activity via the physical interaction of the VDR with IκB-β kinase (IKKβ), which blocks NF-κB [59]. Calcitriol rapidly decreases TNFα-induced p65 nuclear translocation and NF-κB activity in a VDR-dependent manner. As a consequence, 1α,25(OH)_2_D3 downregulates various genes, including IL-12, IL-8, MCP-1, PAI-1, angiotensinogen, and microRNA-155 [60,61].

The calcitriol-VDR complex also acts by modulating target proteins (e.g., pro-inflammatory cytokines IFN-α and TNF-α) by binding to transcription factors such as STAT1 and IKKβ and thus modulating inflammatory mediators [59]. It has been reported that VDR signalling attenuates TLR-mediated inflammation by enhancing negative feedback inhibition [60,62]. However, it has also been stated that the VDR can downregulate gene transcription by directly interacting with other regulatory proteins, such as β-catenin and CREB, through VDRE-independent mechanisms. Moreover, VDR can act in unison with transcription factors and kinases, independent of calcitriol binding, to regulate immune and anti-viral responses and apoptosis [57,58]. Thus, activated VDR leads to the induction of transcription of non-coding RNA and histone-modifying enzymes, affecting the expression of genes located in the genome far away from the major vitamin D target genes. The activated VDR-RXR complex determines the function of over 3000 genes, depending on cell type and physiological conditions. Transcriptional regulation of genes by vitamin D can influence the inhibition of cell proliferation, induction of cell differentiation, immunomodulation, UVB response, ROS response, and alteration of mitochondrial function [63]. Furthermore, the genes coding transcription factors (TFs) activated by calcitriol can in turn regulate the activity of additional sets of genes, which constitutes a secondary genomic response in a highly complex vitamin D signalling cascade. Recently, calcitriol has been described to have regulatory effects on the expression of non-coding RNAs (ncRNAs), i.e., microRNAs (miRNAs), long non-coding RNAs (lncRNAs), and circular RNAs [64,65,66].

#### 1.1.3. Physiological Cell Activities of Vitamin D

Calcitriol raises blood calcium levels by activating multiple mechanisms. It plays a key role in increasing calcium absorption in the intestine, obtaining calcium from bones, and upregulating calcium resorption in the kidney distal tubule, thus indirectly promoting calcium bone incorporation [53,67]. Calcitriol deficiency results in only 10–15% of dietary calcium being absorbed by the gastrointestinal tract [67,68]. Calcitriol also increases circulating calcium levels by suppressing the transcription of calcitonin and PTH. Calcitriol activates osteoblast differentiation and controls the synthesis of proteins such as collagen, alkaline phosphatase, and osteocalcin involved in bone formation. 1α,25(OH)_2_D3 also activates RANKL, a membrane-associated protein in osteoblasts that allows them to transform into osteoclasts [38]. Calcitriol may also exert an anabolic effect in bone, apparently via the VDR in mature osteoblasts, by increasing osteoblast activity and reducing osteoclast activity [67,68]. Calcitriol is also responsible for the formation and activation of osteoclasts via activation of the NF-κB receptor activator ligand (RANKB) [69]. Further, an active form of vitamin D also regulates phosphate levels by increasing intestinal absorption. It is known that approximately 60% of dietary phosphate is absorbed in the absence of calcitriol. The conversion of the inactive metabolite 25(OH)D3 (calcidiol) to the active form 1α,25(OH)_2_D3 (calcitriol) is stimulated by PTH and calcitonin and inhibited by ion regulators, namely FGF-23/Klotho axis and phosphate. It has been shown that the higher level of FGF-23 in osteoclasts results in phosphate excretion in the kidney [70,71]. However, the vitamin D3 endocrine system also regulates about 3% of the human genome, including a few genes involved in cell differentiation, cell-cycle control, and cell function; it also exerts non-calcaemic/pleiotropic effects on extraskeletal target tissues, such as those of the immune and cardiovascular systems, pancreatic endocrine cells, muscle, and adipose tissue [72,73].

The schema of vitamin D synthesis and its systemic metabolism and activity is shown in Figure 1.

##### Genomic Effects (Delayed Response) of Vitamin D

The genomic pathways launched by the VDR remain primary targets for vitamin D for a number of hours or days, with clear clinical implications; this delayed response is mediated via its most potent derivative, calcitriol, and the VDR [74]. The active form of vitamin D3, calcitriol, serves as a ligand for the VDR. The physiological effect of calcitriol is related to the transcriptional function of VDR, which works mainly by controlling the expression of genes whose promoters contain specific DNA sequences known as vitamin D response elements (VDREs). These classically have two half-sites, each with six nucleotides separated by three nucleotides of a non-specific type; this type of VDRE is referred to as DR3, i.e., consisting of direct repeats three nucleotides apart. In the target cell, VDR interacts with nuclear hormone receptors, in particular RXRα, and forms heterodimers that optimise their affinity for VDREs, a distinctive sequence of nucleotides in the promoter of the vitamin D-responsive gene. Intriguingly, the RXR is responsible for maintaining the VDR in the nucleus in the absence of the ligand [38].

The binding of the VDR/RXR complex to the VDREs attracts a complex of proteins known as coactivators to the VDR/RXR complex. The DRIP (Mediator) co-activator complex acts as the link between VDRE, RNA polymerase II, and other proteins in an initiation complex targeting the TATA box or other transcription regulatory elements. SRC1-2 co-activators activate histone acetyltransferases (HATs) in the gene, leading to the opening of its structure and allowing transcription [49,50,51]. SRC1-3 molecules bind to the VDR via the NR box with the central motif LxxLL (L, leucine; x, any amino acid). Each SRC family member contains three well-conserved NR boxes in the region critical for nuclear hormone receptor binding. Likewise, the DRIP complex also has NR regions with LxxLL motifs consisting of 15 or more amino acids [75]. However, DRIP205/MED1 plays a critical role in binding the complex to VDR. It contains two NR boxes. Different NR regions in co-activators characterise specificity for different nuclear hormone receptors. Importantly, these coregulators are not specific to the VDR but interact with a large number of other transcription factors [38].

Calcitriol can also inhibit gene transcription through its VDR. Such regulation may result from the direct association of VDRs with negative VDREs, i.e., parathyroid hormone gene or parathyroid hormone-related peptide gene (PTH and PTHrP), which mediate negative regulation of gene transcription by calcitriol and bind the vitamin D3 receptor. Inhibition can also occur via indirect mechanisms. For example, calcitriol inhibits interleukin-2 secretion by blocking the NFATp/AP-1 transcription factor complex, and 1-αhydroxylase (encoded by *CYP27B1*) may be inhibited indirectly via a mechanism involving VDR binding to VDIR [76,77].

The active form of vitamin D, calcitriol, has also been shown to indirectly control mitochondrial activity through receptor-dependent regulation of the genes engaged in the response to oxidative stress [78,79]. It has been demonstrated that calcitriol inhibits the transcription of genes associated with oxidative phosphorylation and fusion or fission and enhances the expression of those related to mitophagy, reactive oxygen species (ROS) defence, and epigenetic gene regulation [79]. The transcription of genes responding to the response to ROS by 1α,25-dihydroxyvitamin D3 is mediated by a key transcription factor, nuclear factor-erythroid 2-related factor 2 (NRF2) [80,81]. Moreover, since the VDR is essential for mitochondrial action, disturbances in the VDR affect mitochondrial membrane potential and enhance ROS production, resulting in mitochondrial damage [82]. Furthermore, VDR deletion also resulted in increased expression of respiratory chain elements, i.e., cyclooxygenases-2 and -4 (COX-2 and -4), ATP synthase subunits (6MT-ATP6 and ATP5B), and subunits II and IV of cytochrome c oxidase [82,83].

In summary, the control of VDR activity may depend on signalling pathways originating in receptors at the cell plasma membrane as well as within the nucleus. Moreover, stimulated VDR enhances the secondary effects by activating the transcription of non-coding RNA and histone-modifying enzymes, regulating genes that are far from the primary vitamin D target genes.

##### Non-Genomic Effects (Fast Response) of Vitamin D

Vitamin D has a multitude of pleiotropic functions that cannot be fully explained by the activation of classical signalling or by VDR-dependent transcriptional modulation. In vitro and in vivo studies indicate that vitamin D elicits a rapid and non-genomic response, with the physiological effects of vitamin D being observed within minutes or even seconds after stimulation. Hence, they cannot be determined by stimulation of gene transcription and subsequent translation alone [84,85]. The non-genomic vitamin D3-dependent effect occurs via activation of messenger-mediated pathways without the involvement of cell receptors. An active form of vitamin D, calcitriol, 1α,25(OH)_2_D3, acting through membrane receptors, directly regulates the activation or distribution of intracellular proteins forming ion transport channels (Ca^2+^, calcium, and Cl^−^, chloride channels) and various enzymes in osteoblast, liver, muscle, and intestinal cells: protein kinase C (PKC), phospholipase C (PLC), calcium/calmodulin-dependent protein kinase type II gamma chain (CAMK2G) activation, and cyclic adenosine 3′,5′-monophosphate (cAMP). Upon binding to the membrane receptor, calcidiol stimulates the conversion of GDP in the α subunit of the G protein to GTP and thus activation. The α subunit of the G protein binds to phospholipase C (PLC). Activated PLC converts phosphoinositol bisphosphate (PIP2) into inositol triphosphate (IP3) and diacylglycerol (DAG). Calcitriol causes a quick influx of calcium (transcaltachia) through epithelial cells and its accumulation in various cells of the body. This calcium release from the endoplasmic reticulum occurs by activation of the IP3 receptor (IP3R); following this, DAG activates the PKC, which ensures the entry of calcium into the cell through the membrane L-type calcium channel [84,85].

An important protein responsible for rapid non-genomic responses of calcitriol is disulfide-isomerase A3 (PDIA3), a form of vitamin D known to be associated with membranes; the protein mediates calcitriol-dependent membrane-signalling cascades and activates the functions of PLAA, PLA2, and PLC and the opening of Ca^2+^ and Pi (NaPi IIa,c) channels. Furthermore, in addition to its effects on the activation, release, and rapid accumulation of secondary messengers, vitamin D is also involved in the modulation of certain intracellular pathways. VDR appears to be colocalised with caveolin 1 (CAV1) and the Src non-receptor tyrosine kinase family (SRC) in caveolae [86,87]. Its membrane functionality has also been associated with the downregulation of cell cycle, proliferation, or immune responses through the SRC (SRC proto-oncogene, non-receptor tyrosine kinase) WNT pathway, sonic hedgehog signalling molecule (SHH), STAT1-3, NOTCH, MAPK/ERK, PARP-1, P53, and NF-ĸB pathways. In the endoplasmic reticulum, PDIA3 together with calreticulin (CRT) and calnexin (CANX) simplify protein folding [59,88,89,90,91,92].

Interestingly, treatment with vitamin D or its analogue, calcipotriol, elicits a PDIA3-dependent limitation of the inflammatory reaction and offers protection from potential detrimental factors, such as UVB or pathogen contagion [93,94]. Furthermore, WNT5a activates intracellular calcium release and stimulates PKC and CaMKII (calmodulin, CaM-dependent protein kinase II) via binding to its receptors Frizzled2 (FZD2) and Frizzled5 (FZD5) and receptor tyrosine kinase-like orphan receptor 2 (ROR2); this results in stimulation of mitogen-activated protein kinases (MAPK1 and MAPK2) [95,96,97]. Interestingly, it is also believed that in addition to vitamin D receptor activation, various alternative vitamin D metabolites or analogues, i.e., 20(OH)D3 and 20,23(OH)_2_D3, may also regulate other transcription factors, such as RORα and RORγ or AhR [98,99].

Non-genomic vitamin D3-dependent effects have also been noted in chondrocytes in the growth plate and keratinocytes in the skin, where vitamin D exerts its effects by activating the VDR analogue and MARRS (membrane-associated rapid response to steroid) receptors, also known as the endoplasmic reticulum stress protein 57 (ERp57/GRp58/ERp60) located on the cell membrane [47,100]. Moreover, calcidiol regulates osteoblasts and chondrocytes via its membrane-associated receptor, protein disulfide isomerase A3 (PDIA-3) in caveolae. After binding to PDIA-3, calcidiol activates phospholipase-A2 (PLA2)-activating protein (PLAA), stimulating cytosolic PLA2 and resulting in prostaglandin E2 (PGE2) release and PKCα activation, and thus differentiation. PDIA3 was also found in the nucleus. Hence, it is believed to act as a transcriptional regulator [95,96,97,101,102,103].

Alternatively, after binding to DBP 1α,25(OH)_2_D3, it may be transported by the megalin (LRP2)/cubilin (CUBN) or disabled-2 (DAB2) complex (LRP2-CUBN complex) via receptor-mediated endocytosis. Inside the cell, calcitriol binds the VDR-RXR complex, which is transported to the nucleus and affects the function of target genes. It has been confirmed that the megalin complex is associated with the fast and effective resorption of protein compounds, including DBP-binding vitamin D, and that this phenomenon is needed to support the optimum levels of vitamin D in the blood [104,105]. Importantly, it has also been suggested that megalin is engaged in the intracellular and mitochondrial transport of angiotensin II, stanniocalcin-1, and TGF-β. Moreover, megalin itself was found to be colocalised with mitochondria of cultured epithelial and mesenchymal cells, as well as many organs and tissues; in these cases, it was found to be in a complex with stanniocalcin-1 and the NAD-dependent deacetylase sirtuin-3, mitochondrial (SIRT3), which are participating in a defence against ROS [106].

Vitamin D also directly regulates mitochondrial activity, including fusion-fission, energy production, mitochondrial membrane potential, oxidative phosphorylation, and ROS scavenging/response. It thus protects the mitochondria from damage, ion channel activity, and ion flux, i.e., calcium influx, that may affect immune cell physiology as well as apoptosis. The membrane pathways activated by vitamin D can also directly affect mitochondrial proteins, including cytochrome P450 and potassium ion channels from the internal mitochondrial membrane [84,85].

Interestingly, recent studies have shown that vitamin D3 binds to VDR with an affinity of 0.1 nmol/L and to PDIA3 with an estimated Kd of 1 nmol/L. Such observations may justify the fact that higher doses of vitamin D supplementation and higher (≥30 ng/mL) serum 25(OH)D3 levels are needed for effective activation of non-genomic pathways [74]. However, these observations require further extensive laboratory and clinical investigations.

### 1.2. Vitamin D Supplementation

The optimal intake of vitamin D3 needed to ensure its beneficial effects on the skeletal and non-skeletal systems is still a subject of vigorous debate. Optimal serum vitamin D3 levels also vary according to, inter alia, life stage, race, ethnicity, and sex. Unfortunately, no consensus was noted in the categorization of normal serum vitamin D3 values, but some researchers agree that vitamin D3 levels above 30 ng/mL are sufficient for normal function and to maximise the effects on calcium, bone, and muscle metabolism [107,108]. However, an intake of 150 to 200 ng/mL has been indicated to be associated with intoxication and adverse effects [109,110,111]. Despite this, adverse effects related to vitamin D3 self-administration, such as hypercalcemia and hypercalciuria, are rare and usually result from taking extremely high doses of vitamin D3 for a prolonged period of time.

A study by the Central and Eastern European Expert Consensus Statement expert panel, using the Delphi method, recommends a vitamin D3 supplementation dose of 800 to 2000 international units (IU) per day for adults who want to have an adequate level of vitamin D3 (cholecalciferol) [112]. These doses have also been used to treat vitamin D deficiency; however, in cases with a clinical indication for rapid correction of vitamin D3 deficiency, increased dosages of vitamin D3 (e.g., 6000 IU daily) are indicated for the first four to twelve weeks of treatment, and treatment should be continued by supplementation with a maintenance dose of 800 to 2000 IU per day. The results of supplementation should be assessed after at least 6–12 weeks in risk populations (e.g., patients with malabsorption syndromes, obese individuals, patients on anticonvulsants, glucocorticoids, and antifungals) based on serum calcidiol concentration. Target concentrations of vitamin D3 of 30 to 50 ng/mL (75 to 125 nmol/L) were indicated [112].

The European Commission requested the EFSA Panel on Dietetic Products, Nutrition and Allergies (NDA) to determine dietary reference values (DRVs) for vitamin D3. As this was found to be not possible, the EFSA panel instead worked to determine an Adequate Intake (AI) value for achieving a serum 25(OH)D3 concentration of 50 nmol/L for the population as a whole. A subsequent meta-regression analysis indicated that this level can be achieved in the majority of the population with an AI of 15 μg/day [113]. Observational studies have since found that patients with conditions such as cancer may require around 4,000 IU/day [114,115,116]. After reviewing epidemiological evidence, case-control studies, and randomised control trials (RCTs), a multidisciplinary group formulated a set of recommendations for the prophylaxis and treatment of vitamin D3 deficiency both for the general population and for the risk groups of patients in Central European populations. Practical guidelines for cholecalciferol (vitamin D3) levels indicate that total serum calcidiol levels <20 ng/mL (<50 nmol/L) indicate deficiency, 20–30 ng/mL (50–75 nmol/L) are suboptimal, and 30–50 ng/mL are optimal (75–125 nmol/L) [117].

The Clinical Guidelines Subcommittee of the Endocrine Society in the United States recommends 600 to 2000 IU per day for people aged 19–70 years and 800 to 2000 IU/day after 70 years of age to prevent vitamin D3 deficiency. It is also recommended that groups at increased risk of deficiency require two- to three-times higher doses. Selected guidelines for supplementing vitamin D3 deficiency in adults recommend doses of 50,000 IU/week or 6000 IU/day for eight weeks. An oral dose of 1500–2000 IU/day achieved a 25(OH)D target concentration of 75 nmol/L (30 ng/mL) [116]. Verma et al. propose Recommended Dietary Allowances (RDAs) of 600 IU/day to achieve a serum 25(OH)D3 concentration of at least 20 ng/mL (50 nmol/L) in people aged 1–70 years and 800 IU/day in those aged 71 and older [118].

Similar recommendations are provided by the American Cancer Society Guidelines on Nutrition and Physical Activity for Cancer Prevention [119]. Additionally, the Institute of Medicine (IOM) recommends 200 IU/day (5 μg) of vitamin D3 (cholecalciferol) from birth through age 50, whereas people aged 51–70 years should take 400 IU (10 μg), and those over 70 years should take 600 IU (15 μg) [120]. Additionally, Haines et al. [121] indicate that in patients with vitamin D3 deficiency, a cumulative dose of at least 600,000 IU administered over several weeks may be necessary to replenish vitamin D3 stores, although large single doses of 300,000 to 500,000 IU should not be used. Importantly, despite the guidelines and recommendations, almost one billion people globally are at risk of vitamin deficiency.

#### Vitamin D Status as a Diagnostic Parameter

No common definition exists for adequate vitamin D3 status, measured as calcidiol serum concentration. Although results from some prospective clinical trials are promising, most have not been robustly designed and executed. However, the Endocrine Society defines deficiency as 25(OH)D below 20 ng/mL and insufficiency as 20–29 ng/mL. It is also well accepted that 20 ng/mL of 25(OH)D3 in the blood is an appropriate concentration for bone health. According to the latest data, a serum concentration of 75 nmol/L (30 ng/mL) should be regarded as the minimum, with 90 to 100 nmol/L (36–40 ng/mL) being optimal (assuming the conversion factor to IU is 1 ng/mL = 2.496 nmol/L). Numerous sources have confirmed that a level of vitamin D3 > 30 ng/mL is optimum for obtaining extraskeletal benefits, improving muscle efficiency, and decreasing concentration in vitamin-D-deficient older adults [122]. Furthermore, recently, it was also suggested that even higher serum levels of calcidiol, reaching 40–50 ng/mL, could be beneficial for the extraskeletal activities of vitamin D3, including its anticancer effects [123,124,125,126,127]. Numerous studies indicate that serum calcidiol deficiency (<30 ng/mL) leads to activation of PTH production and secretion (secondary hyperthyroidism), resulting in hypophosphatemia, bone loss, a risk of osteoporotic fractures, insufficient mineralization of bone collagen matrix, and osteomalacia in adults [128,129]. Hence, it has been proposed to increase the currently recommended vitamin D3 intake to 200 IU/day in younger adults and 600 IU/day in older adults to ensure bone health. In all cases, a daily intake of ≥1000 IU (25 μg) of cholecalciferol appears to be necessary to achieve a serum vitamin D3 concentration of 75 nmol/L in at least 50% of the population. However, other studies suggest that supplements are better targeted at frail elderly and dark-skinned people living in higher latitudes, in which case a daily dose of 400–800 IU (10–20 μg) is sufficient [130]. In addition, the significant change in recommendation relating to vitamin supplementation (800 IU vs. 4000 IU) and very high doses (50,000–100,000 IU) is recommended for individuals with severe vitamin D3 deficiency, although potential hypercalcemia and hypercaluria must be taken into account as potential side effects. Continuing on, indications regarding optimal vitamin D3 levels that prevent bone fractures occurred only in interventional studies; in these cases, sufficient supplementation was provided to achieve a calcidiol level of at least 40 ng/mL. In addition, studies have shown little or no benefit for serum vitamin D3 concentrations of 26 ng/mL (65 nmol/L) or less [131,132].

In many genes, SNPs have been identified that may be related to levels of circulating 25(OH)D3 [133]. For example, a genome-wide association study of calcidiol concentrations in 33,996 individuals of European descent from 15 cohorts by Wang et al. [133] found variants at three loci to have significant associations with D3 concentrations: 4p12 (rs2282679, in *GC*); 11q12 (rs12785878, near *DHCR7*); and 11p15 (rs10741657, near *CYP2R1*). All were confirmed in replication cohorts. Participants with a genotype with three confirmed variants in the highest quartile had a significantly higher risk of having a calcidiol concentration below 75 nmol/L (OR = 2.47, 95% CI, 2.20–2.78, *p* < 0.00001) or less than 50 nmol/L (OR = 1.92, 95% CI, 1.70–2.16, *p* < 0.0001) compared with those in the lowest quartile (assuming vitamin D3 insufficiency as <75 nmol/L or <50 nmol/L). Similarly, Fu et al. [134] also indicated that common genetic variants of the vitamin D3 binding protein (DBP) predict differences in serum calcidiol’s response to vitamin D3 supplementation. More precisely, the Gc2 (homozygous 436K) phenotype demonstrated the strongest affinity for serum concentrations of DBP, 25(OH)D3, and 1α,25(OH)_2_D3 in vivo. Mean 25(OH)D3 increases were 97% for TT, 151% for TK, and 307% for KK genotypes (*p* = 0.004) in adults receiving 600 or 4000 IU/d vitamin D3 for one year. The researchers confirm that the T436K genotype predicted changes in 25(OH)D3 after long-term vitamin D3 supplementation in the study population. Another study based on an animal model [135] examined whether inactivating mutations (SNPs) in *CYP2R1* can lead to a novel form of vitamin D3-deficiency rickets resulting from impaired 25-hydroxylation of vitamin D3. It was found that serum 25(OH)D3 levels were reduced by more than 50% in Cyp2r1^−/−^ knockout mice, i.e., with the SNP in the *CYP2R1* gene, compared with wild-type mice. Furthermore, in vitro studies with HEK293 cells found completely absent or markedly reduced calcidiol in cells with L99P and K242N mutations of CYP2R1. Interestingly, heterozygous individuals were more likely to display moderate biochemical and clinical features of vitamin D3 deficiency than homozygous individuals. In addition, after an oral bolus of 50,000 IU of vitamin D2 or vitamin D3, heterozygous individuals demonstrated a lower increase in serum 25(OH)D3 levels than controls, and homozygous individuals a minimal increase.

### 1.3. Vitamin D and the Immune System

Numerous in vitro studies indicate that vitamin D3 and its active metabolite calcitriol may modulate the immune response in immune and inflammatory cells such as dendritic cells (DC), macrophages, and activated T and B lymphocytes, resulting from the activity of nVDR; they also confirm that both CYP27B1 and 1α,25(OH)_2_D3 are able to control the activation, proliferation, differentiation, and function of these cells [38,136,137]. Interestingly, immune and inflammatory cells are able to convert 25(OH)D3 into calcitriol. In addition, calcitriol, VDR, and CYP27B1 are also expressed in epithelial cells, which also play roles in host defence through their innate adaptive immune response [38,137]. CYP27B1 enzyme activity is stimulated by immune stimuli, e.g., IFN-γ and lipopolysaccharide (LPS); these activate the transcription factor, CCAAT enhancer binding protein beta (C/EBPβ), which binds to the *CYP27B1* gene [138,139]. It is well known that the active form of vitamin D3, calcitriol, inhibits the adaptive immune response by constraining dendritic cell (DC) maturation, limiting T cell proliferation and the regulation of invariant natural killer T (iNKT) cells, and shifting the balance of T cell differentiation from the Th_1_ and Th_17_ pathways to the Th_2_ and T_reg_ pathways. In vitro studies have shown that this shift can also regulate the differentiation of CD4^+^ T cells [140,141].

In contrast, calcitriol activates the innate immune defence via the generation of the antimicrobial protein cathelicidin. In addition to directly binding to and killing various pathogens, cathelicidin acts as a secondary messenger, driving inflammation mediated by vitamin D3 during infection. Thus, the vitamin D-cathelicidin pathway regulates autophagy machinery, protective immune defences, and inflammation, and contributes to immune cooperation between innate and adaptive immunity [142,143]. Studies on B cells indicate that calcitriol treatment reduces proliferation, plasma cell maturation, and immunoglobulin production [136,144].

A number of genes associated with innate and adaptive immune reactions regulate the active form of vitamin D3; these include *ACVRL1* (activin A receptor such as type 1), *CAMP* (cathelicidin antimicrobial peptide), *CD14* (the plasma membrane-anchored glycoprotein CD14), *CD93*, *CEBPB* (CCAAT enhancer binding protein beta), FN1 (fibronectin 1), *MAPK13* (mitogen-activated protein kinase 13), *LILRB4* (leukocyte immunoglobulin such as receptor B4), *LRRC25* (leucine rich repeat containing 25), *SEMA6B* (semaphorin 6B), *SRGN* (serglycin), *THBD* (thrombomodulin), *THEMIS2* (thymocyte selection associated family member 2), and *TREM1* (triggering receptor expressed on myeloid cells 1). Most of these genes encode membrane proteins or proteins secreted during the immune response to infection and autoimmunity [145].

#### 1.3.1. The Adaptive Immune Response

The activities of activated T lymphocytes (CD4^+^ or CD8^+^) and T helper lymphocyte subpopulations, i.e., Th_1_, Th_2_, Th_17_, CD4^+^CD25^+^Foxp3^+^ T_reg_, are governed by antigen presentation by specialised antigen presenting cells (APC), and this is influenced by systemic factors such as vitamin D3. Although the response to vitamin D3 stimulation is often pro-inflammatory in the early stages, it tends to determine inflammation later on, for example by reducing Th_1_ cell count and increasing Th_2_ and T_reg_ cell numbers or inducing a shift from M1 to M2 macrophages [146,147]. More specifically, calcitriol negatively controls the synthesis of type 1 pro-inflammatory cytokines (IL-12, IFN-γ, IL-6, IL-8, tumour necrosis factor-α, IL-17, and IL-9) and increases the production of type 2 anti-inflammatory cytokines (IL-4, IL-5, and IL-10). This is enabled by inhibiting the production of IL-12, which is important for the development of the Th1 subpopulation and the production of IL-23 and IL-6, and is important for the formation and function of Th17. Furthermore, calcitriol also inhibits the production of Th_1_ cells capable of producing IFN-γ and IL-2 and Th_17_ cells capable of producing IL-17. Importantly, IFN-γ and IL-2 deficiency inhibit downstream antigen presentation, T cell recruitment, and T cell proliferation; IL-12 suppression activates Th_2_ cell formation, leading to increased production of IL-4, IL-5, and IL-13, which further inhibit Th_1_ development and shift the balance to the Th2 cell phenotype [148,149,150,151].

While calcitriol is known to suppress NF-κB activity, the underlying mechanism remains poorly understood. It is postulated that the nVDR interacts with IκBβ and IκBα or IKKα and IKKβ kinases, which inhibit NF-κB function. Calcitriol inhibits TNF-α-induced nuclear translocation of the p65 subunit, indicating that NF-κB(p65) activity is VDR-dependent. VDR overexpression inhibits NF-κB action induced by IKKβ [59,152]. In addition, calcitriol stimulates the production and activation of immuno-inhibitory regulatory CD4^+^CD25^+^Foxp3^+^ T cells and induces a stable tolerogenic phenotype in dendritic cells (DCs) [153,154]. It has been proven that the presence of calcitriol in the environment of DCs induces the production of T_reg_ immunosuppressive cells and their synthesis of IL-10 and TGF-β1, which inhibit the development of other T helper subclasses and induce immunotolerogenic T-regulatory responses. Studies also indicate that differentiation into tolerogenic DCs involves activation of the IL-6-JAK-STAT3 pathway and that JAK2-mediated phosphorylation of STAT3 is specific to vitamin D3 stimulation. The VDR and the phosphorylated form of STAT3 interact to form a complex with the methylcytosine dioxygenase TET2 [155,156].

Furthermore, calcitriol also has a direct and indirect effect on the regulation of the synthesis of many cytokines engaged in the adaptive immune reaction [148,149,150,151,157]. Interestingly, interleukin TNF-α has a VDRE section in its promoter that binds the VDR/RXR complex. Thus, calcitriol can inhibit the activation of NF-κB by increasing the expression of IκBα and preventing it from binding to response elements in genes coding for IL-8 and IL-12. It has also been shown that calcitriol or related D3 analogues promote the attachment of an inhibitory complex containing histone deacetylase 3 (HDAC3) to the rel-B promoter, one of the members of the NF-κB family, which suppresses ligand-dependent rel-B and inhibits gene expression [38,158]. Similarly, a negative VDRE was also found in the IFN-γ promoter, the activation of which leads to vitamin D3-dependent inhibition of IFN-γ activation. Additionally, the expression of granulocyte-macrophage colony-stimulating factor, GM-CSF, is controlled by VDR monomers, which, by binding to the repressive complex in the promoter of this gene, compete with *c-fos*/*c-jun*/*c-myc* and T cell nuclear factor 1 (NFAT1) for promoter binding [159].

#### 1.3.2. The Innate Immune Response

The innate immune response is associated with the function of toll-like receptors (TLRs) in polymorphonuclear cells (PMNs), macrophages, monocytes, and many epithelial cells. Thirteen TLRs (named TLR1 to TLR13) have been identified in humans and in other mammalian species. TLRs are a type of pattern recognition receptor (PRR) and recognise molecules that are broadly shared by pathogens but distinguishable from host molecules; these are collectively referred to as pathogen-associated molecular patterns (PAMPs). TLR activation is the first step in activating different pathways via adapter molecules such as myeloid differentiation factor-88 (MyD88), a member of the TIR family, and the TIR domain containing an IFN-β inducing adapter (TRIF). The MyD88-dependent response follows TLR dimerization by all TLRs except TLR3. The main effect of MyD88 signalling is the activation of NF-κB and mitogen-activated protein kinase. MyD88 then recruits IL-1R-associated kinases (IRAK1-3), which phosphorylate and activate the protein TRAF6, which in turn polyubiquinates the protein TAK1, which phosphorylates IKK-β. This cascade allows NF-κB to diffuse into the cell nucleus, resulting in transcriptional activation and the subsequent induction of inflammatory cytokines [160,161]. The TRIF-dependent pathway activates the kinases TBK1 and RIPK1. The TRIF/TBK1 signalling complex phosphorylates interferon regulatory factor-3 (IRF-3) allowing its translocation into the nucleus and production of type 1 interferons such as IFN-β while also activating NF-κB (late phase). Additionally, the adaptors TIRAP and MyD88 containing the TIR domain initiate activation of NF-κB (early phase) and MAPK. Both late- and early-phase activation of NF-κB is required for the production of inflammatory cytokines [162,163].

TLR activation induces antimicrobial peptides (AMPs) and ROS, which kill the pathogens. These AMPs include cathelicidin antimicrobial peptide/LL37 (CAMP/LL37), which plays a key role in the innate immune response. AMP expression is activated by calcitriol in both myeloid and epithelial cells. The transcription of human AMP and CAMP genes is stimulated by VDR bound to promoter-proximal vitamin D response elements (VDREs) [164,165,166]. Moreover, stimulation of TLR2 by infectious pathogens leads to upregulation of CYP27B1, which stimulates CAMP expression in the presence of an appropriate substrate (calcidiol). Interestingly, Th_2_ cytokines such as IL-4 and IL-13 are known to inhibit AMP induction. Therefore, since calcitriol evokes the differentiation of the Th_2_ phenotype, it may contribute to reducing the predisposition to microbial superinfections. Importantly, patients with vitamin D deficiency appear to be more vulnerable to such infections [167]. Vitamin D3 affects the innate immune system not only via up-regulation of the anti-microbial peptide CAMP but also by the plasma membrane-anchored glycoprotein CD14, which functions as a co-receptor for toll-like receptors [168].

The main activities exerted by the active form of vitamin D3, calcitriol, on immune and inflammatory cell populations are shown in Table 1.

### 1.4. Anti-Cancer Effects of Vitamin D

#### 1.4.1. Promotion of Apoptosis

In vitro and in vivo studies indicate that natural vitamin D3 or synthetic vitamin D3 compounds induce apoptosis of tumour cells in many types of human neoplasms. Calcitriol induces apoptosis through the mitochondrial pathway involving cytochrome c and Bcl-2 family proteins. The pro-apoptotic effect of calcitriol is mainly related to the activation of the intrinsic apoptotic pathway, especially the inhibition of the expression of anti-apoptotic genes, e.g., Bcl-2, Bcl-XL, and Mcl-1, and the stimulation of pro-apoptotic Bax, Bad, and Bak genes [169,170]. It thus favours the release of cytochrome c from mitochondria and the activation of pro- apoptotic caspases-3 and -9 [171,172,173]. Studies on the expression of genes associated with the phenomenon of apoptosis in oral squamous cell carcinoma cell lines (SCC15, SCC25, and CAL27) found that treatment with calcitriol at physiological concentrations (10–125 nmol/L) influenced programmed cell death by modulating the mRNA expression of key cell cycle and apoptotic signalling pathway regulators, such as p53, c-myc, ornithine decarboxylase (ODC), caspase-2, caspase-8, and Bax. The administration of calcitriol induced distinct dose-dependent, growth-inhibitory effects in all three oral cancer cell lines in vitro [174]. Furthermore, in KB cells from an oral floor squamous cell carcinoma, the mRNA of *survivin*, an inhibitor of apoptosis protein (IAP) family member, was clearly decreased by treatment with calcitriol. Survivin also inhibited apoptosis in the G2/M phase in HNSCC cells. In addition, KB cells also have VDR, and the proliferation of KB cells was suppressed by treatment with either the active form of vitamin D3 or its derivative substance, the 22-oxa-1,25-dihydroxyvitamin D3 (OCT) analogue. Survivin appeared to act by directly inhibiting caspase-3 and -7 [175]. Calcitriol has also been confirmed to induce apoptosis in cancer cells by down-regulating telomerase activity and decreasing the expression of telomerase reverse transcriptase (hTERT), the catalytic subunit of telomerase, resulting in shortened telomere length. Studies have shown that calcitriol downregulates *hTERT* mRNA as a result of the vitamin D response elements (VDREs) present in the 5′ regulatory region of the *hTERT* gene, repressing transcription and decreasing *hTERT* mRNA stability [176]. Importantly, telomerase activity and cancer cell apoptosis may be regulated by the induction of miR-498 by calcitriol, which leads to reduced expression of human telomerase reverse transcriptase mRNA [177].

Intriguingly, studies also indicate that the mutated p53 protein physically interacts with the VDR in cancer cells, converting the ligand into an anti-apoptotic factor; however, these phenomena are not fully understood [178]. In addition, in an in vitro model of carcinogenesis, it was observed that calcitriol may promote the sensitivity of tumour cells to TRAIL-induced apoptosis by inhibiting the production of interleukin (IL)-1β by tumour associated macrophages (TAMs) [179].

#### 1.4.2. Inhibition of Proliferation

Calcitriol is known to directly and indirectly exert an antiproliferative influence on tumour cells. These antiproliferative effects typically act via direct inhibition of cell cycle progression in the G0/G1 to S phase transition. The anti-proliferative effect is associated with inhibiting cyclin-dependent kinases (CDKs, i.e., CDK4, CDK6) and repressing the genes that encode cyclins D1 and C (*CCND1*, *CCNC*) by up-regulating vitamin D3 targets, CDK inhibitors p21^CIP1/WAF1//SDI1^ (i.e., cyclin-dependent kinase inhibitor 1 or CDK-interacting protein 1, CKI, encoded by *CDKN1A*), p27^KIP1^ (cyclin-dependent kinase inhibitor 1B, encoded by *CDKN1B*), and p19 (*CDKN2D*). This leads to dephosphorylation of the retinoblastoma protein (pRb) and the eventual arrest of the cell cycle in the G0/G1 phase. Induction of p21 and p27 mRNA regulates transcription by the VDR, thereby stabilising the E2F-pRb complex and inhibiting the E2F family of transcription factors, which transcribe the target genes that are essential to the cell cycle [180,181,182]. G1-phase arrest can also take place as a result of repression of the *myc* oncogene via the Rb-independent pathway. Calcitriol inhibits myc expression directly or indirectly by inhibiting gene transcription via antagonism of Wnt/β-catenin signalling. VDR binds β-catenin and blocks its proliferative effects in the intestinal epithelia; it also acts by activation of cystatin D or induction of the myc antagonist, i.e., the c-myc/MAD1/MXD1 network, to suppress *c-myc* function, as well as by repressing the long non-coding (*lnc*) RNA *CCAT2* (colon cancer associated transcript 2) and promoting the degradation of the myc protein in cancer [183,184,185,186,187]. The *myc* gene locus identified six possible binding sites, i.e., VDREs, for cells with the vitamin D receptor. As a result of CCAT2 inhibition, calcitriol decreased the binding of transcription factor TCF7L2 (TCF4) to the *myc* promoter, resulting in the repression of c-myc protein expression [187]. The E3 ligase and tumour suppressor FBW7 target drivers of cell-cycle progression, such as c-myc, for proteasomal degradation. In vitro studies confirmed a rapid enhancement of the interaction between FBW7 and VDR or c-myc, with simultaneous blockade of FBW7 binding to the c-myc antagonist MXD1. Calcitriol also regulated the function of other FBW7 target proteins such as cyclin E, c-jun, MCL1, and AIB1, while FBW7 depletion attenuated 1α,25(OH)_2_D3-induced cell cycle arrest [186]. Other studies have confirmed that calcitriol also inhibits the transcription of other proliferative genes, e.g., *c-fos*, *c-jun*, *c-junb,* and *c-jund* proto-oncogenes, *G0S2* (G0/G1 switch 2), and *CD44*, while enhancing the expression of *GADD45A* genes (growth arrest and DNA damage 45a), *MEG3* (maternally expressed gene 3, lncRNA), and *NAT2* (N-acetyltransferase 2). It was found that calcitriol and VDR stimulate *MEG3* and *NAT2* gene expression in tumour cells through direct binding to their promoters. Additionally, it was demonstrated that calcitriol inhibits the Wnt/β-catenin signalling pathway, which might lead to the downregulation of CD44 [187,188,189,190].

In addition, calcitriol regulates the expression of a variety of signalling pathway modulators that impact cellular proliferation, including cyclooxygenase-2 (COX-2) and 15-prostaglandin dehydrogenase (15-PGDH), as well as IGF binding protein-3 (IGFBP-3) [191,192,193]. Indeed, 1α,25(OH)_2_D3 has been shown to induce antiproliferative genes such as *CEBPA* (CCAAT-α enhancer binding protein) and *IGFBP3* (insulin-like growth factor binding protein 3) in tumour cells. IGFBP3 mediates the induction of p21^CIP1/WAF1^, supports miR-145 in the repression of the CDK2, CDK6, CCNA2, and E2F3 genes, and enhances the antiproliferative effect of calcitriol in cancer cells [194,195,196].

Furthermore, calcitriol also suppresses mitogenic signalling by inhibiting the activity of various growth factors, including insulin-like growth factor1 (IGF1), via upregulation of IGF-binding protein 3 (IGFBP3) and epidermal growth factor (EGF) [197,198,199]. In vitro studies on oral squamous cell carcinoma have found the calcitriol analogue eldecalcitol (ED-71) to inhibit the mitogenic effects of fibroblast growth factor (FGF1/2) by inhibiting NF-ĸB and inducing exosomal miR6887-5p, which down-regulates the mRNA 3′UTR of heparin-binding protein 17/FGF-1 (HBp17/FGFBP-1) via vitamin D3 receptor (VDR) in tumour cells [198,199]. Calcitriol down-regulates the action of the epidermal growth factor receptor (EGFR) and activates its ligand-induced internalisation in tumour cells [197]. Calcitriol has been proven to mediate the chemopreventive function of growth inhibitors and effectively enhance the inhibitory effects of erlotinib (an anti-EGFR mAb) against tumour proliferation in a patient-derived xenograft model of HNSCC [200].

In contrast, calitriol may lead to an increase in the expression of growth inhibitors, i.e., transforming growth factor β (TGF-β), a significant inhibitor of epithelial cell proliferation in non-cancerous cells, during the initial stages of tumourigenesis, e.g., epithelial-to-mesenchymal transition (EMT), migration, invasion, metastasis, and immunosuppression. It has been proven that the active form of vitamin D3 can activate TGF-β by inducing the expression of the type I TGF-β receptor, which sensitises cancer cells to the growth-inhibiting effects of TGF-β [201]. In addition, calcitriol and its analogues inhibit telomerase activity by reducing the amount of telomerase reverse transcriptase (hTERT) mRNA. Moreover, calcitriol stimulates the activity of various microRNAs, which reduces the expression of hTERT mRNA in some cancer cells [177,202]. Importantly, in pre-clinical studies in animal models, deletion of the VDR gene was associated with a disturbed balance between proliferation and apoptosis, increased oxidative DNA damage, and increased susceptibility to carcinogenesis. Since VDR expression is conserved in many human cancers, vitamin D3 status may be an important modulator of cancer progression.

#### 1.4.3. Induction of Differentiation

The suppression of tumour initiation and growth by calcitriol also results in increased differentiation. In vitro studies on a human solid tumour model have shown that calcitriol is a multi-level suppressor of WNT/β-catenin signalling and thus induces differentiation of cancer cells. Such antagonism acts by increasing interaction between β-catenin and the VDR, thus reducing the amount of β-catenin available for binding to TCF [203,204]. In recent years, a growing number of studies have revealed the other complex crosstalk between Wnt/β-catenin signalling and calcitriol in a variety of cancer types. It was confirmed that by binding to its high affinity receptor VDR, calcitriol induces the formation of β-catenin/VDR complexes and thus inhibits the activity of transcriptionally active β-catenin/TCF4 complexes. In addition, calcitriol increased the transcription of the *CDH1* gene encoding E-cadherin, which sequesters newly-synthesised β-catenin protein at the adherent junctions. Furthermore, calcitriol amplified the levels of the negative regulators of the Wnt/β-catenin pathway TCF7L2 (encoding TCF4), DKK1, and AXIN1; it also antagonised the pathway by reducing the production of IL-1β by TAMs, inhibiting GSK-3β activity in tumour cells, and increasing β-catenin expression. The Wnt/β-catenin pathway also antagonised 1α,25(OH)_2_D3/VDR signalling by the upregulation of miR-372 and miR-373, which reduces the level of VDR RNA and protein [205]. Moreover, active vitamin D3 up-regulates CDH1 expression (E-cadherin), leading to nuclear exportation of β-catenin and relocating it to the plasma membrane, where adherens junctions exist. In addition, calcitriol induces the transcription and activity of the extracellular inhibitor of Wnt signalling, the *DICKKOPF-1 (DKK)-1* or *DICKKOPF-4 (DKK)-4* gene; however, this depends on the presence of a transcription-competent nuclear VDR [206,207].

Importantly, calcitriol also induces increased expression of CDH1 (E-cadherin) and other epithelial markers involved in differentiation, such as ZO-1, and inhibits downstream Wnt/β-catenin signalling targets by blocking transcription of the proto-oncogene *c-myc* and *CCND1* genes (encoded cyclin D1); this prevents the differentiation of tumour cells, a feature that is also involved in angiogenesis, migration, and invasion. Numerous other genes and proteins have also been indicated as regulators of these pro-differentiation actions. The cell-specific mechanisms of action of calcitriol also include the regulation of JUN *N*-terminal kinase, β-catenin, NF-κB, and PI3K signalling pathways and of certain transcription factors such as CCAAT/enhancer binding protein (C/EBP) and activator protein complex 1, such as *PI3K*, *CEBPB*, and *CDKN1A* [208,209,210].

#### 1.4.4. Anti-Inflammatory Effects

Vitamin D3 is believed to limit inflammation by several mechanisms [42,191,194,211]. Numerous studies indicate that calcitriol inhibits the growth of human cancer cells. In vitro, it down-regulates CYP1B1 mediated by the COX-2/PGE2 pathway, inhibits p38 stress kinase signalling and the subsequent production of pro-inflammatory cytokines, inhibits NF-κB signalling, and increases the tissue inhibitor of metalloproteinase 1 (TIMP-1) and E-cadherin responses [42,191,194,211]. The down-regulation of the COX-2/PGE2 pathway lowers prostaglandin (PG) levels by repressing the mRNA and protein expression of endoperoxide synthase/COX-2, the key PG synthesis enzyme; it also inhibits prostaglandin signalling by enhancing the activity of the catabolic enzyme 15-hydroxyprostaglandin dehydrogenase, which initiates PG catabolism, and by down-regulating the expression of prostaglandin receptors. Activation of p38 and downstream production of prominent inflammatory cytokines, such as IL-6 and TNF-α, in malignant cells indicates decreased inflammation in tumour tissue [212,213,214,215,216].

The active form of vitamin D3 may also contribute to the inhibition of the p38 stress MAP kinase pathway, a key mechanism of cellular inflammatory signalling, by activating mitogen-activated protein kinase phosphatase 5 (MKP5) and consequently reducing the production of pro-inflammatory cytokines. This effect is related to the upregulation of MKP5 mRNA by calcitriol, which remains dependent on the VDR [217]. Furthermore, vitamin D3, paricalcitol (19-nor-1,25-(OH)_2_-vitamin D2), and other analogues may also noticeably repress CD44-STAT3 signalling, suggesting they may have the potential to inhibit cancer invasion. Indeed, in vitro studies have found paricalcitol to inhibit the expression of inflammatory mediators such as COX-2 while strongly down-regulating the levels of phosphorylated STAT3, a transcription factor inducing cell proliferation; this transcription factor is believed to act by up-regulating the expression of various cyclins (CKDs) and oncogenes and increasing cell survival by limiting the level of NF-κB in the nucleus via the promotion of anti-apoptotic gene expression [218]. The anti-inflammatory activity of calcitriol was associated with stimulating and stabilising the NF-κB α (IκBα) inhibitory protein, allowing physical interaction between the VDR and the IκB β kinase protein (IKKβ), and blocking NF-κB—DNA binding [219]. The inhibition of NF-κB activity led to down-regulation of key genes involved in oncogenic phenomena such as oxidative stress, EMT, proliferation, inflammation, pro-apoptotic mechanisms, angiogenesis, invasion, and metastasis [220,221].

#### 1.4.5. Inhibition of Angiogenesis

The anti-cancer effect of calcitriol is related to its inhibition of neoplastic angiogenesis via several mechanisms, as indicated in vitro in human/animal cancer cells and in vivo. In cancers of various origins, calcitriol exerts an anti-angiogenic effect via the down-regulation of major promoters of angiogenesis both at protein and transcriptional levels, such as hypoxia-inducible factor 1α (HIF-1α), a key transcription factor under hypoxia, vascular endothelial growth factor (VEGF-A), angiopoietin-1, and platelet-derived growth factor (PDGF). It also affects the induction of thrombospondin-1 (Tsp-1) and the growth of tumour-derived endothelial cells (TDECs), both of which are angiogenesis inhibitors. Calcitriol inhibits HIF-1α transcriptional activity as well as the HIF-1 target genes *VEGF*, *ET-1*, and *GLUT-1* in a VDR-dependent manner [222,223,224,225,226]. Direct transcriptional regulation of VEGF by calcitriol is connected with the presence of functional VDREs in its promoter region [227]. The formation of the angiogenic phenotype of neoplastic cells, induced by vitamin D3, also involves the regulation of *Id1* and *Id2* expression, these being transcriptional regulators of cell proliferation and differentiation and repressing the *DKK-4* gene, which encodes an extracellular Wnt pathway inhibitor that promotes angiogenesis and invasion in various tumours [207,228]. It has also been shown that treatment of tumour cells with calcitriol or its analogue, EB1089 (EB), reduces the level of the nuclear protein Fork headbox M1 (FOXM1), an oncogene regulating the cell cycle and carcinogenesis, and inhibits tumour proliferation and metastasis [224].

In addition, the active form of vitamin D3 also prevents angiogenesis by inhibiting the secretion of IL-8, a pro-angiogenesis factor, and interrupting its signalling by cancer cells by inhibiting the NF-κB factor. This inhibition results from the blockage of the nuclear translocation of p65, thus inhibiting the binding of the NF-κB complex to DNA [229]. Another mechanism is based on calcitriol reducing the expression of prostaglandin E2 (PGE2) generated by cyclooxygenase-2 (COX-2); this potentially inhibits angiogenesis by decreasing the synthesis of vascular endothelial growth factor (VEGF) and impairing the HIF-1α pathway in cancer cells [230]. Other studies have also shown that calcitriol can inhibit the growth of oral squamous cell carcinoma cells by blocking the HBp17/FGFBP-1 signalling cascades on the mRNA and protein levels, which are crucial in cancer angiogenesis. This event appears to take place via the NF-κB pathway since mRNA and protein levels of IκBα were found to be significantly up-regulated [231]. It has also been proven that calcitriol can also inhibit the function of tumour-derived endothelial cells by reducing their proliferation and sprouting in vitro and by down-regulating blood vessel density in human cancer models [226,232].

Interestingly, some studies indicate the opposite effect of calcitriol on angiogenesis. For example, calcitriol inhibited thrombospondin 1 (Tsp-1) in a breast cancer model, leading to an increase in VEGF expression [233].

#### 1.4.6. Inhibition of Epithelial-to-Mesenchymal Transition and Tumour Spread

It is well known that calcitriol inhibits the migratory and invasive phenotypes of human cancer cells by influencing the cytoskeleton and adhesive properties and the expression of proteases, protease inhibitors, and ECM proteins. This antineoplastic effect is linked to inhibition of EMT and the TGF-β and Wnt/β-catenin signalling pathways.

A loss of epithelial differentiation observed in cancers of various origins. This results from the EMT caused by changes in gene expression triggered by a group of transcription factors (EMT-TFs), e.g., mainly *SNAIL1*, *SNAIL2*, *ZEB1*, *ZEB2,* and *TWIST1*, in signals that induce or activate the EMT-TFs, such as TGF-β, Wnt, and Notch, and in ligands of several receptors with tyrosine kinase activity and cytokine receptors. EMT is associated with the acquisition of malignant features that play key roles in the process of carcinogenesis and tumour progression, such as cell migration, that support tumour-initiating and metastatic potential and increase resistance to cytotoxic chemotherapy, radiotherapy, and immunotherapy [234]. Numerous in vitro studies indicate that calcitriol may regulate the differentiation of various types of cancer cells through direct control of epithelial genes and/or repression of key EMT-TFs, thus increasing cell-cell and cell-ECM adhesion. On the other hand, certain key EMT inducers inhibit calcitriol activity by repressing the transcription of the *VDR* gene encoding the high-affinity vitamin D receptor (VDR) that mediates the effects of vitamin D3 [235,236]. By increasing the expression of CDH1 (E-cadherin), calcitriol thereby promotes intercellular adhesion, inhibits the rolling of tumour cells and their adhesion to microvascular endothelial cells, i.e., the extravasation stage preceding metastasis, and blocks the CXCL12/CXCR4 chemokine axis [237,238].

In cancer cells, the active derivative of vitamin D3 can activate the formation and accumulation of focal adhesive contacts, structures that bind to the ECM; it does this by up-regulating the expression of α5 integrin, β5 integrin, focal adhesion kinase (FAK), and paxillin in focal adhesion plaques while increasing FAK phosphorylation. Moreover, calcitriol inhibits the activity of the mesenchymal indicator N-cadherin and the myoepithelial molecules β-cadherin, α6 and β4 integrins, and α-smooth muscle actin, which determine the aggressive and lethal phenotype of cancer cells [239]. In addition, calcitriol also inhibits cancer cell mobility and invasiveness by inhibiting extracellular matrix (ECM) components, tenascin C and periostin, several integrins and metalloproteases, i.e., MMP-1, MMP-2, and MMP-9, and serine proteases such as plasminogen activator (uPA) function. In addition, it also increases the activity of protease inhibitors and pro-adhesion proteins of the actin cytoskeleton adapter [240,241]. In another in vitro study, calcitriol was observed to increase the activity of intercellular adhesion molecules that act as components of adherence and tight junctions, including CDH1 (E-cadherin), occludin, claudin-2 and -12, and ZO-1 and -2 in cancer cells [242,243]. Studies also indicate that calcitriol has a negative interference on the activity of proto-oncogenes and the Wnt/β-catenin pathway, which regulate the functions of the cytoskeleton. In addition, it has been shown that calcitriol can inhibit CDH2 and CDH3 (N- and P-cadherins) [239,244]. Furthermore, studies performed in tumour cell lines derived from various human cancers indicate that calcitriol may have an impact on the redistribution of cytokeratins (CKs) such as F-actin, plectin, filamin A, vinculin, and paxilin, which regulate the actin cytoskeleton and the intermediate filament network in the ECM binding structures [245]. In addition, calcitriol and the calcitriol analogue MART-10 may also inhibit the expression of SLUG, SNAIL1, SNAIL2, TWIST1, and ZEB1 molecules and increase the expression of mesenchymal markers fibronectin, LEF-1, and claudin-7 in tumour cells, which determines the increase in E-cadherin and p120-catenin. They may also reduce the expression of vimentin, thus inhibiting EMT and maintaining normal epithelial morphology [246,247,248].

The *SPRY2* gene, which encodes the SPROUTY-2 molecule, a modulator of tyrosine kinase receptor signalling, is also downregulated in in vitro models of human cancer cells. Inhibition of SPROUTY-2 eliminates EMT via inhibition of ZEB1, up-regulation of the epithelial splicing regulator ESRP1, and downregulation of transcription of the E-cadherin, claudin-7, and occludin genes. Moreover, SPRY2 represses LGL2/HUGL2, PATJ1/INADL, and ST14 proteins, the main regulators of the polarised epithelial phenotype, and ESRP1, an EMT inhibitor mediated by ETS1 and miR-200/miR-150. Furthermore, SPRY2 increased AKT activation by EGF, whereas AKT and Src inhibition reduced the induction of ZEB1 [249,250].

In recent decades, many studies have confirmed the role of vitamin D3 in inhibiting the cellular and molecular mechanisms of invasion and metastasis in human cancers [42,191,194]. The plasminogen activator system and matrix metalloproteinases (MMTs), which regulate the ECM, angiogenesis, and remodelling, are key promoters of metastasis and invasion of tumour cells, and their expression is regulated by calcitriol. Calcitriol inhibits gelatinase-B (MMP-9) and up-regulates tissue inhibitor of metalloproteinase1 (TIMP1) [251]. The glutathione peroxidase-1 (GPX1) gene is also related to tumour proliferation in humans. Calcitriol was found to downregulate the expression of GPX1 and thus inhibit cell proliferation, motility, chemoresistance, and urokinase plasminogen activator (uPA) secretion; however, it appears to stimulate apoptosis via the NF-κB pathway [252]. Moreover, 1α,25-(OH)_2_D3 enriches the pro-invasive function of tumour necrosis factor (TNF-α) by decreasing the expression of miR-221 and increasing that of the tissue inhibitor of metalloproteinase 3 (TIMP-3) in human neoplastic cells [253].

A few studies have also confirmed the hypothesis that calcitriol has direct anti-lymphangiogenic effects in vitro and is able to attenuate lymphangiogenesis in vivo through VDR-dependent mechanisms. Specifically, calcitriol has blocked lymphatic endothelial cell (LEC) tube formation, reduced LEC proliferation, and induced LEC apoptosis [254].

The potential anti-cancer mechanisms of calcitriol are illustrated in Figure 2.

## 2. Materials and Methods

A comprehensive search of relevant literature was performed regarding the role of vitamin D3 in pathogenesis of head and neck squamous cell cancer (HNSCC) and regulatory mechanisms of the initiation of carcinogenesis and invasion; in addition, the search covered the prophylaxis, i.e., chemopreventive effects, and potential therapy of HNC of various site origins. The searched corpus encompassed a wide range of molecular, observational, and interventional studies in humans. The final search (conducted on 31 March 2023) included articles published from January 2000 to March 2023, all of which are accessible via the PubMed/Medline/EMBASE/Cochrane Library databases. The following keywords were used as search criteria: “head and neck neoplasm or HNSCC”, “squamous cell carcinoma of head and neck”, “head and neck cancer”, “head and neck carcinoma”, “HPV-related tumours”, “vitamin D or vitamin D3”, “calcitriol”, “1α,25(OH)_2_D3 or 1α,25-dihydroxyvitamin D3”, “calcidiol”, “25(OH)D3 or of 25-hydroxy-vitamin D3”, “nutrient”, “nutrients”, “dietary”, “diet”, “supplementation”, “exogenous and endogenous vitamin D sources”, “synthetic vitamin D compounds”. There was no restriction on language or research group characteristics. No exclusion criteria were employed.

This article extensively discusses biological mechanisms of anti-cancer effect of vitamin D3 on the tumourigenesis, initiation, and progression of HNC and its implications on tumour cell differentiation as well as proliferation, angiogenesis, epithelial-to-mesenchymal transition (EMT), tumour metastasis, immune function, and immunotherapy in HNSCC. The review also focuses on the crucial results of molecular studies followed by latest pre- and clinical early trials, i.e., studies on in vitro models of head and neck carcinoma, as well as significant long-term observational and intervention population studies and key opinion-forming systematic reviews. This work discusses the issues in detail in order to increase clinical credibility.

## 3. Results

### 3.1. Animal and In Vitro Models of Head and Neck Squamous Cell Cancer

#### 3.1.1. Animal Models of HNSCC

Different animal models have been analysed to define the role of vitamin D3 in the process of HNSCC development. In mouse and hamster studies, vitamin D3 has been confirmed to regulate the development and growth of HNSCC; however, only a few studies can be indicated [118,255,256,257,258]. For instance, Meier et al. [255] investigated whether systemic intraperitoneal 1α,25(OH)_2_D3 injection prevents tumour formation in a hamster buccal pouch model of HNSCC carcinogenesis induced via a known carcinogen, 7,12-dimethylbenz(a)anthracene (DMBA). Only one hamster treated with calcitriol developed a confirmed neoplasm compared with all of the control animals (*p* < 0.01). Moreover, the mean time to the onset of lesion formation was significantly delayed in those animals treated with calcitriol. This effect suggests that systemic vitamin D3 therapy may delay carcinogenesis in an animal model. Another interesting in vivo study by Shintani et al. [256] examined the effect of implantation of A431 epidermoid carcinoma cells in nude BALB/c mice and the effect of eldecalcitol (ED-71, 1α,25-dihydroxy-2β-3-hydroxypropoxy vitamin D3), an analogue of calcitriol, as a potential anti-cancer agent for oral squamous cell carcinomas. The data indicate that tumour formation was reduced by the ED-71 treatment, with no significant difference in tumour CYP24A1 expression observed between the ED-71-treated and control groups. These results suggest that ED-71 is a potential anti-cancer agent for OSCC. In a recent study, the same group of researchers noted that ED-71 inhibited the growth of squamous cell carcinoma (SCC) cells in vitro and in vivo by down-regulating the expression of HBp17/FGFBP-1, a factor that enhances angiogenesis as well as promotes tumour growth, and FGF-2. Oral administration of ED-71 significantly inhibited the growth of A431-derived tumours in athymic nude mice, indicating that this compound may act directly on the cancer cells or on endothelial cells by modulating the tumour microenvironment [257].

There is also great interest in the chemopreventive and therapeutic potential of vitamin D3 and its derivatives in oral cancer (OSCC), one of the most frequently diagnosed cancers of the head and neck area [118,200,258]. For instance, the impact of dietary vitamin D3 supplementation on the initiation and progression of oral cancer has been investigated by Verma et al. [118] in preclinical trials on an animal model of oral carcinogenesis. C57BL/6 mice were maintained on one of three vitamin D3 diets: 25 IU, 100 IU, or 10,000 IU, and exposed to the carcinogen 4-nitroquinoline-1-oxide (4NQO). The animals on the 100 IU diet displayed a lesser occurrence of high-grade dysplasia, or OSCC, accompanied by a greater infiltration of CD3^+^ T cells compared with animals on the 25 IU and 10,000 IU diets. Interestingly, histological examination found the highest incidence of OSCC in animals maintained on a 10,000 IU diet. Furthermore, mice on 100 IU and 10,000 IU diets presented higher VDR and CYP24A1 enzyme immunostaining in high-degree dysplasia and OSCC compared with normal ones. However, it should be noted that serum 25(OH)D3 levels prior to carcinogen exposure were highest in mice on a 10,000 IU diet, but thereafter a ~50% decrease was observed in the following weeks. Collectively, an OSCC animal model analysing the chemopreventive properties of vitamin D3 displayed that short-term treatment of animals on a 25 IU diet with calcitriol significantly inhibited tumour growth compared with controls, but unfortunately had no significant effect on neoplastic lesions in animals on a reference diet of 1000 IU. Similar observations were introduced by Vincent-Chong et al. [258] in a preclinical prevention trial of calcitriol. The study examined the influence of the stage of intervention with the oral carcinogen 4NQO and the duration of treatment on the tumourigenesis of the oral cavity in an animal model. The findings confirm that calcitriol administered concurrently with 4NQO for 16 weeks significantly decreased the frequency of oral dysplastic premalignant lesions by 57% compared with 4NQO-only controls. Moreover, mice treated with calcitriol for longer periods exhibited higher CYP24A1, lower serum 1,25(OH)_2_D3 levels, and a higher incidence of an invasive OSCC phenotype. Interesting conclusions were also drawn by Paparella et al. [259] in a study of animal models of oral carcinogenesis. The authors confirm that oral-specific ablation of the zinc-finger transcription factor Krϋppel-like factor 4 (Klf4), which suppresses cell proliferation and promotes differentiation, may disrupt epithelial differentiation, thus increasing premalignant lesions and accelerating the neoplasia process upon chemical carcinogenesis. Additionally noteworthy, Bothwell et al. [200] analysed the effect of calcitriol on the chemopreventive efficacy of the EGFR inhibitor erlotinib against HNSCC in patient-derived xenografts (PDX) and the 4NQO carcinogen-induced HNSCC model. Longitudinal monitoring of cancer progression revealed the greatest reduction in the degree of dysplasia and tumour incidence and volume among animals treated with the combination regimen. Interestingly, phospho-EGFR and phospho-AKT pathway downregulation were noted in the tongues of PDX-bearing mice treated with 1,25(OH)_2_D3 and erlotinib. These data clearly highlight the potential of calcitriol to augment the efficacy of erlotinib against HNSCC.

In summary, research using knockout mice and transgenic mice provides ongoing information on the physiological role of vitamin D3 in classical target tissues and extraskeletal effects, including effects on cancer progression, cancerogenesis, and immunomodulation. Recent observations suggest that short-term treatment with calcitriol may yield therapeutic benefits that may be more pronounced in individuals with vitamin D3 deficiency, although further studies in larger cohorts are needed.

#### 3.1.2. In Vitro Models of HNSCC

As the active form of vitamin D3, calcitriol is a key regulator of cell differentiation and proliferation, and many in vitro studies have examined its activity as an agent with broad potential as an anticancer agent. For instance, 1,25-dihydroxyvitamin D3 appears to have differential effects on oral squamous cell carcinomas in vitro [260] in CAL27, SCC15, and SCC25 OSCC cell lines. The researchers found that vitamin D3 administration has growth-inhibiting effects, both at physiological and supraphysiological concentrations. Furthermore, calcitriol supplementation induced differential dose-dependent modulation of proliferative tumour phenotypes; this was associated with significant decreases in the viability of cancer cells and the activation of two key apoptotic pathways, i.e., caspase and Bcl/Bax. Recent experimental research has found that the anticancerogenic effects of vitamin D3 and its analogues are mediated through several mechanisms [34]. An interesting study by Lu et al. [261] investigated the effect of eldecalcitol (ED-71), an analogue of calcitriol, on the progression of a mouse xenograft tumour model of OSCC and the related anti-cancer mechanism. It was found that ED-71 is able to significantly inhibit the proliferation and migration of oral cancer cells (SCC-15 and CAL-27), block the cell cycle in the G0/G1 phase, and enhance apoptosis through glutathione peroxidase-1 (GPX1) downregulation. In another study, Shintani et al. [257] examined the potential anti-tumour effect of eldecalcitol for oral squamous cell carcinoma (OSCC, i.e., cell lines NA-HO-1-n-1 and UE-HO-1-u-1) and an epidermoid carcinoma/SCC (A431 cell line) in vitro. The growth assay in serum-free culture revealed that ED-71 inhibited the growth of the cancer cell lines in a dose-dependent manner. In addition, ED-71 and 1α,25(OH)_2_D_3_ suppressed HBp17/FGFBP-1 expression by inhibiting the NF-κB pathway. These results suggest that ED-71 possesses potential anti-tumour activity for SCCs. Another study by Shintani et al. [256] also confirmed that ED-71 was an active anti-cancer agent in OSCC cell lines (NA and UE); however, ED-71 was able to decrease the growth of cancer cell lines at concentrations a hundred times lower than calcitriol. As a result, it was noted that *CYP24A1* mRNA and protein expression in the OSCC cells also increased in a dose-dependent manner after ED-71 treatment. Furthermore, HBp17/FGFBP-1 was also found to be highly expressed in the tissue and cell lines of OSCC [231]. The results indicate that calcitriol may be used as a potential factor to restrict the activity of HBp17/FGFBP-1 in cancer cells (UE cells) treated with 40 nM 1α,25(OH)_2_D3. Consequently, it was found that *HBp17/FGFBP-1* mRNA and protein levels were significantly down-regulated following vitamin D3 treatment. This study also showed that HBp17/FGFBP-1 expression is affected by calcitriol via the NF-κB pathway (IκBα up-regulation), thus suggesting this molecule may have a potential role as a target for anti-tumour therapy by calcitriol. Another study aimed to explore the role of the glutathione peroxidase-1 (*GPX1*) gene associated with tumour progression in vitamin D3-mediated progression of salivary adenoid cystic carcinoma (SACC) [252]. It was found that decreasing GPX1 expression inhibited SACC (ACC-M, SACC-83, and ACC-2) proliferation, motility, and invasion, as well as chemoresistance and uPA secretion, but promoted apoptosis via the NF-κB(p65) pathway (IκBα activation). Moreover, it was supported that calcitriol worked as a modifier of NF-κB/GPX1/uPA expression, inhibiting cisplatin resistance and the cell invasion ability of the SACC cells [252]. As expected, it was found that the active metabolite of vitamin D3 inhibited tumour cell proliferation, cisplatin resistance, invasion, and migration and alleviated cancer progression in vivo. Furthermore, the expression of GPX1, P65, P-P65, and uPA in the experimental group was lower than that in the control and blank groups; it appears that these regulators were needed for calcitriol to perform its antineoplastic function. Taken together, establishing calcitriol as a modifier of NF-κB/GPX1/uPA expression provides a novel therapeutic strategy for the treatment of SACC [252]. Recent studies have also found that upregulation of GPX1 expression induces drug resistance in cancer cells by reducing ROS produced by platinum-based chemotherapy drugs [262]. Vitamin D3 was found to have a significant anti-tumour effect on head and neck squamous cell carcinoma by Satake et al. [175]. The findings indicate that calcitriol and its derivative 22-oxa-1,25(OH)_2_D3 suppress cell proliferation, induce apoptosis, and arrest the cell cycle; they also increase the sensitivity of chemotherapeutics and downregulate several angiogenesis factors and pro-apoptotic survivin in KB cell lines of oral floor squamous cell carcinoma.

In vitro studies have also found vitamin D3 to inhibit HNSCC cell proliferation, apoptosis induction, and cell cycle arrest, as well as angiogenesis, due to a higher sensitivity to chemotherapeutic agents. Based on recent data, the researchers propose that calcitriol and its derivatives may be a potential anti-cancer agent for head and neck squamous cell carcinomas. However, a high concentration of calcitriol would be required to achieve an effective therapeutic outcome, which could induce hypercalcemia, an important limiting factor for the use of calcitriol in cancer therapy. Fortunately, many potent analogues of vitamin D3 have been developed with reduced calcemic effects. One such analogue, EB1089, contains a side chain modified to render it less susceptible to catabolic degradation. It was confirmed that nanomolar concentrations of EB1089 arrested proliferating human SCC25 HNSCC cells in G0/G1 [263]. For instance, expression profiling in human SCC25 head and neck squamous carcinoma cells revealed EB1089 to have significant pleiotropic effects on cell proliferation, differentiation, and immune system regulation in an experimental investigation [264]. The identified targets included control of inter- and intra-cellular signalling, intracellular redox balance, cell adhesion, extracellular matrix composition, cell cycle progression, steroid metabolism, and more than 20 genes modulating immune system function. The results indicate that 10 μg/per day of EB1089 acted as a key chemoprevention agent, and its anti-cancer properties were related to antiproliferative, pro-differentiating, and gene-protective effects through the induction of GADD45α. Furthermore, the findings suggest that EB1089 treatment may reverse the malignant phenotype of SCC25 cells via “cadherin switching”, i.e., down-regulation of N-cadherin expression. Additionally, studies on tissue culture and a mouse model of HNSCC showed that both calcitriol and an EB1089 dose of 0.25 μg/kg/d reduced tumour growth by up to 80% in the absence of hypercalcemia [265]. Vitamin D3 and its derivative arrested the proliferation of cultured murine AT-84 SCC cells in G0/G1 cell-cycle phase through down-regulation of p21^WAF1/CIP1^ and up-regulation of p27^KIP1^ protein expression of cyclin-dependent kinase inhibitors (CDKI). Moreover, 1α,25(OH)_2_D3 also enhanced the expression of GADD45α, apparently by a p53-independent mechanism. In conclusion, EB1089 inhibited the growth of cancer cells at nanomolar concentrations and reduced tumour growth without calcemic effects. These data underline the need for further investigations of EB1089 as a chemopreventive and chemotherapeutic agent for HNSCC. In addition, Alagbala et al. [266] examined the antitumour effects of two less-calcemic vitamin D3 analogues, i.e., 19-nor-1α,25-dihydroxyvitamin D2 (paricalcitol) and 1α-hydroxymethyl-16-ene-24,24-difluoro-25-hydroxy-26,27-bis-homovitamin D3 (QW-1624F2-2), in human squamous cell carcinoma cell lines. Both vitamin D3 analogues inhibited cell cycle regulators such as p21^WAF1/CIP1^ and CDK2 and increased p27^KIP1^ protein expression, inducing G0/G1 cell cycle arrest and blockage of DNA synthesis. Furthermore, the vitamin D3 derivatives induced apoptosis, caspase-3 cleavage, and increased expression of pro-apoptotic MEKK-1 kinases. Phosphorylation of AKT, MEK, and ERK1/2, which promote cell growth and survival, was inhibited. Similar interesting observations were presented by Xiao et al. [267], who evaluated the potential effects of aspirin combined with vitamin D3 on cell proliferation and apoptosis in oral cancer CAL-27 and SCC-15 cells. The data demonstrated that vitamin D3 synergistically or additively potentiated the anti-proliferative and apoptosis-induced effects of aspirin against both OSCC cell lines in dose- and time-dependent manners. The anti-carcinogenic activity of ASA and vitamin D3 was related to the suppression of Bcl-2, p-ERK1/2/AKT, and Ras/MEK signalling pathways, which resulted in decreased rates of cell proliferation and higher rates of cell apoptosis.

Few studies have examined the anti-metastatic potential of vitamin D3 [268,269,270]. Chiang et al. [268] investigated the anti-cancer properties of a new less-calcemic 1α,25(OH)_2_D3 derivative, MART-10 (19-nor-2α-3-hydroxypropyl-1α,25-Dihydroxyvitamin D3), as a potent therapeutic agent. The results indicate that MART-10 has greater antiproliferative activity than calcitriol in inhibiting cancer cell growth in vitro and in vivo without inducing hypercalcemia. MART-10 was found to show greater potential than calcitriol in suppressing FaDu and SCC-25 squamous carcinoma cell lines; this occurred through increased cell cycle arrest at G0/G1, accompanied by a greater downregulation of Ki-67 expression and upregulation of p21^WAF1/CIP1^ and p27^KIP1^. Further observations also showed that MART-10 suppressed telomerase expression in the SCC-25 cell line to a greater extent than calcitriol. Recent research by Yang et al. [269] confirmed the antimetastatic effect of MART-10 on the FaDu HNSCC cell line. Both calcitriol and MART-10 effectively blocked the mobility and invasion of tumour cells, with MART-10 being much more potent than vitamin D3. The antimetastatic effect of both molecules was mediated by attenuating the epithelial-mesenchymal transition; indeed, both calcitriol and MART-10 inhibit EMT-promoting Snail and Twist, as well as intracellular MMP-9 expression and extracellular MMP activity. Furthermore, increased E-cadherin and decreased N-cadherin expression have also been observed in FaDu cells. Collectively, the above data suggests that MART-10 has potential as a chemopreventive and therapeutic agent to treat HNSCC. However, further investigation into the mechanisms through which vitamin D3 and its derivatives inhibit carcinogenesis is necessary, and it may lead to the development of effective chemopreventive agents to combat head and neck cancer in the future.

Publications on proteomic profiling have found vitamin D3 treatment to have a modulatory effect on protein expression and the pathways that play a significant role in HNC cancer progression, metastasis, chemoresistance, and cancer recurrence; their findings further support the potential role of vitamin D3 in targeted cancer prevention [15,271,272,273,274]. To date, numerous in vitro studies have investigated the potential molecular mechanisms of action of vitamin D3 on head and neck cancer cells, as vitamin D3 treatment has been found to significantly alter five microRNAs (miRs), each regulating between 41 and 2898 genes [271]. Molecular analyses indicate that vitamin D3 influences multiple pathways and processes associated with cancer. For instance, Ibrahimovic et al. [271] report that hsa-miR-7-1 and hsa-miR-335, i.e., two oncogenic molecules known to play roles in head and neck cancer or precursor lesions, were modulated by vitamin D3; they were significantly down-regulated in two human HNC cell lines of squamous cell origin, CAL-27 and SCC-25, and acted through multiple pathways relevant to HNCs, including MAPK, PI3K, RAS, and chemokine signalling. Furthermore, inhibition of hsa-miR-7-1 and hsa-miR-335 activity led to negative regulation of cell cycle and tumour proliferation and enhanced angiogenesis. In addition, in HSR cells, hsa-miR-7-1 and hsa-miR-335 were found to modulate Canonical Wnt signalling, important signal transduction pathways, and cell adhesion. Studies also indicate an increase in the activity of miRs, i.e., hsa-miR-331-5p, hsa-miR-616, and hsa-miR-632, in HNSCC model cells in vitro [271]. Importantly, these miRs can down-regulate the transcription of more than 157 target genes, many of which are well documented as regulators of cell proliferation, signal transduction, DNA damage response, adhesion, epithelial-mesenchymal transition (EMT), genome stability, and transcription of important genes, i.e., *PRMT1*, *MDM4*, *CDK4*, *XIAP*, *SMAD4*, *FGF2*, *FOXF2*, and *SOX5* [275,276,277]. For instance, Jackson-Weaver et al. [275] report that PRMT1 regulates the splicing of *MDM4*, a key controller of p53 stability. Loss of PRMT1 leads to the accumulation of p53, which enhances SLUG degradation and blocks EMT in in vitro studies. In addition, cyclin-dependent kinase 4 gene (*CDK4*) transcription significantly regulates tobacco-dependent oral carcinogenesis by acting as a potent regulator of cyclin-dependent kinase 4 (KRF) [278]. Additionally, another interesting study by Xuefang et al. [279] found miR-331-3p overexpression to markedly inhibit the proliferation and invasion of nasopharyngeal carcinoma cell lines (CNE-1 and 5-8F cells); it also promotes cell apoptosis via reduced expression of the transcription factor elF4B and thus the inhibition of phosphorylation of PI3K/AKT signalling molecules. Results for miR-632 in the context of HNC are inconclusive. For example, a genomewide study evaluating salivary miRs in the detection of oral cancer found under-expression of miR-632 in saliva from OSCC patients compared with healthy controls, suggesting increasing levels of vitamin D3 may bring benefits for the occurrence of OSCC [280]. Similarly, studies by Lu et al. [281] indicate that the CCL20/CCR6 chemokine pathway may increase p38MAPK activation, associated with changes in the miRNA profile in laryngeal cancer cells; the findings confirm that CCL20/CCR6 activity was positively associated with miR-20a-5p, miR-489, and negatively associated with miR-29-3p, miR-632, and miR-1276 in laryngeal cancer tissues, and suggested miR-632 may be a tumour suppressor in laryngeal cancers. Unfortunately, another study by Zhou et al. [282] observed that increased expression of miR-632 has oncogenic activity in laryngeal tissues and cell lines by promoting enhanced cell proliferation, migration, and invasion via negative regulation of GSK3β. Hence, recent studies suggest that vitamin D3 plays an important role in regulating the development and progression of HNSCC by affecting key pro- and anti-oncogenic molecular factors [15,271].

Other proteomic profiling studies have also confirmed that vitamin D3 induces significant changes in other intracellular proteins essential for HNC cancerogenesis. An example is nucleophosmin (NPM), an oncogenic factor that blocks mitochondrial localization of p53 and protects cells from apoptosis by reducing the mitochondrial level of p53 and by regulating the myc-ARF-p53 pathway and the molecule functions of histone H2A type 1-j (H2A1J). Recent studies have indicated that vitamin D3 treatment reduces the expression of NPM in SCC-25 cells, potentially promoting apoptosis avoidance and increased viability, growth, and proliferation in HNC cancer cells [271]. Similarly, lactoylglutathione lyase 1 (GLO-1) was down-regulated in oropharyngeal FaDu and Cal27 cell lines in vitro after vitamin D3 treatment. It has been documented that GLO-1 is significantly overexpressed in OSCC tissues. In addition, strong nuclear GLO-1 staining in vivo has also been significantly correlated with shorter progression-free and disease-specific survival and represents an independent risk factor for an unfavourable prognosis in oropharyngeal squamous cell carcinoma patients [271,283]. Vitamin D3 has also been indicated as a regulatory factor in the expression of heat shock protein β-1 (HSP27), a multifunctional protein with a well-documented role in HNC inflammation, proliferation, cancer progression, EMT, and more recently, in radiation hypersensitivity and therapeutic resistance [271]. For instance, a proteomic study by Zeng et al. [272] found that HSP27 was a potential mediator for chemoresistance in squamous cell carcinoma of the tongue (SCCT). Extracellular HSP27 was found to bind to TLR5 in vitro and in vivo and then activate NF-κB signalling to maintain SCCT cell survival. Moreover, intracellular HSP27 binds to BAX and BIM molecules, inhibiting their translocation to the mitochondrion and the subsequent release of cytochrome c after chemotherapy, resulting in the inhibition of mitochondrial apoptosis. Moreover, since HSP27 is also a downstream target of the PI3K/AKT signalling pathway, this factor may be a link between vitamin D-induced miR proteomic changes and pathways regulating cancerogenesis in the head and neck region. Additionally, Ibrahimovic et al. [271] propose that vitamin D3 activity in human OSCC in CAL-27 and SCC-15 cells may also be related to peroxiredoxin-1 (PRX-1) expression, an important antioxidant enzyme that promotes the cascade of tobacco-associated oral carcinogenesis. The study by Niu et al. [284] also confirms a down-regulation of PRX-1 protein level and the subsequent inhibition of tumour cell invasion and migration via regulation of EMT and NF-κB-linked activity during oral carcinogenesis, further supporting a role for vitamin D3 in suppressing HNC-associated pathways.

Furthermore, the antitumour function of the immune system in head and neck cancers also seems to depend on sufficient vitamin D3 activity. Numerous in vitro studies and studies with tumour-bearing animals from recent years clearly indicate that calcitriol and its analogues act as key modulators of the activity of various immune and inflammatory cell populations. The specific anti-cancer immune mechanisms in HNSCC are discussed in a later paragraph.

The effect of vitamin D3 in the selected animal and in vitro models of HNSCC is shown in Table 2.

#### 3.1.3. Limitations of Animal and In Vitro Studies

To date, several preclinical studies have been conducted on the role of vitamin D3 and its analogues in the successive stages of carcinogenesis and tumour progression in head and neck cancer. In the last twenty years, several key in vitro and in vivo studies, as well as those with tumour-bearing animals, have been performed. These have attempted to determine whether active metabolites of vitamin D3 modulate phenomena directly and indirectly related to cancer, i.e., apoptosis, differentiation, proliferation, invasion and metastasis, angiogenesis, and growth factor signalling in head and neck squamous cell carcinoma cells. Unfortunately, the ability of the active form of vitamin D3 and its derivatives to regulate the initiation or modulate the course of cancer, as indicated in epidemiological studies, differs from preclinical data acquired in experimental models. This may be due to a number of reasons. Firstly, the lack of methodological standardisation and the use of different laboratory methods are obstacles. Secondly, many preclinical studies do not include certain factors that have a significant effect on the metabolic and anticancer roles of vitamin D3, i.e., the length of outdoor exposure, skin pigmentation, degree of UVB radiation, and ethnicity, which influence cultural behaviour, clothing, diet, and lifestyle. Therefore, the preclinical researcher cannot conclusively confirm structural and functional differences in metabolic competencies and pathways/biomarkers related to vitamin D3-modulated stages of carcinogenesis with regard to clinical analyses and final clinical conclusions.

### 3.2. Studies on the Role of Vitamin D as Predictors of Cancer Risk, Progression, and Prognosis in HNC—The Chemopreventive Efficacy of Vitamin D on Precancerous Lesions

Experimental and epidemiological data suggest that vitamin D3 plays a role in the pathogenesis and progression of cancer, but more precise data on head and neck cancers of various origins are limited. The relationship between vitamin D3 and HNSCC has been analysed in only a few observational, prospective, and case-control studies. The results of a recently published systematic review were consistent with the observation that vitamin D3 deficiency is very common in patients with HNSCC; however, no credible data exists on the causal relationship between these two conditions [285]. Recent studies indicate quantitative associations between exposure to vitamin D3 and the incidence of HNC: cancer risk assessment included circulating calcidiol plasma/serum concentrations, vitamin D3 intake, the presence of the *VDR* gene polymorphism, and genes involved in the vitamin D3 metabolism pathway, *CYP27B1* and *CYP24B1* [194,286,287,288].

Intriguingly, vitamin D3 deficiency may be much more common in patients with HNSCC than in the healthy population; food insecurity, malnutrition, and cancer cachexia caused by low socio-economic status result in poor nutrition, impaired absorption of fat-soluble vitamins, and deficiency of dietary nutrients and supplements rich in essential vitamins. However, so far, there is no clear evidence of a causal relationship between vitamin D3 deficiency and the pathogenesis of head and neck cancer. Importantly, clinical analyses have revealed discrepancies in the reported data and final conclusions regarding the role of vitamin D3 in the cancer risk, progression, and prognosis of HNSCC in patients with diagnosed neoplastic disease.

#### 3.2.1. Limitation of Clinical Studies

Errors and potential bias may derive from heterogeneity of patient samples, insufficient sample sizes of comparative groups of patients and controls, short observation periods, or variable follow-up times; in addition, studies may fail to consider the seasonal variability of calcidiol concentrations associated with, inter alia, outdoor sun exposure, the alcohol consumption and smoking habits of HNSCC patients (tobacco smoke may influence vitamin D3 metabolism and function), or fail to relate the degree of vitamin D3 deficiency to the local/regional dissemination of the tumour. Many studies also used non-standardised food frequency questionnaires to assess dietary intake of vitamin D3 [107,108,289,290,291,292,293]. Furthermore, the studies are often carried out between different populations in heterogeneous ethnic groups with different degrees of exposure to solar ultraviolet B (UVB) radiation and subsequent vitamin D3 (the “sunshine” vitamin) deficiency status. It is well known that vitamin D3 absorption, which determines the concentration of circulating vitamin D3, depends on the length of exposure and skin pigmentation; in the case of the latter, D3 level decreases with darker pigmentation due to the protective effects of melanin and therefore should be taken into account in cross-sectional observational population studies. In addition, the data composition was also inadequate for conducting in-depth subgroup analysis or accurate stratified and sensitivity analyses [294,295,296,297]. Moreover, ethnicity also influences cultural behaviour, clothing, diet, and lifestyle and therefore has a significant impact on exposure to light radiation and carcinogens [298,299]. In addition, many studies use different reference points for the measured levels of vitamin D3, and therefore it is difficult to compare the primary and secondary endpoints. In addition, although the concentration of calcidiol in plasma is a useful indicator of the risk of clinical deficiency, its clinical use is limited by the lack of methodological standardisation. Additionally important is the fact that the samples had calcidiol measured in different laboratories at different times using different assay methods (albeit each considered valid), which limits the generalizability of the final findings.

#### 3.2.2. Vitamin D Plasma Concentrations, Vitamin D Intake, and VDR Gene Polymorphisms in HNC Risk—The Chemopreventive Efficacy of Vitamin D on Precancerous Lesions

The hypothesis that low vitamin D3 levels increase cancer risk has received strong experimental and practical support over the past two decades [286,287,288,300,301,302]. Recent research clearly indicates that hypovitaminosis D3 may increase the likelihood of progression to HNSCC in patients with premalignant lesions in the head and neck region; this may also be related to an increase in adverse effects associated with anticancer treatment, increased recurrence of neoplastic lesions, and a poorer prognosis for patients [108,303]. It has been proposed that vitamin D3 may fulfil a protective function by inhibiting angiogenesis, cellular proliferation, and inflammation and promoting apoptosis and cellular differentiation. Many pre- and clinical studies have shown that exposure of cancer cells to high doses of active vitamin D3 metabolites inhibits cell cycle progression, induces apoptosis, and inhibits tumour growth in vivo. Vitamin D3 has also been shown to potentiate the anticancer effects of several cytotoxic anticancer agents in preclinical in vivo models [304,305]. Numerous clinical studies indicate that vitamin D3 deficiency is a very common phenomenon in patients with cancers of various origins, even with vitamin D3 supplementation. Unfortunately, head and neck cancer has been rarely studied in relation to circulating vitamin D3, with only a few prospective studies published to date [288,301,306,307]. It is postulated that the greatest predisposition was related to a diagnosis of head and neck or squamous cell carcinoma, older age, male sex, and body mass loss [308,309,310,311]. The observations may be confirmed in a cross-sectional study of 500 consecutive Caucasian patients with a diagnosis of neoplastic disease by Kapała et al. [308]. Serum 25(OH)D3 < 30 ng/mL was assumed to indicate vitamin D3 deficiency and 25(OH)D3 > 100 ng/mL as potentially toxic, and optimal concentrations were in the range of 30–50 ng/mL. Thus, vitamin D3 deficiency was observed in 66.8% of the study population and in very low concentrations (≤20 ng/mL) in 38.6%. Vitamin D3 supplementation was declared by 27.2% of patients in the studied population, but deficiency was still observed in 31.6% of these. Importantly, a 2250 IU dose of vitamin D3 per day was found to be optimal for preventing deficiencies in Caucasian cancer patients. It is worth emphasising that the current guidelines for adults rarely recommend supplementation above 2000 IU/day; the Central European Guidelines indicate such levels only among obese or elderly adults, while the American Geriatrics Society (AGS) and Association of Directors of Geriatric Academic Programs (ADGAP) Education Committee recommend this intake in people over 70 years of age [312,313]. However, some researchers indicate that calcidiol deficiency is associated with a specific disease entity [314]. Assuming the minimum effective serum concentration of calcidiol in cancer patients to be 40–100 ng/mL, they require higher levels than in the general population [315]. Additionally, a study of over 40,000 individuals in the Health Professionals Study [302] found an increase in 25(OH)D3 level of 62.5 ng/mL to be associated with a >50% reduction in the risk of different neoplastic diseases, including head and neck and oesophagus cancers.

Most studies have shown that patients with head and neck cancer have significantly lower levels of circulating 25(OH)D3, the indicator of vitamin D3 status, than healthy individuals [107,108,288,289,290,316]. For example, Udeabor et al. [288] indicate that serum calcidiol may be used as a predictor for OSCC development. The study divided serum vitamin D3 levels into normal (>35 ng/mL), mild deficiency (25–35 ng/mL), moderate deficiency (12.5–25 ng/mL), and severe deficiency (<12.5 ng/mL). Among oral cancer patients, nearly 75% had moderate to severe vitamin D3 deficiencies, whereas only ~20% had a moderate deficiency in the control group, with no severe deficiency. The data also showed a positive relationship between vitamin D3 deficiency and OSCC risk, especially at levels below 25 ng/mL; this was found to increase the possibility of growing a malignant tumour by 1.65-fold. For instance, Anand et al. [108] confirmed that vitamin D3 scores were significantly lower in both oral premalignant lesions and oral cancer compared with healthy controls. They noted that supplementation with D3 or D2 resulted in increased serum 25(OH)D3 levels, with a peak in the values at day 14. The researchers also showed that individuals with OSCC who received vitamin D3 at a dose of 1000 IU/per day for three months had a significantly lower incidence of chemotherapy-related adverse events, such as oedema, erythema, ulcers, and pain. In the study cohort of patients, a decrease was noted in the incidence of oral mucositis (reduction of congestion, swelling, ulceration, and pain associated with chemotherapy), as well as an improvement in swallowing function and an increase in quality of life compared with patients who did not receive vitamin D3. It is possible that vitamin D3 may exert this protective action by stimulating differentiation and epithelization in oral mucosa cells. A study of serum vitamin D3 levels in oral precancerous lesions and OSCC by Grimm et al. [290] confirmed severe vitamin D3 deficiency in the OSCC patient cohort. An analysis of serum vitamin D3 levels in patients with oral precancerous lesions and oral squamous cell carcinoma revealed moderate or severe vitamin D3 deficiency in most of the OSCC patient cohort. The study also assessed the expression of VDR in oral precursor lesions such as simple hyperplasia, squamous intraepithelial neoplasia, SIN I-III, and OSCC specimens, with normal oral mucosa as a control; the findings confirm a significant increase of up to 50% of VDR expression in precancerous and OSCC tissue compared with normal tissue. Furthermore, compared with SIN I-III lesions, VDR expression also significantly decreased in OSCC; however, there was no significant correlation between serum 25(OH)D3 and corresponding immunohistochemically detected VDR expression in oral cancer. The same conclusion was confirmed by Mostafa et al. [316] in a study assessing the value of vitamin D3 assessment in patients with HNSCC before treatment. This study included individuals with various HNSCC sites and sex-matched and age-matched healthy volunteers as controls. The results indicate the median vitamin D3 serum level to be 40.35 nmol/mL (31.9–55) in the patient cohort and 118.75 nmol/mL (55.0–175) (*p* < 0.001) in the control group, confirming significantly lower 25(OH)D3 levels in HNSCC patients. The prevalence of vitamin D3 deficiency (<37.5 nmol/mL) in patients was 42%, i.e., significantly higher than in the control group (3%); this confirms that vitamin D3 deficiency is prominent in HNSCC patients before treatment compared with controls. Similarly, Bochen et al. [107] confirmed vitamin D3 deficiency in head and neck cancer patients. The median calcidiol serum concentration was significantly lower in the HNSCC group (11.1 ng/mL) than in the healthy control group (21.8 ng/mL; *p* < 0.0001). Moreover, only a small percentage of HNSCC patients demonstrated a sufficient vitamin D3 serum level before treatment (30–100 ng/mL). Interestingly, the authors also indicated that normalisation of vitamin D3 levels in cancer patients increased the cytotoxic activity of natural killer cells, favouring an antitumour immune response. The high prevalence of vitamin D3 insufficiency in patients with head and neck cancer at diagnosis was also supported in a prospective cohort study by Orell-Kotikangas et al. [289]. The concentrations of plasma calcidiol measured before cancer treatment confirmed D3 hypovitaminosis (37.5–50 nmol/L) in 20% of the patients and vitamin D3 deficiency (<37.5 nmol/L) in 45% of the patients, with no seasonal variation observed. Moreover, circulating calcidiol was less than 50 nmol/L in 65% of HNC patients. It was also hypothesised that combining vitamin D3 with chemotherapy increased the effectiveness of chemotherapy in oral cancer [317]. Dalirsani et al. [317] examined the effects of the combination of 5-fluorouracil, 13-cis retinoic acid, and vitamin D3 separately and in combination in oral squamous cell carcinoma (OSCC) culture. It was found that the combination of a chemotherapeutic with both retinoic acid derivatives and vitamin D3 had a greater inhibitory effect on cell proliferation and apoptosis than each drug alone. Moreover, other studies have also confirmed that local or systemic administration of vitamin D3 and its derivatives may lead to a greater sensitization of OSCC to apoptosis associated with radio- and chemotherapy treatment, supporting the idea that natural vitamin D3 or synthetic vitamin D3 compounds could be useful for chemoprevention [290]. In summary, all these studies support the claim that vitamin D3 deficiency can be a useful indicator of OSCC. Therefore, routine prescription of vitamin D3 supplements to people with moderate or severe deficiencies may reduce the chances of developing head and neck cancers with a higher risk of chemoresistance. Regarding the risk of cancer, studies suggest an inverse relationship between calcidiol levels and HNSCC risk [300,301]. The European Prospective Investigation into Cancer and Nutrition (EPIC) study, comprising nearly 400,000 head and neck cancer participants, compared circulating 25(OH)D3 levels in pre-diagnostic samples with the risk of HNC and oesophageal cancer and post-diagnosis all-cause mortality. Cases where vitamin D3 was confirmed to have an inverse association with HNC risk were usually associated with smoking. For instance, Fanidi et al. [300] observed this association only in smokers or ex-smokers; the researchers confirm that after adjusting for risk factors, doubling the circulating concentrations of calcidiol was associated with a 30% lower probability of HNSCC carcinogenicity (OR = 0.70, 95% CI, 0.56–0.88, p*_trend_* = 0.001). Further anatomical studies have shown an inverse relationship between 25(OH)D3 level and the combined risk of laryngohypopharyngeal cancer (OR = 0.55, 95% CI, 0.39–0.78) and oral cancer (OR = 0.60, 95% CI, 0.42–0.87). Moreover, low levels of calcidiol were significantly correlated with a higher risk of all-cause mortality in the study cohort. In a prospective population-based Danish study of tobacco-related cancer from the Copenhagen City Heart Study (CCHS), lower plasma calcidiol levels, divided by clinical categories or seasonally adjusted percentile categories, were found to be related to an increase in the cumulative incidence of neoplastic disease. The CCHS study reported a clear inverse association between 25(OH)D3 and smoking-related cancers and circulating vitamin D (OR_1/2_ = 1.20), with a particularly strong association for HNC. Afzal et al. [301] found 25(OH)D3 levels to be related to only smoking-related cancers, such as lung, head and neck, bladder, kidney, liver, and oesophageal cancers. In particular, the multivariate adjusted risk ratios for smoking-related cancers were 1.75 (95% CI, 1.33–2.30), including head and neck cancer (OR = 1.28, 95% CI, 1.06–1.54) for 25(OH)D3 < 5 vs. ≥20 ng/mL, while no association was noted for non-smoking-related cancers overall (OR_1/2_ = 0.95). Additionally, among 122 head and neck cancer patients, multivariable adjusted hazard ratios for a 50% reduction in calcidiol concentration were 1.44 (95% CI, 1.19–1.73) [301]. Taken together, these findings indicate that the protective effect of high vitamin D3 might only be associated with HNC cases related to smoking. This, however, is an observation that requires further detailed analysis in a larger cohort of patients. Of great interest is a meta-analysis performed by Pu et al. [286] on a large cohort of 5272 HNSCC patients in which calcidiol concentrations were assessed by radioimmunoassay, automated immunoassay, and chromatographic methods. Consequently, the researchers found significant inverse associations between the incidence of HNSCC and circulating calcidiol levels. After adjusting for potentially confounding risk factors, in the study population, the combined OR of the highest 25(OH)D3 levels with the lowest levels was 0.68 (95% CI, 0.59–0.78). Similarly, a prospective study of predictors of vitamin D3 status and cancer in multivariable models by Giovannucci et al. [302] confirmed that low levels of circulatory vitamin D3 may be associated with increased cancer incidence and mortality in men, particularly for digestive-system cancers: an increment of 25 nmol/L in predicted calcidiol level was associated with a 17% reduction in total cancer incidence (multivariable relative risk: RR = 0.83, 95% CI, 0.74–0.92). Moreover, the authors indicated that the level of vitamin D3 supplementation necessary to achieve a 25(OH)D3 increment of 25 nmol/L may be at least 1500 IU/day. Additionally, a meta-analysis of observational studies by Muñoz et al. [194] indicated that serum calcidiol levels were inversely correlated with the incidence of 12 types of human cancer. The researchers observed that an increase in the blood concentration of 25(OH)D3 from 10 to 80 ng/mL would decrease cancer incidence rates by 70 ± 10%. Unfortunately, meta-analyses of cancer incidence with respect to dietary intake have had limited success due to the low amount of vitamin D in most diets.

In contrast, other prospective cohort studies conducted in the HNSCC cohort of patients did not indicate any association between calcidiol and the risk of HNSCC [306,307,318]. Skaaby et al. [307] conducted a prospective population-based study of the relationship between serum 25-hydroxyvitamin D3 levels and the incidence of certain types of cancer based on data collected from the Danish Cancer Registry. The DCR study did not observe any clear relationship with circulating vitamin D3. The authors confirmed no statistically significant associations between calcidiol status and total or specific cancers, including head and neck carcinoma (HR = 0.97; 95% CI, 0.84–1.12). A case-control study evaluated the association between serum 25(OH)D3 and the risk of developing squamous cell carcinoma of the head and neck as part of the prospective Alpha-Tocopherol Beta Carotene (ATBC) study [306] among 9791 male Finnish smokers. The ATBC study confirmed no association between 25(OH)D3 and HNC risk, although the latter was underpowered to detect any such notable association. In results comparing serum 25(OH)D3 levels below 25 nmol/L with reference values of 50 to <75 nmol/L, the OR was 1.35 (95% CI, 0.53–3.43, p*_trend_* = 0.65) for whole head and neck cancer. It was found that p*_trend_* values were 0.93, 0.78, and 0.26 for cancer of the oral cavity, pharynx, and larynx, respectively. The season of blood collection had no effect. Therefore, the research showed no association between serum calcidiol and the risk of head and neck cancer. Similar results were reported by Budhathoki et al. [318] in a large case-cohort study within the Japan Public Health Center-based Prospective Study cohort. The study analysed plasma calcidiol concentrations and the subsequent risk of total and site-specific cancers. The results of the research confirmed that high levels of circulating 25(OH)D3 had a protective effect against several human cancers but did not influence HNC.

Studies have noted inverse relationships between vitamin D3 intake from dietary and supplemental sources, or none at all. For example, a meta-analysis of sixteen studies with a total of 81,908 participants examined the effect of circulating calcidiol and vitamin D3 intake on HNSCC results [286]. The cumulative OR was 0.77 (95% CI, 0.65–0.92) when comparing the categories of highest and lowest vitamin D3 intake. Furthermore, higher levels of circulating calcidiol significantly decreased the incidence of HNC by 32% (OR = 0.68, 95% CI, 0.59–0.78). Additionally, two case-control studies in Italy by Lipworth et al. [319] identified inverse associations between dietary 25(OH)D3 intake and risk of squamous cell carcinoma of the oesophagus (SCCE) and oral/pharyngeal cancer (OPSCC), which were most pronounced among heavy current smokers and heavy consumers of alcohol. More specifically, using a reference group of those in the highest tertile of vitamin D3 who were never/former smokers, ORs among heavy smokers in the lowest vitamin D3 tertile were 8.7 (95% CI, 4.1–18.7) for SCCE and 10.4 (95% CI, 6.9–15.5) for oral/pharyngeal cancer. Similarly, compared with those in the highest tertile of vitamin D3 who drank <3 alcoholic drinks/day, corresponding ORs were 41.9 (95% CI, 13.7–128.6) for SCCE and 8.5 (95% CI, 5.7–12.5) for oral/pharyngeal cancer among heavy alcohol drinkers in the lowest vitamin D tertile.

Other authors, however, do not confirm these observations. For instance, Negri et al. [320] investigated the relationship between selected micronutrients, including vitamin D3, and oral/pharyngeal cancer risk in a case-control study in Italy and Switzerland. No association was found between vitamin D3 intake and cancer risk (OR = 0.83, 95% CI, 0.67–1.04, p*_trend_* = 0.066). There was also a general consistency across strata of sex and age (<60 and ≥60 years). Similar results concerning the relationship between the risk of HNC and vitamin D3 intake were also presented by Peters et al. [321], who confirmed no relationship between estimated intake of vitamin D3 and increased HNSCC risk (*p* = 0.09 in the log-reg model adjusted for age, population type, sex, BMI, education, tobacco, alcohol, and non-alcohol caloric intake).

Since the single nucleotide polymorphism (SNP) of the *VDR* gene has been shown to affect vitamin D3 function, several studies have also investigated the role of such genetic variation in both HNSCC risk, occurrence, and prognosis [286,322,323]. The studies included an assessment of the risk of HNSCC in relation to the polymorphism *FokI* (rs2228570), located at the transcription initiation site of the *VDR* gene; this leads to the formation of a protein isoform that plays an important role in post-transcriptional modification. However, SNPs *BsmI* (rs1544410) and *TaqI* (rs731236), located in the vicinity of the 3’ end of the *VDR* gene, do not change the amino acid sequence of the encoded protein but determine the level of the protein by regulating the stability of the *VDR* mRNA [324,325]. Importantly, the relationships between specific VDR polymorphisms and cancer types remain ambiguous [326]. A meta-analysis by Pu et al. [286], comprising 81,908 participants from Europe, North America, and Asia-Pacific, found a relationship between the *FokI* polymorphism and HNC risk. The findings indicate a significantly reduced incidence of HNC associated with this mutation based on two genetic models (*ff* vs. *Ff* + *FF*: OR = 0.77, 95% CI, 0.61–0.97, and *ff* vs. *FF*: OR = 0.75, 95% CI, 0.58–0.97, respectively). Furthermore, subsequent analyses found Caucasians to have a reduced risk of HNC for the recessive model (*ff* vs. *Ff* + *FF*: OR = 0.72, 95% CI, 0.55–0.94), and the *TaqI* polymorphism was associated with greater HNC risk. A significant adverse relationship in HNC risk was observed in the general population (*tt* vs. *Tt* + *TT*: OR = 0.70, 95% CI, 0.55–0.90, and *tt* vs. *TT*: OR = 0.72, 95% CI, 0.55–0.95), as well as among Caucasian populations (*tt* vs. *Tt* + *TT*: OR = 0.73, 95% CI, 0.56–0.95, and *tt* vs. *TT*: OR = 0.74, 95% CI, 0.56–0.98). A number of studies have examined whether the *BsmI* polymorphism is not associated with HNC risk. Huang et al. [322] did not identify any significant differences between nasopharyngeal carcinoma patients (NPC) and controls in a Chinese population with regard to the genotype and allele frequencies of VDR *FokI* and *BsmI* polymorphisms (for VDR *FokI*: OR = 1.03, 95% CI, 0.76–1.41; for VDR *BsmI*: OR = 0.80, 95% CI, 0.48–1.33). However, it is important to note that the final results of these studies can be influenced by small sample sizes and the limited number of studies examined. Any discrepancies may also be explained by the distinct genetic backgrounds of different cancer types and the functional mechanisms of vitamin D3 in different tissues. A case-control study of 719 HNSCC cases also examined polymorphisms at the *TaqI* and *FokI* restriction sites in genomic DNA [323]. The findings confirm that both homozygous variant genotypes (*ff* and *tt*) were related to a lower incidence of HNSCC (OR =0.72, 95% CI, 0.53–0.98 and OR = 0.64, 95% CI, 0.47–0.87, respectively) compared with the common *FF* and *TT* genotypes. Interestingly, the *VDR* variant genotypes were associated with a decreasing risk of neoplastic disease depending on the presence of the variant allele, and their presence was associated with a significant decreasing trend in OR, particularly for the combined genotypes (p_trend_ < 0.001). These data support the hypothesis that the *VDR f* and *t* alleles and their genotypes may protect against HNSCC. A study of *TaqI* SNPs in the 30 region of *VDR* by Bektaş et al. [327] showed that the *VDR Tt* genotype in patients with OSCC is associated with a significantly higher risk of cancerogenesis than those with other genotypes (*p* = 0.036). In particular, female OSCC patients were more likely to develop oral cancer (*p* < 0.001). The obtained results of the analysis indicated that the *VDR TaqI* polymorphism may be associated with susceptibility to OSCC and may serve as a way to prevent OSCC in the future. These findings also suggest that the *VDR Tt* genotype may be a risk factor for OSCC and that the *tt* genotype has a protective role against OSCC, especially in women. Similarly, Małodobra-Mazur et al. [328] report a stronger genetic correlation between the risk and incidence of OSCC and rs2238135 in the *VDR* gene (*p* = 0.0007). The *G/C* genotype of rs2238135 in the *VDR* gene was characterised by a 3.16-fold increased risk of OSCC in the study population.

A single study indicates that genes involved in the vitamin D metabolism pathway, *CYP27B1* and *CYP24B1*, may also affect individual susceptibility to head and neck carcinoma. For example, a PCR–RFLP study by Zeljic et al. [329] indicates that polymorphisms in the vitamin D3 receptor genes *CYP27B1* and *CYP24A1* are significantly associated with the incidence of oral cancer. The researchers note that individuals with heterozygote genotype AG of the *CYP24A1* gene (rs2296241) (OR = 0.28, 95% CI, 0.14–0.57, *p* = 0.000) demonstrate lower OSCC risk compared with wild-type. In addition, the *VDR* polymorphisms *TaqI* and *BsmI* were not found to be associated with oral cancer risk.

#### 3.2.3. Vitamin D Plasma Concentrations, Vitamin D Intake, and VDR Gene Polymorphisms as Predictors of HNC Mortality, Survival, and Recurrence

Vitamin D3 deficiency is frequently observed in human cancer patients, and this can be prognostic for head and neck carcinoma. A prospective study of predictors of vitamin D3 status and cancer mortality in men by Giovannucci et al. [302] found a higher D3 level to be associated with a 29% reduction in total cancer mortality (RR = 0.71, 95% CI, 0.60–0.83) and even a 45% reduction in digestive-system cancer mortality (RR = 0.55, 95% CI, 0.41–0.74). Furthermore, a significant inverse association was observed for oral and pharyngeal cancers (RR = 0.30, 95% CI, 0.11–0.81). Several recent analyses have also pointed out that, from a practical point of view, HNSCC patients should receive vitamin D3 supplementation to reduce the risk of death [300]. For example, the European Prospective Investigation into Cancer and Nutrition (EPIC) study compared circulating calcidiol with post-diagnosis all-cause mortality. Low calcidiol concentrations were found to be associated with a higher risk of death from any cause among HNSCC cases. The chance of death was 1.72 times higher (95% CI, 1.11–2.51) for participants with circulating levels of 25 nmol/L compared with those with 50 nmol/L. Interestingly, no further survival benefits were seen for cases with calcidiol concentrations above 50 nmol/L. The expected 5-year post-diagnostic survival probabilities were 0.58 for cases with 25 nmol/L 25(OH)D3 (95% CI, 0.49–0.65) and 0.71 (95% CI, 0.64–0.77) for 50 nmol/L. However, no relationship was seen between survival and oesophageal cancer in the EPIC cohort. A study performed on 231 HNSCC patients and 232 healthy controls by Bochen et al. [107] estimated the prognostic value of vitamin D3 deficiency. HNSCC patients with low vitamin D3 demonstrated significantly shorter overall survival (OS) than those with high vitamin D3 (p_log-rank_ = 0.0085). Moreover, HPV^(+)^ tumour status was a significant predictor of better OS in HNSCC patients (p_log-rank_ = 0.0188); however, no such prognostic relationship was found between vitamin D3 supply and OS in HPV^+ve^ patients. In contrast, low vitamin D3 serum levels correlated with significantly shorter OS in HPV-negative patients (p_log-rank_ = 0.0187). Similarly, Gugatschka et al. [293] found HNSCC patients to demonstrate significantly decreased serum levels of calcidiol on diagnosis. When compared with a cohort from an epidemiological study, the authors found disease-free survival (DFS) as well as overall survival (OS) times to be associated with 25(OH)D3 levels. In addition, significantly shorter OS was found in patients with a vitamin D3 serum level below 10 ng/mL; also, higher 25(OH)D3 concentrations prior to treatment were related to longer post-treatment with DFS (RR = 0.85, 95% CI, 0.75–0.96, *p* = 0.01) and OS (RR = 0.89, 95% CI, 0.83–0.97, *p* = 0.006) in patients with newly diagnosed squamous cell carcinoma of the upper aerodigestive tract. The results of this epidemiological study are in line with those of others. For example, a meta-analysis by Pu et al. [286] was an interesting population-based study that estimated the association between vitamin D3 exposure and secondary outcome, i.e., mortality, on the basis of sixteen studies with a total of 81,908 HNC patients. In total, the pooled estimates were found to be HR= 0.75 (95% CI, 0.60–0.94) for HNC mortality and HR = 1.13 (95% CI, 1.05–1.22) for HNC survival, based on fixed effects models. Similarly, the University of Michigan Head and Neck SPORE project also suggested that HNSCC patients with lower levels of vitamin D3 intake are at higher risk of recurrence. A significant inverse trend was found between total vitamin D3 intake and recurrence (HR = 0.47, 95% CI, 0.20–1.10, p_trend_ = 0.048). However, no relationship was observed between dietary or supplemental intake, separately or together, and all-cause or HNC-specific mortality [330]. Similarly, Yokosawa et al.’s [330] findings conducted in the University of Michigan Head and Neck SPORE also suggested that HNSCC patients with lower levels of vitamin D3 intake are at higher risk of recurrence. Recurrence was seen in 9.2% of patients with ≥16.875 μg of vitamin D3 supplementation, while the use of lower oral doses of vitamin D3 ≤ 5 μg was associated with a recurrence rate of 12.96%. The researchers have found a significant inverse trend between total vitamin D3 intake and recurrence (HR = 0.47, 95% CI, 0.20–1.10, p_trend_ = 0.048). However, no relationship between dietary or supplemental intake separately and no association with all-cause or HNC-specific mortality have been observed in the study population.

Interestingly, some meta-analyses on the effect of vitamin D3 on HNSCC risk in general have so far yielded contradictory results. For instance, Weinsten et al. [331] analysed the serum calcidiol status and patient survival in the Alpha-Tocopherol, Beta-Carotene Cancer Prevention (ATBC) Study, which comprised 4616 Finnish male smokers, all of whom were participants in the Finnish Cancer Registry. It was found that the circulating serum calcidiol level was significantly decreased among individuals who subsequently died from their malignancy compared with those who did not (medians 34.7 vs. 36.5 nmol/L, respectively; *p* = 0.01). In addition, higher calcidiol status was connected to lower overall cancer mortality (HR = 0.76, 95% CI, 0.67–0.85 for highest vs. lowest quintile, p_trend_ < 0.0001). Vitamin D3 status also non-significantly improved survival from HNC; however, a vitamin D-survival relationship was observed for cancer stages I–II at diagnosis but not stages III–IV, and the interaction test was not statistically significant. Similarly, Meyer et al. [332] report that vitamin D3 status before treatment did not influence disease outcomes among Canadian HNSCC patients. No significant relationship was confirmed between combined dietary or supplemental vitamin D3 pretreatment intake or serum vitamin D3 measures and HNC overall mortality, i.e., for OS: comparing the highest and lowest quartiles of dietary/supplemental vitamin D3 intake yielded HR = 1.27, 95% CI, 0.87–1.84. The effect was not modified by cancer stage, season of initial treatment, or trial arm. In addition, no significant relationship was found between calcidiol status and cancer recurrences (HR = 1.05, 95% CI, 0.63–1.74) or second primary cancer (SPC) incidence (HR = 1.27, 95% CI, 0.87–1.84).

Although the effect of vitamin D3 status on outcome based on tumour infiltrating lymphocyte (TIL) and immune status subsets has not been discussed in HNSCC, a few articles have evaluated variations in the immune response of the subjects according to their vitamin D levels [107,303,333,334]. For instance, Bochen et al. [107] confirmed that higher calcidiol circulating concentrations were related to higher peri-tumoural immune cell infiltration. The researchers report an increase in CD4^+^ T cells, CD8^+^ T cells, CD3^+^ T cells, NK cells, macrophages, and M1 macrophages in intratumour tissue in cancer patients who received vitamin D supplementation. This suggests that the effects of vitamin D3 depend on tumour immune status. In HNSCC, one possible biological mechanism involves vitamin D regulating immune cells within HNC tumours. Interestingly, some studies confirm that vitamin D3 has a stimulating effect on the immune system of patients with HNC [303,333,334]. An example is a clinical pilot trial conducted by Walsh et al. [303] in untreated patients with newly diagnosed HNSCC and patients treated orally with 4 μg of calcitriol in the three-week interval between cancer diagnosis and surgical treatment. The clinical effectiveness of calcitriol treatment in this small clinical trial was measured by the time to HNSCC recurrence. Calcitriol was shown to reverse the immunosuppressive mechanism and to approximately double the time to cancer recurrence following surgical treatment. Specifically, immunostaining showed increased numbers of CD4^+^ cells and a highly significant increase in CD8^+^ T cells within the tumour milieux of patients who received vitamin D3 treatment. Furthermore, patients who had completed three weeks of calcitriol treatment demonstrated an approximately 10-fold greater presence of immune cells, i.e., lymphocytes, natural killer cells, and monocytes, expressing the early activation marker CD69 (*p* = 0.0007). Most importantly, the patients with HNSCC who received preoperative treatment with vitamin D3 showed a longer time to recurrence, with a 3.5-fold greater median time compared with control patients, i.e., those untreated before surgery (*p* = 0.048). Although these promising results are in line with the immune-enhancing effect of D3, they cannot confirm a definitive causal link to the effects. 

It is important to indicate that these conclusions suggesting a clinical response are from a relatively small study that needs to be expanded to a larger cohort of patients before definitive conclusions can be made about the clinical effectiveness of vitamin D3. Intriguingly, several pilot studies have been conducted using vitamin D3 analogues to modulate the maturation of immune inhibitory progenitor cells [333,334,335,336]. For instance, Lathers et al. [333] have attempted to determine if treatment with the differentiation-inducer calcidiol could diminish immune inhibitory CD34^+^ progenitor cell levels and improve a panel of immune parameters in a phase IB clinical trial. It was found that in patients with advanced HNSCC, treatment with orally administered escalating doses of 20, 40, and 60 μg of calcidiol significantly reduced levels of CD34-positive immune cells in the circulating blood while increasing the immune reactivity of peripheral blood T cells. Although no clinical responses were noted, the results of this pilot study confirm that treatment with calcidiol also increases HLA-DR expression and plasma IL-12 and IFN-γ levels in HNSCC patients. Additionally, in an earlier study with newly diagnosed HNSCC patients, Lathers et al. [334] observed that giving calcitriol decreased intratumoural infiltration of CD34^+^ cells while increasing the levels of mature dendritic cells (DCs) within the tumour tissue. Similarly, Kulbersh et al. [335] indicate that the numbers of CD34^+^ progenitor cells can be diminished by inducing their maturation and differentiation into immune stimulatory DCs during culture with calcitriol; their numbers are increased in HNSCC patients. Biopsy tumour tissues collected from newly diagnosed HNSCC patients treated for three weeks with calcitriol before surgical treatment displayed a relatively higher intratumoural proportion of mature dendritic cells with regard to CD34^+^ progenitor cells.

The full effects of vitamin D3 have not been fully characterised in HNSCC cancer patients, and the immune effects, i.e., cytokine production as a marker of Th_1_/Th_2_ balance, remain unclear. However, Walker et al. [336] confirmed that 1α,25-dihydroxyvitamin D3 and its analogues may act as immunological modulators via stimulation of intratumoural immune infiltration. Supplementation with calcitriol prior to surgery enhanced the plasma levels of both Th_1_ and Th_2_ cell mediators. Patients treated for three cycles with 4 μg of calcitriol demonstrated an increase in peripheral plasma IFN-γ, TNF-α, and IL-2, suggesting Th_1_ skewing as a result of treatment with vitamin D3. Moreover, when compared with untreated patients, HNSCC patients treated with vitamin D3 displayed significant increases in IL-6 and IL-10 plasma levels, i.e., typical Th_2_ cell mediators. The induced Th_1_ and Th_2_ cytokine profiles suggest that pre-treatment supplementation with vitamin D3 stimulates both T cell subpopulations in patients with HNSCC. Furthermore, both IL-2 and IFN-γ levels were increased in treated tumour tissues compared with those in untreated patients. Similarly, higher levels of the Th_2_ cytokine IL-6 and, to a lesser extent, IL-10 were found in the HNSCC tissue of vitamin D3-supplemented patients compared with that of untreated patients. It was also found that treatment with vitamin D3 may promote reductions in the pro-angiogenic and pro-tumourigenic factors VEGF and IL-8, as well as IL-1α and IL-1β, in the HNSCC tissue; however, significant elevations in both proangiogenic cytokines VEGF and IL-8 were noted. It can hence be concluded that, due to the variability in the relationship between plasma and tissue cytokine levels, a peripheral cytokine profile may not be an accurate reflection of the intra-tumoural immunologic profile. The cited analyses are not without limitations, i.e., the sample size within the calcitriol treatment arm is limited, the confounding factors were not sufficiently incorporated, and a mismatch exists between plasma and tissue samples, which could account for the noticeable discrepancy between the two sets of sample sizes within each treatment arm. Additionally, it is important to note that normal plasma was obtained from healthy controls, while normal tissue was pathologically normal tissue obtained from HNSCC patients, and some patient groups did not participate in the studies. Young et al. [337] indicate differences in the levels of inflammation-modulating cytokines and adipokines in patients with premalignant oral lesions versus patients that develop squamous cell carcinoma of the head and neck. The authors report that individuals with premalignant oral lesions supplemented weekly with 12 µg of calcitriol 3 in a three-week period had higher levels of anti-inflammatory mediators such as adiponectin and a decrease in pro-inflammatory mediators such as IL-6, IL-17, and leptin compared with healthy controls. It is worth noting that in cancer patients, defects in immune cell maturation and increased levels of immature forms of myeloid suppressor cells or CD34^+^ progenitor cells were also associated with lower vitamin D3 levels. These phenomena may explain the susceptibility of tissues to malignancies with oral premalignant lesions.

In summary, there is as yet no clear indication that treatment with vitamin D3 has any clinical impact. Of course, the best evidence for the benefits of vitamin D3 in the treatment of cancer would be provided by a randomised controlled trial (RCT). However, such studies have not been performed for HNSCCs so far. These would require long periods of observation. In addition, the optimal dose of vitamin D in the treatment or prevention of cancer remains uncertain, which makes it difficult to interpret and compare the results of a detailed analysis.

A large multifactorial meta-analysis was also conducted by Vaughan-Shaw et al. [270], who estimated the impact of genetic variation in the vitamin D3 pathway and circulating calcidiol on cancer outcome in a cohort of 44,165 cancer cases. Throughout the population, increased levels of 25(OH)D3 were associated with better OS (HR = 0.74, 95% CI, 0.66–0.82) and progression-free survival (PFS) (HR = 0.84, 95% CI, 0.77–0.91). The rs1544410 (BsmI) genetic variant was related to OS (HR = 1.40, 95% CI, 1.05–1.75), and the rs2228570 (FokI) variant had an association across all human cancers (HR = 1.26, 95% CI, 0.96–1.56). Unfortunately, the results did not confirm any link between VDR genetic variation and HNSCC prognosis; however, they indicated a non-significant trend between better survival and an increased 25(OH)D3 level. Furthermore, it was noted that higher circulating calcidiol was related to a significant reduction in disease progression for all cancers combined (HR = 0.84, 95% CI, 0.77–0.91). Unfortunately, the association between 25(OH)D3 level and disease progression observed in the subgroup analysis for head and neck cancer was non-significant. Another study examined the influence of genetic sequence variants (GSVs) in the vitamin D3 metabolism pathway, candidate-based GSVs, i.e., VDR, GC, CYP24A1, CYP27A1, CYP27B1, and CYP2R1, on overall survival (OS) and second primary cancer (SPC) in head and neck cancer patients [338]. Significantly lower serum vitamin D3 levels were found in patients carrying the minor alleles of GC (rs4588: p.Thr436Lys) and CYP2R (rs10500804). Moreover, the CYP24A1 (rs2296241: c.552A>G) genotype was significantly associated with OS (HR = 1.23, 95% CI, 1.00–1.51, *p* = 0.05), and CYP2R1 (rs1993116: c.226-2771C>T) was related to SPC (HR = 0.59, 95% CI, 0.43–0.81, *p* = 0.001). Zeljic et al. [329] confirmed that oral cancer cases with the VDR FokI ff wild-type genotype had significantly lower OS (p_log rank_ = 0.012) compared with heterozygous and mutated genotypes combined. A stratified analysis by lymph node metastases and stage found the ff variant to be associated with shorter OS in patients with and without lymph node involvement (*p* = 0.025, *p* = 0.040, respectively) and in stage III tumours (*p* = 0.026). The findings confirm that VDR polymorphisms could be considered independent prognostic factors.

Despite their great number, cancer treatment toxicities have only been analysed by single studies in relation to vitamin status [339,340,341]. Vitamin deficiency has several possible causes in people with cancer, including unbalanced dietary intake, problems associated with swallowing disorders, altered metabolism, adverse effects of treatment, and inflammation. The problem is serious because cancer cachexia is associated with adverse clinical outcomes, including poor prognosis, dose-limiting toxicity, impaired performance and immunity, and reduced quality of life [342,343]. These observations were confirmed in a study by Canadian researchers who analysed micronutrient, protein, and energy intakes in HNSCC individuals based on a tool to assess Nutrition Impact Scores (NIS) and compared the results with the European Society for Parenteral and Enteral Nutrition guidelines for cancer patients. The findings support the fact that the majority of HNSCC patients did not meet recommended dietary intakes for vitamins D3, E, C, folate, and magnesium at any time point in the study [344]. Unfortunately, there is currently a low level of evidence regarding vitamin status in cancer patients, and the European Society for Clinical Nutrition and Metabolism (ESPEN) recommends further research into the estimation of micronutrients in relation to oncological outcome [345]. However, current ESPEN recommendations indicate that oral consumption of supplements may contribute to avoiding malnutrition and vitamin and micro/macroelement deficiency [346]. Additionally, the United Kingdom National Multidisciplinary Guidelines Team indicated that nutritional support and intervention (dietary counselling and/or supplements) constitute an integral component of head and neck cancer management [347]. A prospective cohort study by Nejatinamini et al. [348] estimated the changes in vitamin D3 status at diagnosis of HNSCC and after 6–8 weeks of RT/CRT in relation to body composition, inflammation, and mucositis. It was found that poor vitamin D3 status was associated with cancer complications such as skeletal muscle loss and mucositis. Patients who developed mucositis had significantly lower dietary intakes of vitamins and plasma calcidiol levels (*p* < 0.02) and lost a considerable amount of muscle and fat mass over the course of treatment. There was a trend towards greater muscle loss in patients with 25(OH)D3 < 50 nmol/L compared with patients with 25(OH)D3 ≥ 50 nmol/L (*p* = 0.07). In summary, dietary intervention or the use of supplements may be a feasible intervention for the prevention of cancer treatment toxicities and recurrence in HNC patients [330,347]. Furthermore, to avoid malnutrition, including deficiency of vitamins and micro/macroelements, nutritional intervention should be implemented early when deficits are detected. Nutritional parameters should be regularly monitored throughout the course of the disease, and quality of life parameters, including nutrition and swallowing, should be measured at diagnosis and at regular intervals post-treatment [330,347,348].

Table 3 summarises the key studies found in the review and the data collected from them. 

## 4. Conclusions

The main goal of this review article was to summarise the possible role of vitamin D3 in the development and progression of cancer of the head and neck region. It hence summarises the current state of knowledge on the aetiopathogenesis of HNC related to vitamin D3 together with some other selected issues. Although the anti-cancer effects of vitamin D3 have been proven in many in vitro and in vivo studies, recent evidence indicates that these effects are also regulated by several other factors and that existing knowledge is often inconclusive, as confirmed by studies in animal models and human biological material and in epidemiological studies.

There is a clear need for further studies evaluating the effects of the vitamin D3 system on the evolution of HNSCC and the potential benefits of alleviating vitamin D3 deficiency on tumour growth and progression; this is spurred by the growth of knowledge on the pleiotropic effects of vitamin D3, together with data acquired by multicentre analyses, i.e., molecular studies, pre- and clinical early trials, long-term observational and intervention population studies, and key opinion-forming systematic reviews. Nevertheless, the overwhelming majority of publications indicate that a higher level of dietary vitamin D3 and its analogues, and consequently, higher blood serum levels, inhibit the molecular landscapes of head and neck carcinogenesis and tumour progression. However, a number of studies have inconclusive final results, which may be due to several factors, such as heterogeneity of patient samples, insufficient sample sizes among comparative patient groups and controls, short observation periods or variable follow-up times, and a failure to consider seasonal and lifestyle factors. In addition, many studies fail to use standardised food frequency questionnaires to assess dietary intake of vitamin D3 and draw on groups of patients from different population types.

Despite the above limitations, it should be emphasised that almost all recent publications clearly indicate that proper supplementation of dietary vitamin D3 analogues has a beneficial inhibitory effect on all stages of head and neck cancer progression and that it has the potential to improve patient prognosis. Further studies with long-term follow-ups, larger study groups, and extended intervention studies are needed to establish definitive recommendations for vitamin D3 supplementation in both healthy individuals and cancer patients, which will allow for unequivocal conclusions regarding the role of dietary vitamin D3 in physiological conditions and cancer.

More randomised controlled trials (RCTs) on the effectiveness of vitamin D3 supplements as a preventive measure against HNC or on treatment outcomes are needed to provide more data and guide recommendations for vitamin D3 supplementation in HNC patients. The recent findings on vitamin D3 in HNC could also play a key role in the design of chemoprevention and new potential therapeutic targets for better diagnosis and treatment.

## Figures and Tables

**Figure 1 nutrients-15-02592-f001:**
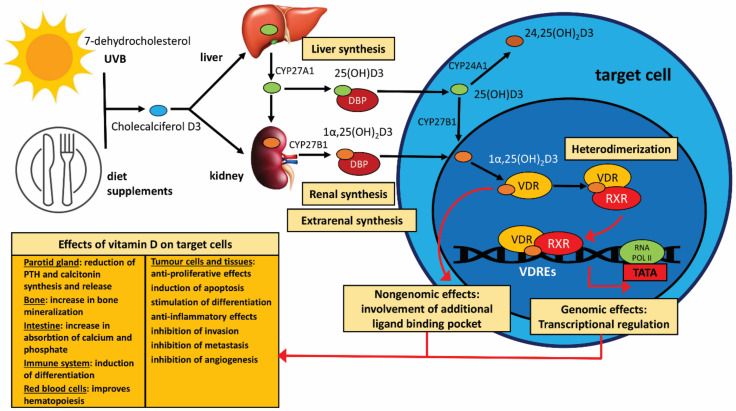
Vitamin D metabolism. Cholecalcipherol received through either sub skin exposure to UV-B or diet is converted to calcidiol—25-hydroxyvitamin D3 [25(OH)D3] via CYP27A1 enzyme in the liver. Next, 25(OH)D3 is converted to calcitriol—1α,25-dihydroxyvitamin D3 [1α,25(OH)_2_D3] via CYP27B1 enzyme in the kidney. Both 25(OH)D3 and 1α,25(OH)_2_D3 circulate bound to a specific vitamin D binding protein (DBP). Calcitriol, the active form of vitamin D, constitutes the primary ligand for the vitamin D receptor (VDR). Vitamin D-bound VDR heterodimerises with retinoid X receptor (RXR), and then the complex of ligand-bound VDR-RXR couple translocates to the target cell nucleus and engages with VDR response elements (VDREs) in several regulatory regions within the target gene promoters, recruits co-modulators, and effects transcription. Both direct and indirect transcriptional modulation lead to anti-proliferative effects, induction of apoptosis, stimulation of differentiation, anti-inflammatory effects, inhibition of invasion and metastasis, and inhibition of angiogenesis. Vitamin D is metabolised for excretion by CYP24A1 enzyme, which produces the inactive metabolite 24,25(OH)_2_D3. Abbreviations: 25(OH)D3: 25-hydroxyvitamin D3, calcidiol; 1α,25(OH)_2_D3: 1α,25-dihydroxyvitamin D3, calcitriol; 24,25(OH)_2_D3: 24,25-dihydroxyvitamin D3; VDR: vitamin D receptor; RXR: retinoid X receptor; VDREs: VDR response elements.

**Figure 2 nutrients-15-02592-f002:**
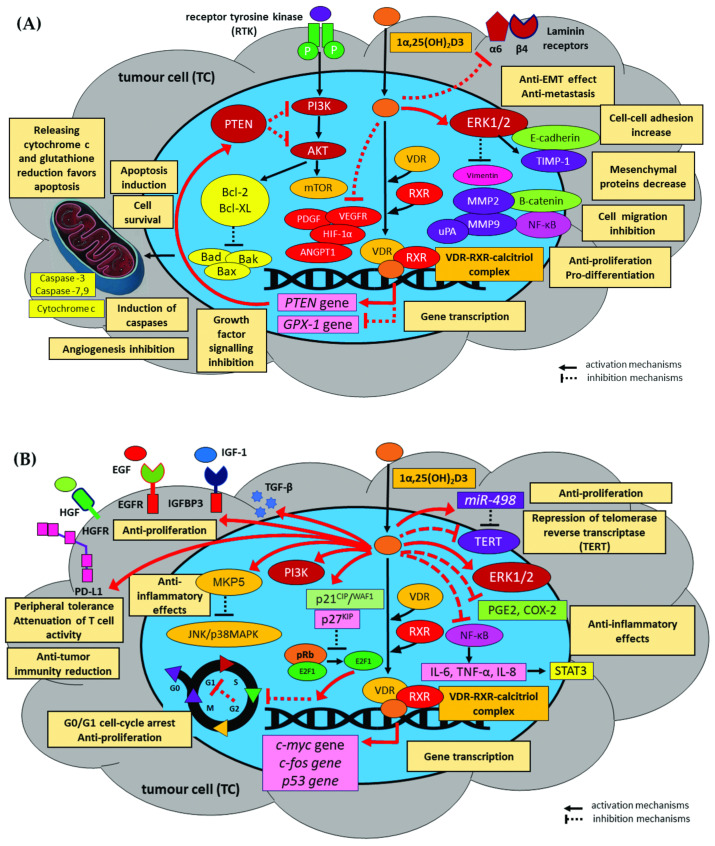
The molecular mechanisms relevant to the pathophysiology of HNSCC regulated by calcitriol. (**A**) Calcitriol induces apoptosis, stimulates differentiation, and inhibits proliferation, invasion, and metastasis, as well as angiogenesis and growth factor signalling. (**B**) Other anticancer mechanisms of calcitriol cause anti-proliferative and anti-inflammatory effects; they also reduce anti-tumour immunity and peripheral tolerance and attenuate T cell activity; 1,25(OH)_2_D3 hampers the transition from the G0/G1 to the S phase of the cell cycle either directly, through upregulation of different cyclin-dependent kinase inhibitors, or indirectly, through the induction of other growth factors (e.g., TGF-β, EGF). In addition, 1,25(OH)_2_D3 induces apoptosis through activation of the intrinsic apoptotic pathway or by interference with other signalling pathways such as TNF-α, EGF, β-catenin, and prostaglandins. 1,25(OH)_2_D3 also has immunosuppressive activity, as indicated by the repression of NFκB-mediated gene transcription; this results in the suppressed production of inflammatory cytokines IL-1, IL-6, IL-8, and TNF-α. Abbreviations: PD-L1: programmed death-ligand 1, PI3K: phosphoinositide 3-kinase, AKT: protein kinase B, also known as PKB, mTOR: mammalian target of rapamycin, VEGFR: receptors for vascular endothelial growth factor, MMP: matrix metalloproteinase enzymes, ERK: extracellular signal-regulated kinase, TIMP: tissue inhibitors of metalloproteinase, PDGF: platelet-derived growth factor, VEGF: vascular endothelial growth factor, ANGPT1: angiopoietin 1, HIF-1α: hypoxia-inducible factor 1-alpha, Bad: Bcl-2-associated agonist of cell death, Bcl-XL: B-cell lymphoma extra-large, c-fos, c-myc: transcription factors, pRb: retinoblastoma protein, PIK3: phosphoinositide 3-kinase, AKT: protein kinase B (PKB), MKP5: MAPK phosphatase 5, MAPK: mitogen-activated protein kinase, PTEN: phosphatase and tensin homolog, GPX-1: glutathione peroxidase 1, RKT: receptor tyrosine kinase, PGE2: prostaglandin 2, COX-2: cyclooxygenase-2, TNF-α: tumour necrosis factor-α, IL-6: interleukin 6, IL-8: interleukin 8, NF-ĸB: nuclear factor kappa-light-chain enhancer of activated B cells, E2F1: transcription factor E2F1, JNK: c-Jun *N*-terminal kinase, EGFRL: epidermal growth factor receptor, HGFR: hepatocyte growth factor receptor, IGFBP3: insulin-like growth factor-binding protein 3, IGF-1: insulin-like growth factor 1, PD-L1: programmed death-ligand 1, TERT: telomerase reverse transcriptase, STAT3: signal transducer and activator of transcription 3.

**Table 1 nutrients-15-02592-t001:** The main activities exerted by 1α,25(OH)_2_D3 on the activity of various immune and inflammatory cell populations.

Immune Cells	Mechanisms	
	Stimulation	Inhibition
Dendritic cells (DCs)	- Increased production of the anti-inflammatory cytokine, e.g., IL-10.	- Reduction in DCs, the expression of class II MHC and costimulatory molecules (CD40, CD80, CD86), and thus induction of T cell proliferation and activation, and the differentiation of T regulatory cells (Treg). - Decreased production of pro-inflammatory cytokines, e.g., IL-12 and TNF-α. - Suppression of IL-12 expression (by binding of VDR/RXR to the NF-ĸB site in the IL-12p40 promoter)
Monocytes and macrophages	- In monocytes, activation of Toll-like receptors (TLRs) and IFN-α upregulates CYP27B1, thus inducing autocrine and paracrine production of calcitriol. - Calcitriol stimulates the differentiation of monocytes into macrophages. - In monocytes and macrophages, 1α,25(OH)_2_D3 induces the production of cathelicidin antimicrobial peptide/LL37 (CAMP/LL37), thus contributing to antimicrobial activity. - Calcitriol enhances the anti-inflammatory activity of macrophages by inducing anti-inflammatory cytokines, e.g., IL-10.	- Calcitriol enhances the anti-inflammatory activity of macrophages by decreasing the production of pro-inflammatory factors, e.g., IL-1β, IL-6, TNF-α, RANKL, and COX-2. - The phosphorylation of both AKT and its downstream target IκBα in macrophages was markedly suppressed by calcitriol through up-regulation of THEM4, an AKT modulator protein. - Calcitriol restrains macrophage-mediated inflammatory processes by suppressing the AKT/NF-ĸB/COX-2 pathway. - 1α,25(OH)_2_D3 promotes negative feedback regulation of TLR signalling via targeting microRNA-155-SOCS1 in macrophages.
CD4+ T cells	- Calcitriol has a direct effect on naïve CD4+ T cells to enhance the development of Th2 cells. - In Th2 cells cultured in the absence of polarising cytokines but in the presence of Ag, anti-CD3, and CD28, calcitriol induces IL-4 and GATA3. - Calcitriol upregulates the expression of FoxP3 by binding to the FoxP3 promoter and inducing CDS4+CD25+Foxp3. Tregs differentiate in vitro and in vivo. - 1α,25(OH)_2_D3 induces IDO-1+ tolerogenic DCs and enhances Treg activation. - In vitro calcitriol induces Treg cells, the production of anti-inflammatory cytokines IL-10 and TGFβ, and increases Treg cell numbers.	- Calcitriol inhibits IFN-γ production in Th1 cells via a consequence of VDR/RXR binding to a silencer region in the hIFN-γ promoter. - Calcitriol inhibits IL-2 production in Th1 cells by inhibiting the NFAT/AP-1 complex by VDR/RXR and RUNX1 sequestration by nVDR. - In Th2 cells, calcitriol inhibits IL-4 production in naïve activated (CD62 ligand-CD4^+^) T cells during in vitro polarisation. - Calcitriol inhibits proliferation but not the suppressive function of regulatory T cells in the absence of antigen-presenting cells. - In vitro Th17 cell differentiation and activation are inhibited by calcitriol (this effect is mediated by the inhibition of production of Th17-related cytokines and transcription factors such as IL-17A, IL-17F, RORC, and CCR6). - The VDR-RXR complex inhibits IL-17A production by competition with NFAT binding to the IL-17A promoter, the inhibition of Smad7 transcription, sequestration of RUNX1 by VDR, and induction of Foxp3 (which inhibits NFAT and RUNX1 function). - Calcitriol reduces the formation of Th17 by inhibiting the transcription factor RORγt, both in the presence and absence of IL-23. - Calcitriol reduces the frequency of CCR6+ cells producing both IFN-γ and IL17A (non-classic Th1 cells). - Calcitriol inhibits IFN-γ and IL-17A, IL-17F, GM-CSF, TNF-α production by activated TCRγδ T cells.
CD8+ T cells (CTLs)	- In vivo calcitriol downregulates CD8+ T cell activity and the production of IFN-γ and TNF-α.	
NK cells	- In NK cells, calcitriol induces the cytolytic killing capacity. - 1α,25(OH)_2_D3 induces maturation and function of invariant NKT cells (iNKT). - Induction by calcitriol of production of IL-4 from iNKT cells.	- Calcitriol inhibits the development of NK cells, their cytotoxicity, and IFN-γ production.
B cells	- Calcitriol exerts different activities in the various stages of B cell differentiation. - 1α,25(OH)_2_D3 induces B cell apoptosis. - Calcitriol upregulates IL-10 production by B cells by binding directly to the IL-10 promoter region.	- Calcitriol exerts inhibitory activities on proliferation. - Calcitriol inhibits differentiation of B cells into plasma cells, immunoglobulin class switching, and antibody production. - Calcitriol downregulates CD86 expression on B cells, suggesting reduced stimulation of T cells. - Calcitriol inhibits the expression of CD74, contributing to the surface exposure of MHC-II molecules on activated B-lymphocytes (a plausible mechanism for downregulation of acute inflammatory conditions). - Calcitriol impairs NF-κB function and CD40 activation in human naïve B cells. - Calcitriol controls circulating IgE by the vitamin D receptor (in vitro, unresponsiveness of B cells to vitamin D is associated with augmented IgE expression).

DCs: dendritic cells, CTLs: CD8^+^ cytotoxic cells, CD4^+^CD25^+^Foxp3^+^T_regs:_ regulatory T cells, NF-ĸB: nuclear factor NF-kappa-B, TLRs: toll-like receptors, RANKL: receptor activator for nuclear factor κB ligand, COX-2: cyclooxygenase 2 enzyme, CAMP/LL37: cathelicidin antimicrobial peptide/LL37, AKT: protein kinase B (PKB), SOCS1: suppressor of cytokine signalling 1, GATA3: GATA family of transcription factor 3, TLRs: toll-like receptors, RUNX1: runt-related transcription factor 1, IDO-1: Indoleamine 2,3-dioxygenase 1 enzyme, NFAT: nuclear factor of activated T-cells, RORγt: RAR-related orphan receptor gamma (transcriptional factor).

**Table 2 nutrients-15-02592-t002:** The effect of vitamin D in the selected animal and in vitro models of HNSCC.

Author	Vitamin D in Animal and In Vitro Models of HNSCC	
	Study Design	Mechanisms/Underlying Signalling Pathway of Calcitriol and Its Analogues
Animal models		
Meier et al. [255]	- The role of systemic intraperitoneal calcitriol injection in tumour formation in a hamster buccal pouch model of HNSCC carcinogenesis. - Induction of carcinogenesis via 7,12-dimethylbenz(a)anthracene (DMBA).	- Calcitriol injection prevented tumour formation in the animal model of HNSCC carcinogenesis. - The mean time to tumour formation was delayed in those animals treated with calcitriol.
Shintani et al. [256]	- The anti-tumour effect of eldecalcitol (ED-71, 1α,25-dihydroxy-2β-3-hydroxypropoxy vitamin D3) after implantation of A431 epidermoid carcinoma cells in nude BALB/c mice.	- Tumour formation was reduced by the ED-71 treatment. - No significant difference in tumour CYP24A1 expression was observed between the ED-71-treated and control groups.
Shintani et al. [257]	- The anti-tumour effect of oral administration of vitamin D3 analogue eldecalcitol (ED-71) in athymic nude mice with A431-derived squamous cell carcinoma (SCC).	- ED-71 inhibited the growth of SCC cells. - Vitamin D3 analogue down-regulated the expression of HBp17/FGFBP-1, a factor that enhances angiogenesis as well as promotes tumour growth, and FGF-2. - Oral administration of ED-71 significantly inhibited the growth of A431-derived tumours in athymic nude mice. - ED-71 may act directly on the cancer cells or on endothelial cells by modulating the tumour microenvironment.
Verma et al. [118]	- The anti-tumour effect of oral administration of vitamin D3 in C57BL/6 mice with dysplasia or OSCC. - Induction of carcinogenesis via 4-nitroquinoline-1-oxide (4NQO).	- Mice on the 100 IU diet displayed a lesser occurrence of high-grade dysplasia, or OSCC, accompanied by a greater infiltration of CD3^+^ T cells compared with animals on 25 IU and 10,000 IU diets. - The highest incidence of OSCC in animals maintained on 10,000 IU diet. - Mice on 100 IU and 10,000 IU diets presented higher VDR and CYP24A1 enzyme immunostaining in high-degree dysplasia and OSCC compared with normal ones. - Short-term treatment of mice on a 25 IU diet with calcitriol inhibited tumour growth.
Vincent-Chong et al. [258]	- The anti-tumour effect of oral administration of vitamin D3 in C57BL/6 mice with OSCC. - Induction of carcinogenesis via 4-nitroquinoline-1-oxide (4NQO) in drinking water for 16 weeks.	- Calcitriol administered concurrently with 4NQO decreased the frequency of oral dysplastic premalignant lesions by 57% compared with 4NQO-only controls. - Mice treated with calcitriol for longer periods exhibited higher CYP24A1, lower serum 1,25(OH)_2_D3 levels, and a higher incidence of an invasive OSCC phenotype.
Bothwell et al. [200]	- The chemopreventive effect of calcitriol on the efficacy of the EGFR inhibitor erlotinib against HNSCC in patient-derived xenografts (PDX) and 4NQO carcinogen-induced HNSCC model.	- Treatment with the combination regimen was related to the greatest reduction in the degree of dysplasia and tumour incidence and volume. - Phospho-EGFR and phospho-AKT pathway downregulation was observed in the tongues of PDX-bearing mice treated with calcitriol and erlotinib.
In vitro models		
Osafi et al. [260]	- The anti-tumour effect of calcitriol on CAL27, SCC15, and SCC25 OSCC cell lines.	- Growth-inhibiting effects. - Induction of differential dose-dependent modulation of proliferative tumour phenotypes. - Decrease in the viability of cancer cells and the activation of two key apoptotic pathways, i.e., caspase and Bcl/Bax.
Lu et al. [261]	- The anti-tumour effect of eldecalcitol (ED-71), an analogue of calcitriol, in the progression of mouse xenograft tumour models of SCC-15 and CAL-27 OSCC cell lines.	- Inhibition the proliferation and migration of oral cancer cells. - Blocking the cell cycle in the G0/G1 phase. - Increasing apoptosis through glutathione peroxidase-1 (GPX1) downregulation.
Shintani et al. [256,257]	- The anti-tumour effect of calcitriol and eldecalcitol (ED-71) for OSCC, i.e., cell lines (NA-HO-1-n-1 and UE-HO-1-u-1) and an epidermoid carcinoma/SCC (A431 cell line).	- Inhibition by ED-71 the growth of the cancer cell lines in a dose-dependent manner. - ED-71 and calcitriol suppressed HBp17/FGFBP-1 expression by inhibiting the NF-κB pathway. - ED-71 decreased the growth of cancer cell lines by 100 times less than calcitriol. - *CYP24A1* mRNA and protein expression in the OSCC cells increased in a dose-dependent manner after ED-71 treatment.
Rosli et al. [231]	- The anti-tumour effect of calcitriol on HBp17/FGFBP-1 gene/protein expression in UE cell lines of oral squamous cell carcinoma (OSCC).	- Restriction of the activity of HBp17/FGFBP-1 in cancer cells treated with calcitriol (*HBp17/FGFBP-1* mRNA and protein levels were down-regulated). - HBp17/FGFBP-1 expression is affected by calcitriol via the NF-κB pathway (IκBα up-regulation).
Huang et al. [252]	- The role of glutathione peroxidase-1 (*GPX1*) gene in vitamin D3-mediated progression of ACC-M, SACC-83, and ACC-2 cell lines of salivary adenoid cystic carcinoma (SACC).	- Decreasing GPX1 expression inhibited SACC proliferation, motility and invasion, as well as chemoresistance and uPA secretion. - Decreasing GPX1 expression promoted apoptosis via the NF-κB(p65) pathway (IκBα activation). - Calcitriol modified NF-κB/GPX1/uPA expression, inhibiting cisplatin resistance and cell invasion ability of the SACC cells.
Satake et al. [175]	- The anti-tumour effect of calcitriol and its derivative 22-oxa-1,25(OH)_2_D3 on KB cell lines of oral floor squamous cell carcinoma.	- Calcitriol and 22-oxa-1,25(OH)_2_D3 suppressed cell proliferation, induced apoptosis, and arrested the cell cycle. - Calcitriol and its derivatives increased the sensitivity of chemotherapeutics. - Calcitriol and 22-oxa-1,25(OH)_2_D3 downregulated several angiogenesis factors and pro-apoptotic survival in KB cell lines.
Chiang et al. [268]	- The anti-cancer effect of a calcitriol derivative, MART-10 (19-nor-2α-3-hydroxypropyl-1α,25-Dihydroxyvitamin D3), in FaDu and SCC-25 cell lines of squamous head and neck carcinoma (HNSCC) in vitro and in vivo.	- MART-10 has greater antiproliferative activity than calcitriol in inhibiting cancer cell growth without inducing hypercalcemia (MART-10 showed greater potential than calcitriol in suppressing FaDu and SCC-25 squamous carcinoma cell lines). - MART-10 increased cell cycle arrest at G0/G1, accompanied by a greater downregulation of Ki-67 expression and upregulation of p21^WAF1/CIP1^ and p27^KIP1^. - MART-10 suppressed telomerase expression in the SCC-25 cell line to a greater extent than calcitriol.
Atuksu et al. [263]	- The anti-tumour effect of vitamin D3 analogue EB1089 in human SCC25 HNSCC cell lines.	- EB1089 arrested proliferating human SCC25 HNSCC cells in G0/G1.
Lin et al. [264]	- The anti-tumour effect of vitamin D3 analogue EB1089 in human SCC25 HNSCC cell lines.	- EB1089 acted as a key chemoprevention agent. - Anti-cancer effect was related to antiproliferative, pro-differentiating, and gene-protective effects through the induction of GADD45α. - EB1089 down-regulated of N-cadherin expression.
Yang et al. [269]	- The antimetastatic effect of MART-10 on the FaDu HNSCC cell line.	- Calcitriol and MART-10 repressed the migration and invasion of tumour cells, with MART-10 being much more potent than vitamin D3. - The antimetastatic effect was mediated by attenuating the epithelial-mesenchymal transition (EMT) through inhibition of EMT-promoting Sail and Twist, as well as intracellular MMP-9 expression and extracellular MMP activity. - Calcitriol and MART-10 increased E-cadherin and decreased N-cadherin expression.
Prudencio et al. [265]	- The anti-tumour effect of vitamin D3 analogue, EB1089, in murine AT-84 SCC cell lines.	- Calcitriol and EB1089 arrested the proliferation of cultures in G0/G1 cell-cycle phase through down-regulation of p21^WAF1/CIP1^ and up-regulation of p27^KIP1^ protein expression of cyclin-dependent kinase inhibitors (CDKI). - Calcitriol also enhanced the expression of GADD45α, apparently by a p53-independent mechanism.
Alagbala et al. [266]	- The anti-tumour effects of two vitamin D3 analogues, i.e., 19-nor-1α,25-dihydroxyvitamin D2 (paricalcitol) and 1α-hydroxymethyl-16-ene-24,24-difluoro-25-hydroxy-26,27-bis-homovitamin D3 (QW-1624F2-2) in human squamous cell carcinoma cell lines.	- Both vitamin D3 analogues inhibited cell cycle regulators such as p21^WAF1/CIP1^ and cyclin-dependent kinase 2 (CDK2) and increased p27^KIP1^ protein expression, inducing G0/G1 cell cycle arrest and blockage of DNA synthesis. - The vitamin D3 derivatives induced apoptosis, caspase-3 cleavage, and increased expression of pro-apoptotic MEKK-1 kinases. - Both vitamin D3 analogues inhibited the phosphorylation of AKT, MEK, and ERK1/2, which promote cell growth and survival.
Xiao et al. [267]	- The potential effects of aspirin combined with vitamin D3 on cell proliferation and apoptosis in oral cancer CAL-27 and SCC-15 cells.	- Vitamin D3 synergistically or additively potentiated the anti-proliferative and apoptosis-induced effects of aspirin against both OSCC cell lines in dose- and time-dependent manners. - The anti-carcinogenic activity of ASA and vitamin D3 was related to suppression of Bcl-2 and p-ERK1/2/AKT and Ras/MEK signalling pathways, which resulted in decreased rates of cell proliferation and higher rates of cell apoptosis.
Ibrahimovic et al. [271]	- The effect of vitamin D3 on hsa-miR-7-1 and hsa-miR-335 and nucleophosmin (NPM) expression in human CAL-27 and SCC-25 HNC cell lines. - The effect of vitamin D3 on radiation hypersensitivity and therapeutic resistance.	- Hsa-miR-7-1 and hsa-miR-335, two oncogenic molecules known to play roles in HNC or precursor lesions, were modulated by vitamin D3. - Vitamin D3 acted through MAPK, PI3K, RAS, and chemokine signalling. - Inhibition of hsa-miR-7-1 and hsa-miR-335 activity led to negative regulation of cell cycle and tumour proliferation and enhanced angiogenesis. - Hsa-miR-7-1 and hsa-miR-335 were found to modulate Canonical Wnt signalling. - Vitamin D3 treatment reduced the expression of NPM in SCC-25 cells, potentially promoting apoptosis avoidance, and increased viability, growth, and proliferation in HNC cancer cells. - Vitamin D3 regulated the expression of heat shock protein β-1 (HSP27) and played a role in radiation hypersensitivity and therapeutic resistance.
Niu et al. [284]	- The relation of vitamin D3 activity to peroxiredoxin-1 (PRX-1) expression in human OSCC in CAL-27 and SCC-15 cells.	- Vitamin D3 down-regulated the PRX-1 protein level, which subsequently inhibited tumour cell invasion and migration. - Vitamin D3 acted via regulation of EMT and NFκB activity.

OSCC: oral squamous cell carcinoma, ED-71: eldecalcitol, GPX1: glutathione peroxidase-1, HBp17/FGFBP-1: heparin-binding protein 17/fibroblast growth factor-binding protein-1, MART-10: 19-nor-2α-3-hydroxypropyl-1α,25-Dihydroxyvitamin D3, GADD45G: growth arrest and DNA-damage-inducible protein GADD45 gamma, NPM: nucleophosmin, HSP27: heat shock protein β-1, PRX-1: peroxiredoxin-1.

**Table 3 nutrients-15-02592-t003:** The selected studies evaluating vitamin D levels in HNC patients.

Author	Vitamin D Plasma/Serum Concentrations in Head and Neck Cancer	
	Study Design	Results
Kapała et al. [308]	- Cross-sectional observational study; 500 Caucasian cancer patients with HNC (Polish patients). - The concentration of vitamin D3 was defined as the sum of 25(OH)D2 and calcidiol in the serum of patients.	- Vitamin D3 deficiency (defined as a serum concentration of calcidiol ≤30 ng/mL) in 66.8% and very low vitamin D3 concentration (defined as ≤20 ng/mL) in 38.6%. - HNC were also more frequent in the subgroup of patients with vitamin D3 deficiency regardless of the declared cholecalciferol supplementation (*p* = 0.039). - Analysis did not show that the distribution of patients with or without metastases varied within the subgroups of vitamin D3 status *p* = 0.159.
Pu et al. [286]	- Meta-analysis; sixteen observational studies, including nine case–control and seven cohort studies. - 81,908 participants (Europe, North America, and Asia-Pacific populations). - Calcidiol and vitamin D3 dietary intake, additional supplements of vitamin D3, and *VDR* gene polymorphisms.	- A significant inverse association between HNC incidence and vitamin D3 exposures, including a circulated concentration of calcidiol and vitamin D3 intake (OR 0.68, 95% CI: 0.59–0.78 and OR 0.77, 95% CI: 0.65–0.92). - A significantly reduced risk of HNC incidence for this mutation in two genetic models (*ff* vs. *Ff* + *FF* and *ff* vs. *FF*). - A reduced HNC risk was observed in Caucasians for the recessive model (*ff* vs. *Ff + FF*). - A significant reduction in HNC risk was observed in the overall population (*tt* vs. *Tt* + *TT* and *tt* vs. *TT)*, as well as among Caucasian populations. - An inverse association between HNC mortality and calcidiol levels (HR 0.75, 95%CI: 0.60–0.94).
Fanidi et al. [300]	- The European Prospective Investigation into Cancer and Nutrition (EPIC) cohort study; nested case-control study. - A total of 350 patients with HNC (oral, oropharynx, pharynx, larynx, and others) vs. 940 controls. - Calcidiol in the serum of patients.	- A doubling of calcidiol was associated with 30% lower odds of HNC (OR_log2_ 0.70, 95% CI: 0.56–0.88, p_trend_ = 0.001). - An inverse association with risk of larynx and hypopharynx cancer combined (OR 0.55, 95 CI%: 0.39–0.78) and oral cavity cancer (OR 0.60, 95 CI%: 0.42–0.87). No association was observed with risk of oropharynx cancer. - An association only in smokers or ex-smokers. - An overall inverse association between pre-diagnostic circulating calcidioland survival and post-HNC (HR 0.73 95%, CI: 0.55–0.97).
Afzal et al. [301]	- The prospective population-based cohort study; tobacco-related cancer individuals from the Copenhagen City Heart Study, CCHS (Danish population). - A total of 1081 participants with a tobacco-related cancer and 1506 with other cancers. - Calcidiol plasma level.	- A clear inverse association between calcidiol and smoking-related cancers and circulating vitamin D3 (OR_1/2_ = 1.20), with a particularly strong association for HNC. - Decreasing calcidiol levels were associated with increasing cumulative incidence of tobacco-related cancer. - Concentrations for smoking-related cancers were OR 1.75 (95% CI: 1.33–2.30), including head and neck cancer OR 1.28 (95% CI: 1.06–1.54) for calcidiol < 5 vs. ≥20 ng/mL, while no association was noted for non-smoking-related cancers overall (OR_1/2_ = 0.95). Additionally, multivariable adjusted hazard ratios for a 50% reduction in calcidiol concentration were OR 1.44 (95% CI, 1.19–1.73) for 122 head and neck cancer patients.
Anand et al. [108]	- Cross-sectional study; 110 cases: 87 oral squamous cell carcinoma (OSCC), 23 premalignant lesions, and healthy (Indian population). - Case-control study (110 cases vs. 95 controls). - Calcidiol serum level and VDR expression by IHC + the quality of life questionnaire QOL.	- All patients with advanced oral cancer were deficient in vitamin D3 scores (76.3%). - In cancer, significantly lower calcidiol, D3-score: −1.90 vs. −1.33 (*p* = 0.002). - The mean VDR expression was found to be higher in cases compared with controls, but the difference was not statistically significant. - The quality of life was improved with vitamin D3 supplementation.
Udeabor et al. [288]	- Case-control study (110 cases vs. 95 controls); 51 patients with oral squamous cell carcinoma, OSCC (Arab population). - Calcidiol in the serum of patients.	- A total of 74.51% had moderate to severe vitamin D3 deficiencies, whereas only 20.35% had a moderate deficiency in the control group with no severe deficiency (OR 1.65, 95% CI: 0.98–2.77, *p* = 0.001). - A positive relationship between vitamin D3 deficiency and OSCC risk (especially at levels below 25 ng/mL) increases the likelihood of developing a malignant neoplasm by 1.65-fold.
Meyer et al. [332]	- Cross-sectional study; 522 HNC (Canadian population). - Calcidiol serum level and food frequency questionnaire (FFQ): total intake of vitamin D3 from diet and supplements.	- No significant association between dietary or serum vitamin D3 measures and the three HNC outcomes: HNC recurrence (HR 1.12, 95% CI: 0.63–1.74), second primary cancer, SPC incidence (HR 0.72, 95% CI: 0.40–1.30), and overall mortality (HR 0.85, 95% CI: 0.57–1.28).
Arem et al. [306]	- Nested case-control, prospective study; 340 cases of HNCSCC (oral, pharynx, and larynx) and 340 controls; Finnish male smokers. - The Alpha-Tocopherol, Beta-Carotene Cancer Prevention (ATBC) Study. - Calcidiol serum level.	- No association between calcidiol and HNC risk, although the latter was underpowered to detect any notable association with risk. In results, calcidiol < 25 vs. 50–75 nmol/L (OR 0.96, 95% CI: 0.58–1.59, *p* = 0.65). - Lowest vs. highest quartile of 25(OH)D3 OR 1.02, 95% CI: 0.62–1.69.
Grimm et al. [290]	- Cross-sectional case-control study. - Oral precursor lesions: simple hyperplasia, 11; squamous intraepithelial neoplasia, SIN I-III, 35; and OSCC specimen, 42 (German population). - Calcidiol serum level and expression of VDR by IHC.	- No patient showed normal (>35 ng/mL) serum vitamin D3 levels or mild (25–35 ng/mL) vitamin D3 deficiency. - A total of patients had 38% moderate (12.5–25 ng/mL) vitamin D3 deficiency, and 62% had severe (<12.5 ng/mL) vitamin D3 deficiency. - In comparison with normal tissue. a significantly (*p* < 0.05) increased expression of VDR was observed in tumour cells of OSCC.
Bochen et al. [107]	- Cross-sectional study; 231 HNSCC and 232 controls (German population). - Survival and immune cells (intra- and peritumoural infiltration with T-cells, NK-cells, and macrophages) and NK cell activity. - Calcidiol serum level and immune cells by IHC.	- The median calcidiol serum level was significantly lower in the HNSCC group than in the healthy control group (*p* < 0.0001). - A significant shorter OS in HNSCC patients with low vitamin D3 levels compared with the vitamin D3-high HNSCC group (p_log-rank_ = 0.0085). - A significantly higher intratumoural and/or stromal infiltration of immune cells in the vitamin D3-high patients compared with the vitamin D3-low patients.
Nejatinamini et al. [348]	- Cross-sectional study; 38 HNC patients (Canadian population). - Calcidiol pre- and post-treatment levels (of radiation therapy with or without chemotherapy) and FFQ (3 days). - Mucositis and muscle loss level.	- Mucositis calcidiol mean ± SD: 47.2 ± 17.9 vs. 62.3 ± 14.0 nmol/L, *p* = 0.025. - In mucositis, significantly lower dietary intake of vitamins and calcidiol levels (*p* < 0.02) are observed. - A trend towards greater muscle loss in patients with calcidiol < 50 nmol/L compared with patients with calcidiol ≥50 nmol/L (*p* = 0.07).
Mostafa et al. [316]	- The prospective cross-sectional study; 80 participants: 50 patients with various HNSCC sites and 30 controls (Egyptian population). - Calcidiol serum level.	- A significant decrease of vitamin D3 in cancer group vs. control (calcidiol medians 40.35 vs. 118.75. nmol/L, *p* < 0.001) - Vitamin D3 deficiency (<37.5 nmol/mL) in cases was 42%.
Orell-Kotikangas et al. [289]	- The prospective cohort study; 65 cases of HNSCC and the general population (Finn population). - Calcidiol serum level.	- Hypovitaminosis D3 (37.5–50 nmol/L) was found in 20% of the patients, and vitamin D3 deficiency (<37.5 nmol/L) in 45% of the patients. - No seasonal variation was seen. - In cancer, vitamin D3 deficiency, calcidiol < 50 nmol/l, 65.6% vs. 21.3% (*p* < 0.0001).
Gugatschka et al. [293]	- Case-control study; 88 patients with SCC of the upper aerodigestive tract (European population) and 88 controls. - Calcidiol serum level.	- The serum level of calcidiol was significantly associated with disease-free survival, DFS (RR 0.85, 95% CI: 0.75–0.96, *p* = 0.01) and overall survival, OS (RR 0.89, 95% CI: 0.83–0.97, *p* = 0.006).
Giovannucci et al. [302]	- The Health Professionals Follow-Up Study; 1095 men cohort. - Human cancers of various origins, including aerodigestive tract cancers.	- An increment of 25 nmol/L in predicted calcidiol level was associated with a 17% reduction in total cancer incidence (RR 0.83, 95% CI: 0.74 to 0.92). - An increase in calcidiol level of 62.5 ng/mL was associated with a reduction in the risk of different neoplastic diseases, including head and neck and oesophagus cancers, by >50%.
Weinstein et al. [331]	- The prospective cohort study. - The Alpha-Tocopherol, Beta-Carotene Cancer Prevention (ATBC) Study. - A total of 398 cancer cases (Finnish male smokers). - Multiorgan-specific cancers, including HNC (oral, pharynx, and larynx). - Serum clacidiol measurements.	- Serum calcidiol was significantly lower among cases who subsequently died from their malignancy compared with those who did not. - 25(OH)D3 upper vs. lower quintile (OR 0.74, 95% CI: 0.42–1.30). - Improved survival was also suggested for head and neck cancers, though not statistically significantly.
Skaaby et al. [307]	- The prospective population-based study. - A total of 44 individuals from the Danish Cancer Registry. - HNC, three different cohorts (Monica10, Inter99, Health2006). - Calcidiol serum level.	- No statistically significant associations between calcidiol status (10 nmol/L calcidiol increase) and total or specific cancers, including head and neck carcinoma (HR = 0.97; 95% CI, 0.84–1.12).
Walsh et al. [303]	- Randomised trial; 16 newly diagnosed HNSCC patients (USA population). - Patients treated orally with 4 μg of calcitriol in the 3-week interval between cancer diagnosis and surgical treatment. - 25(OH)D3 serum level and immune cells intra- and peritumoural infiltration.	- Patients with HNSCC who received preoperative treatment with vitamin D3 had a longer time to recurrence, with a median time that was almost 3.5-fold that for the control patients who were untreated before surgery (*p* = 0.048). - The presence of immune cells, i.e., lymphocytes as well as natural killer cells and monocytes expressing an early activation marker CD69, was approximately 10-fold greater (*p* = 0.0007) in tumours from patients who had completed 3 weeks of calcitriol treatment.
Yokosawa et al. [330]	- Prospective cohort study; 434 men HNC patients (USA population) - Dietary or supplemental vitamin D3 intake.	- A significant inverse trend between total vitamin D3 intake and recurrence (HR = 0.47, 95% CI, 0.20–1.10, p_trend_ = 0.048). - Recurrences in 9.2% (≥16.875 μg) and in 12.96% (≤5 μg). - No relationship between dietary or supplemental intake separately. - No association with all-cause or HNC-specific mortality.

FFQ: food frequency questionnaire, QOL: quality of life questionnaire, HNC: head and neck cancer, HNSCC: head and neck squamous cell carcinoma, OSCC: oral squamous cell carcinoma, VDR: vitamin D receptor, HR: the hazard ratio, OR: an odds ratio, RR: relative risk, 95% CI: 95% confidence interval.

## Data Availability

Data available in a publicly accessible repository.

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
