# Peer review of "Role of Vitamin D in Head and Neck Cancer—Immune Function, Anti-Tumour Effect, and Its Impact on Patient Prognosis"

_nutrients, 2023, doi:10.3390/nu15112592_

Round 1

Reviewer 1 Report

This is an extensive interesting review that provides comprehensive depth, into the role vitamin D (specifically calcitriol) in head and neck cancer (HNC)

I have a few minor comments:

Concerning sections 1.2 and 1.3.4 and the effect of vitamin D on immune and inflammatory response: Although informative both sections concern a similar subject on two different sections of the manuscript. I think that the “readability” of this review would benefit if somehow these two sections become one in the proper place inside the manuscript (e.g. 1.2 and 1.3.4. become a concise new section 1.3.4). This will make the text easier to read.

Concerning sections 3.1.1 and 3.1.2: As in other sections of the text, I think that these sections could benefit if their information were summarized in  a table to show the effect of vitamin D in the models in use (cell line/animal): effect of calcitriol, the underlying signaling pathway being modulated/mechanism altered.

Lines 907 - 910:  “Another mechanism is based on calcitriol reducing the expression of prostaglandin E2……… of HIF-1α in cancer cells [231].”. I think the phraseology of this sentence should change because it alters the effect of vitamin D on HIF-1A pathway. Reference 231 by Fukuda reports only the effect of PE2 on HIF1A pathway and angiogenesis not of vitamin D. As it reads , inside the text, it suggests that vitamin D increases HIF1A synthesis. However, the opposite is true. As reported initially by Ben-Shoshan (2007; ref 255) vitamin D impairs HIF-1A pathway and, moreover, as recently reported this negative effect is mediated via PI3K/AKT pathway and independent of VDR (Gkotinakou et al 2020).

The manuscript needs a final proof reading (e.g. in some places e.g. references 1-10 the fonts are not consistent or there are blue fonts out of place (line 2383).

Author Response

Detailed response to Reviewer #1

I  would like to thank you for your considered, substantive and helpful review of my work. All your comments and suggestions have been taken into consideration when improving the work, and that they have made an invaluable contribution to the redrafting and editing of the revised text. A point-by-point answer to the Reviewer’s comments is given below.

This is an extensive - interesting review that provides comprehensive depth, into the role vitamin D (specifically calcitriol) in head and neck cancer (HNC).

I have a few minor comments:

Comment 1: Concerning sections 1.2 and 1.3.4 and the effect of vitamin D on immune and inflammatory response: Although informative both sections concern a similar subject on two different sections of the manuscript. I think that the “readability” of this review would benefit if somehow these two sections become one in the proper place inside the manuscript (e.g. 1.2 and 1.3.4. become a concise new section 1.3.4). This will make the text easier to read.

Answer 1: Separation of section 1.3.4. from 1.2. was the intention of the author. As subchapter 1.3.4 describes an anti-inflammatory effect directly related to the neoplastic environment, it concerns vitamin D3-regulated inflammation in the tumour milieu and is placed in section 1.3: Anti-Cancer Effects of Vitamin D. Chapter 1.2. is included in the general role of vitamin D3 in humans because it provides a general overview of the mechanisms through which vitamin D3 influences the adaptive and innate immune response throughout the body. I hope that the Reviewer will be able to accept the justification for leaving the publication in this way.

Comment 2:  Concerning sections 3.1.1 and 3.1.2: As in other sections of the text, I think that these sections could benefit if their information were summarized in  a table to show the effect of vitamin D in the models in use (cell line/animal): effect of calcitriol, the underlying signaling pathway being modulated/mechanism altered.

Answer 2: I entirely agree. In accordance with the Reviewer’s suggestion, information from the sections 3.1.1 and 3.1.2 has been supplemented by an additional Table 2 summarizing the in vitro and animal studies.

Comment 3: Lines 907 - 910:  “Another mechanism is based on calcitriol reducing the expression of prostaglandin E2……… of HIF-1α in cancer cells [231].”. I think the phraseology of this sentence should change because it alters the effect of vitamin D on HIF-1A pathway. Reference 231 by Fukuda reports only the effect of PE2 on HIF1A pathway and angiogenesis not of vitamin D. As it reads , inside the text, it suggests that vitamin D increases HIF1A synthesis. However, the opposite is true. As reported initially by Ben-Shoshan (2007; ref 255) vitamin D impairs HIF-1A pathway and, moreover, as recently reported this negative effect is mediated via PI3K/AKT pathway and independent of VDR (Gkotinakou et al 2020).

Answer 3: I entirely agree. In accordance with the Reviewer’s suggestion, the phraseology of this sentence has been changed:

“Another mechanism is based on calcitriol reducing the expression of prostaglandin E2 (PGE2) generated by cyclooxygenase-2 (COX-2); this potentially inhibits angiogenesis by decreasing the synthesis of vascular endothelial growth factor (VEGF) and impairing HIF-1α pathway in cancer cells [231].”

I thank the Reviewer for reviewing my work. I hope that the new version complies with the Reviewer’s suggestions, and may be again considered for publication in Nutrients.

Reviewer 2 Report

Dr. Starska-Kowarska Katarzyna in this review summarized the role of vitamin-D in Head and neck cancer.

Although the review is written with lot of information, it is not structured as a review-

Material and methods we write in original article usually not in the review.

Although the review is focused on Head and neck cancer, most of the place it is out of focus and mostly discussed only the biology and physiology of vitamin-D.

In most of the paragraph it is difficult to understand the meaning because of the grammatical mistake.

In most of the paragraph it is difficult to understand the meaning because of the grammatical mistake.

Author Response

Detailed response to Reviewer #2

I  would like to thank you for your considered, substantive and helpful review of my work. All your comments and suggestions have been taken into consideration when improving the work, and that they have made an invaluable contribution to the redrafting and editing of the revised text. A point-by-point answer to the Reviewer’s comments is given below.

Comment 1: Dr. Starska-Kowarska Katarzyna in this review summarized the role of vitamin-D in Head and neck cancer. Although the review is written with lot of information, it is not structured as a review-

Material and methods we write in original article usually not in the review.

Answer 1: Unfortunately, I cannot fully agree with the reviewer's comment. In review articles, Nutrition journal and other journals allow the use of a structure typical for systematic reviews (examples are the author's articles, i.e. PMC: 10046400, PMCID: PMC8838110). The proposed layout of the manuscript allows for separating the theoretical general part of the article, a section containing the data collected so far and confirmed knowledge on the biology and anticancer activity of vitamin D3 and its derivatives (Introduction) and presenting the method of searching for data and selecting references (Materials and Methods) from the detailed part related to the discussed issue in the context of specific human cancers, including the latest laboratory results, clinical studies on HNSCC (Results and Conclusions). In the author's opinion, due to the wide spectrum of activity of vitamin D3 and its analogues in physiological processes and patho-oncological phenomena in human cancers, it seems necessary to familiarize the reader with general knowledge about vitamin D3 for easier and better understanding of its role in the pathogenesis of specific cancers, including HNSCC. In this way, the work becomes more readable and leads the reader step-by-step from physiological mechanisms through in vitro and in vivo research to clinical observations in cancer, taking into account the controversies and limitations of the performed experiments. The author would like to emphasize that the article makes every effort to present the role of vitamin D3 in head and neck cancers in comparison to other human cancers in a possibly comprehensive and reliable way.

Comment 2:  Although the review is focused on Head and neck cancer, most of the place it is out of focus and mostly discussed only the biology and physiology of vitamin-D.

Answer 2: The last decade has seen the considerable growth of new theories regarding the role of the vitamin D3 and its analogues in tumorigenesis, particularly those associated with immune oncology, anti-tumorigenic effects and clinical prognosis. However, despite numerous publications and extensive discussion, the importance of vitamin D3 and its derivatives on initiation, progression and clinical impact in human cancers has not been definitively established. Over the last years, many articles have been published in which the authors present numerous complex and multidirectional mechanisms through which vitamin D3 can regulate pathways crucial for the development of cancer. This also applies to squamous cell cancers, including head and neck cancer.

Arguably, to understand the biology of vitamin D3 in human cancers, researchers and clinicians should look at the subject much more broadly. Limiting knowledge only to the regulatory mechanisms, the chemopreventive efficacy on precancerous lesions and prognostic use of vitamin D3 plasma concentrations or dietary intake strictly related to head and neck cancer obviously narrows the scientific perspective and makes it impossible to fully understand the complex role of vitamin D3 and its analogues in the pathogenesis of human cancer. Therefore, all papers on this issue make an important contribution to the dissemination of knowledge about the role of vitamin D3 in carcinogenesis, including the pathogenesis of head and neck cancer; they also facilitate a better understanding of the complex, multidirectional intracellular signalling and intercellular relations, which can often be obscure for clinicians i.e. ENT-oncologists, as well as the latest results from original research and clinical trials.

The article is intended to summarize current knowledge and recent discoveries regarding the role of the vitamin D3 on immune function, anti-tumour effect and its impact on carcinogenesis in HNSCC against the background of other cancers.

Additionally, while review papers are intended to be a compendium of knowledge on issues selected by the author of the article, each author of a review paper also hopes that their work will become a source of inspiration; in this case, it is hoped that the article provokes further important questions about carcinogenesis in the readership and a desire to seek answers to key issues. Such inspiration may open doors for new therapeutic strategies to help prevent malignant disease.

Comment 3: In most of the paragraph it is difficult to understand the meaning because of the grammatical mistake.

Answer 3: I entirely agree. In accordance with the Reviewer’s suggestion, the text of the manuscript was corrected by a professional native speaker translator at the university.

I thank the Reviewer for reviewing my work. I hope that the new version complies with the Reviewer’s suggestions, and may be again considered for publication in Nutrients.

Reviewer 3 Report

I appreciate the opportunity to review this article due to the relevant context of vitamin D and head and neck cancer. 

The mega review was prepared covering subjects that generate complexity in the reading; therefore, the text is hard to protect all sub-items. There is a strong focus on vitamin D metabolism, the immune response, and cancer mechanisms. However, understanding the main messages of the review needs to be better brought to readers' attention.

Overall, the organization of sessions should be modified to attract readers.

However, I present some considerations:

First, I suggest a reformulation in the distribution of sessions.

1. The abstract must present the subject's objective, methodology, primary results, and consensus/controversies. Unfortunately, it appears very timid in the face of such a rich text.

2. Summarize to a maximum of 10 keywords. For example, micronutrient, nutritional supplement, and oxidative stress make no sense.

3. The Introduction should be concise, contextualizing the review's objective and novelty about what exists in the literature and highlighting mainly the results of systematic reviews, which are more representative in providing evidence. What is the applicability of the summarized knowledge? This session serves to situate the reader in what will be described throughout the text and not to "develop" the theme, as the author did.

4. Methods section comes next.

5. Development of review with subitem items as they discuss: (1) Basic aspects of vitamin D (VitD); (2) VitD and the immune system, (3) VitD and anticancer effects, (4) Results - elaborate a more concise title and check repetitions of contents in the subitems ( 3.2). The author could gather the human studies more objectively, separating epidemiological interventions and RCTs. It is essential to add (6) Limitations and (6) "Status of VitD and Prognosis of head and neck cancer (pieces of evidence) that I understand to be of most significant interest for clinical application. Item (7) Conclusions should be reformulated in a more concise format, without going back to discussing results and mechanisms, but focusing on existing evidence or not on the subject and applicability for research and clinical practice. Ending with well-structured conclusions and recommendations.

Item 1.1.4. Vitamin D Supplementation includes VitD diagnostic parameters. It is advisable to separate these subtopics due to the diagnostic complexity marked by the various guidelines worldwide. The reader needs to know this before analyzing the results of the supplementation studies.

Congratulations to the author for the excellent figures and tables! Check all abbreviations at the bottom of the tables.

Other points:

- I consider excessive abbreviations in the text. The ideal would be to present the list of acronyms and exclude unnecessary ones, as this breaks the flow of reading and hinders understanding. Still, in this sense, many terms already described in the first citation reappear in full instead of just citing the abbreviation (for example, in lines 51, 95, 179, 203, 358, 318, 370, 485, etc.).

- Standardize throughout the text the denomination and abbreviations of the metabolites and forms of vitamin D. It is enough to quote once on denominations - calcidiol or calcifediol and not always repeat the chemical structure and the name (lines 177, 178).

Author Response

Detailed response to Reviewer #3

I  would like to thank you for your considered, substantive and helpful review of my work. All your comments and suggestions have been taken into consideration when improving the work, and that they have made an invaluable contribution to the redrafting and editing of the revised text. A point-by-point answer to the Reviewer’s comments is given below.

I appreciate the opportunity to review this article due to the relevant context of vitamin D and head and neck cancer.

The mega review was prepared covering subjects that generate complexity in the reading; therefore, the text is hard to protect all sub-items. There is a strong focus on vitamin D metabolism, the immune response, and cancer mechanisms. However, understanding the main messages of the review needs to be better brought to readers' attention.

Overall, the organization of sessions should be modified to attract readers.

However, I present some considerations:

Comment 1: First, I suggest a reformulation in the distribution of sessions.

The abstract must present the subject's objective, methodology, primary results, and consensus/controversies. Unfortunately, it appears very timid in the face of such a rich text.

Summarize to a maximum of 10 keywords. For example, micronutrient, nutritional supplement, and oxidative stress make no sense.

Answer 1: I entirely agree. In accordance with the Reviewer’s suggestion, the text of the abstract has been supplemented with the recommended details and the number of keywords have been reduced. However, the author would like to point out that according to the guidelines of the Editor of Nutrients journal, the abstract is limited to 150 words, which obviously limits the possibility of describing in more detail the various and multi-directional issues discussed in the publication.

Comment 2:  The Introduction should be concise, contextualizing the review's objective and novelty about what exists in the literature and highlighting mainly the results of systematic reviews, which are more representative in providing evidence. What is the applicability of the summarized knowledge? This session serves to situate the reader in what will be described throughout the text and not to "develop" the theme, as the author did.

Answer 2: Unfortunately, I cannot fully agree with the reviewer's comment. The author's deliberate intention was to use the proposed structure of the work. This review structure has already been used and accepted in the author's published articles, including in the journal Nutrition and others (examples are articles, i.e. PMC: 10046400, PMCID: PMC8838110). After reading the Abstract the reader is introduced to the subject matter of the publication, the current literature selection methods used and the issues discussed. The proposed layout of the manuscript allows for separating the theoretical general part of the article, a section containing the data collected so far and confirmed knowledge on the biology and anticancer activity of vitamin D3 and its derivatives (Introduction) and presenting the method of searching for data and selecting references (Materials and Methods) from the detailed part related to the discussed issue in the context of specific human cancers, including the latest laboratory results, clinical studies on HNSCC (Results and Conclusions).

Due to the wide spectrum of activity of vitamin D3 and its analogues in physiological processes and patho-oncological phenomena in human cancers, it is arguably necessary to familiarize the reader with general knowledge about vitamin D3 to provide a better understanding of its role in the pathogenesis of specific cancers, including HNSCC. In this way, the work becomes more readable and leads the reader step-by-step from physiological mechanisms through in vitro and in vivo research to clinical observations in cancer, taking into account the controversies and limitations of the experiments carried out.

The Introduction chapter has been supplemented with a short summary of recent data and novelty about what exists in the literature and mainly highlights the results of epidemiological and systematic reviews on the role of vitamin D3 in HNC with the aim of providing an introduction to a more detailed presentation of the discussed issues. A more in-depth examination of existing systematic reviews, and epidemiological, cross-sectional, longitudinal, prospective and interventional studies, even a cursory one, would be a repetition of the knowledge discussed in depth later in the publication. The author would like to emphasize that every effort has been made to present the role of vitamin D3 in head and neck cancers in comparison to other human cancers in a possibly comprehensive and reliable way.

Comment 3: Methods section comes next.

Development of review with subitem items as they discuss:

(1) Basic aspects of vitamin D (VitD); (2) VitD and the immune system, (3) VitD and anticancer effects,

Answer 3: I entirely agree. In accordance with the Reviewer’s suggestion, the text of the manuscript was corrected and supplemented.

As suggested, the text of the publication has been redacted and includes the following chapter: 1.1. Basic aspects of vitamin D - The Biochemistry and Physiology of Vitamin D, 1.2. Vitamin D Supplementation, 1.3. Vitamin D and the Immune System, and 1.4. Vitamin D and Anti-cancer Effects

(4) Results - elaborate a more concise title and check repetitions of contents in the subitems (3.2). The author could gather the human studies more objectively, separating epidemiological interventions and RCTs.

As suggested, the title of section 3.2. has been changed to be more concise: 3.2. Studies on the Role of Vitamin D as Predictors of Cancer Risk, Progression and Prognosis in HNC. The Chemopreventive Efficacy of Vitamin D on Precancerous Lesions. To emphasize the role of vitamin D3 in the risk of HNC and further development of cancer, and thus the prognosis, these issues have been kept in the title of the chapter. For the same reason, two subsections have been introduced: 3.2.1. on risk and chemoprevention and 3.2.2. on HNC mortality, survival and recurrence. Section titles 3.2.1. and 3.2.2. have also been shortened.

The data from the present review, taking into account the characteristics of studies, including epidemiological analyses, key opinion-forming systematic reviews, as well as epidemiological, cross-sectional, longitudinal, prospective and interventional studies (RCTs), are described in detail in the text and summarized in Table 3 (before review Table 2). The aim was to discuss individual study results in relation to clinical aspects, i.e. risk, progression and prognosis in patients, as reflected in the following subchapters, rather than analyzing data in terms of study type; this is particularly appropriate as many simply describe laboratory and clinical studies. I hope that this approach will be accepted by the Reviewer.

It is essential to add (6) Limitations and (6) "Status of VitD and Prognosis of head and neck cancer (pieces of evidence) that I understand to be of most significant interest for clinical application.

In accordance with the Reviewer’s suggestion, the text lists the controversies and limitations of the cited laboratory and clinical studies. For better emphasis on the issues discussed and for easier identification, the limitations for in vitro/in animal studies and clinical trials are placed directly under the relevant chapters. Sections "Limitations of in vitro and animal studies” and “Limitation of clinical studies" have been included in the text. Furthermore, in the text of the work, an extensive chapter on the meaning of Vitamin D Plasma Concentrations, Vitamin D Intake and VDR Gene Polymorphism as Predictors of HNC Mortality, Survival and Recurrence was distinguished. The author would like to emphasize that every effort has been made to reliably present the role of vitamin D3 in the prognosis of patients with head and neck cancer.

Item (7) Conclusions should be reformulated in a more concise format, without going back to discussing results and mechanisms, but focusing on existing evidence or not on the subject and applicability for research and clinical practice. Ending with well-structured conclusions and recommendations.

In accordance with the Reviewer’s suggestion, the Conclusions section has been reformulated in a more concise manner. The previously discussed results and mechanism were removed, focusing on existing evidence and applicability in research and clinical practice.

Item 1.1.4. Vitamin D Supplementation includes VitD diagnostic parameters. It is advisable to separate these subtopics due to the diagnostic complexity marked by the various guidelines worldwide.

As suggested, the section Vitamin D Supplementation has been replaced and numbered 1.2.
Furthermore, the subsection 1.2.1. Vitamin D Status as A Diagnostic Parameter has also been added in the text.

Congratulations to the author for the excellent figures and tables! Check all abbreviations at the bottom of the tables.

As suggested, all abbreviations at the bottom of the tables have been checked and supplemented.

Other points:

- I consider excessive abbreviations in the text. The ideal would be to present the list of acronyms and exclude unnecessary ones, as this breaks the flow of reading and hinders understanding. Still, in this sense, many terms already described in the first citation reappear in full instead of just citing the abbreviation (for example, in lines 51, 95, 179, 203, 358, 318, 370, 485, etc.).

According to the Reviewer's suggestion, the work has been edited so that the terms already described in the first quote appear again in the text in the form of an abbreviation.

- Standardize throughout the text the denomination and abbreviations of the metabolites and forms of vitamin D. It is enough to quote once on denominations - calcidiol or calcifediol and not always repeat the chemical structure and the name (lines 177, 178).

According to the Reviewer's suggestion, the denomination and abbreviations of the metabolites and forms of vitamin D were standardized, i.e. the names calcidiol and calcitriol were used, without repeating names and chemical formulas in the text.

I thank the Reviewer for reviewing my work. I hope that the new version complies with the Reviewer’s suggestions, and may be again considered for publication in Nutrients.

Round 2

Reviewer 3 Report

I congratulate the author for improving the text and organizing the sessions. In this way, I understand that the previous version of the abstract will be deleted and replaced with the new version (highlighted in yellow). As it is presented, the summary has been expanded with an overlay of the previous text.

The questions were satisfactorily answered, making the review suitable for publication in Nutrients.